# Theoretical Guarantees for Iterative Alignment of Self-Rewarding Language Models

## Abstract

Self-Rewarding Language Models (SRLMs) achieve notable success in iteratively improving alignment without external feedback. Yet, despite their striking empirical progress, the core mechanisms driving their capabilities remain unelucidated, leaving a critical gap in theoretical understanding. This paper provides the first rigorous theoretical guarantees for SRLMs. We first establish a lower bound that characterizes the fundamental limits of a single update step, revealing a critical dependence on the quality of the initial model. We then derive finite-sample error bounds for the full iterative paradigm, showing that performance improves at a rate of $\mathcal{O}\left(1/\sqrt{n}\right)$ with sample size $n$. Crucially, our analysis reveals that the dependence on the initial model decays exponentially with the number of iterations $T$. This provides a formal explanation for *why* iterative self-rewarding succeeds: it robustly overcomes the limitations of a poor initialization. Finally, we instantiate our theoretical framework for the linear softmax model class, yielding tailored guarantees that connect our high-level insights to practical model architectures.

## 1 Introduction

Contemporary language models have achieved unprecedented success in numerous areas of natural language processing. Aligning these powerful models with human preferences is critical for their safe deployment, a task conventionally addressed by Reinforcement Learning from Human Feedback (RLHF) (Ouyang et al., 2022; Rafailov et al., 2023). However, the efficacy of RLHF is fundamentally predicated on the availability of large-scale, high-quality human preference data. This reliance introduces two major challenges: first, the process of collecting human annotations is expensive and difficult to scale (Gao et al., 2023); second, the inherent cognitive limitations of human evaluators may cap the performance of models intended to achieve superhuman capabilities (Burns et al., 2024). Consequently, this dependence on external supervision forms a critical bottleneck, hindering the development of more autonomous and capable AI systems (Huang et al., 2022; 2025).

To overcome these limitations, the paradigm of Self-Rewarding Language Models (SRLMs) has emerged, demonstrating considerable potential (Yuan et al., 2024; Wu et al., 2024; Prasad et al., 2025). This approach facilitates iterative self-alignment without external feedback by enabling the language model to serve as both its own policy and reward model. Within this framework, the model generates candidate responses and leverages its intrinsic judgment to assign rewards, using this self-generated supervision to produce a refined policy that serves as the reward model for the subsequent iteration. The efficacy of this approach has been empirically validated in recent studies (Yuan et al., 2024; Zhou et al., 2024; Wang et al., 2025; Xiong et al., 2025; Li et al., 2025), which report significant performance gains across a variety of tasks.

However, despite the remarkable empirical success of SRLMs, our understanding of their core working mechanisms remains limited. Why do these models achieve stable and continuous improvement rather than succumbing to degeneration or even collapse (Shumailov et al., 2024)? Current research largely remains at the level of empirical observation and methodological refinement, lacking a solid theoretical foundation to explain the reasons for their success or to characterize their performance boundaries. This theoretical gap prevents us from deeply understanding the capability limits and potential risks of SRLMs, and it constrains our ability to pursue more profound improvements.

To fill this critical theoretical void, this paper provides the first rigorous theoretical guarantees for the iterative self-rewarding alignment process in language models. Our main contributions can be summarized as follows:

- **Fundamental Limitations of Single-Step Updates.** We establish a theoretical lower bound on the failure probability of a single-step update, proving its critical dependence on the initial model's quality. This result theoretically explains why poorly initialized models fail to achieve effective alignment in one step.

- **Finite-Sample Guarantees for Iteration.** We derive finite-sample error bounds for the iterative self-rewarding paradigm, proving that its performance steadily improves at a rate of $\mathcal{O}\left(1/\sqrt{n}\right)$ as the sample size $n$ increases. Our analysis explicitly shows that the dependence on the initial model's quality decays exponentially with the number of iterations $T$.

- **The Core Mechanism of Iterative Alignment.** We identify the core mechanism for overcoming poor initialization: the iterative update acts as a contraction mapping on the *policy condition number* $\kappa_t$, which captures both the model's internal consistency and the stability of self-alignment. This causes the influence of the initialization to vanish exponentially, allowing the process to first self-stabilize before improving performance, thus bypassing the single-step failure barrier.

- **Application to Linear Softmax Models.** We apply our general theoretical framework to the linear softmax model class, deriving performance guarantees for this specific model architecture. This connects our theoretical insights with practical application.

## 2 RELATED WORK

In this section, we review the literature most relevant to our work. We begin by discussing recent advancements in SRLMs and then survey the existing theoretical guarantees for self-training.

**Self-Rewarding Language Models.** Existing alignment methods, particularly RLHF, heavily rely on high-quality reward models or continuous human feedback, which creates a major bottleneck for scalability. To overcome this, Yuan et al. (2024) proposed the SRLM paradigm, which leverages the language model itself to act as both the policy and the reward model. In this framework, the policy model generates response candidates for unlabeled prompts, while the reward model uses the LLM-as-a-Judge (Zheng et al., 2023; Wataoka et al., 2024; Gu et al., 2024) to score these responses based on their quality. This process can be iteratively repeated to improve alignment performance without human intervention (Zhao et al., 2025; Xiong et al., 2025; Chen et al., 2024).

Recent work has focused on further refining the SRLM framework. To improve the quality of the generated rewards, Wu et al. (2024) suggested using the same LLM as a meta-judge to evaluate its own LLM-as-a-Judge outputs, while Zhang et al. (2025) introduced a step-wise LLM-as-a-Judge approach. Others have focused on enhancing training stability and data quality. For instance, Wang et al. (2025) introduced regularization to enhance the consistency of DPO rewards across different iterations, thus providing more robust preference data. Additionally, Prasad et al. (2025) and Zhou et al. (2025) focused on enhancing consistency to improve the reliability of both the reward models and the preference data. The paradigm has also been extended to new modalities, with Zhou et al. (2024) and Li et al. (2025) successfully applying it to Vision-Language Models (VLMs). However, despite significant empirical progress and effectiveness, to the best of our knowledge, a complete and rigorous theoretical analysis of SRLMs is still missing. This gap prevents a deep understanding of the internal mechanisms that explain why this paradigm is successful.

**Theoretical Guarantees for Self-Training.** Theoretical work on self-training remains limited. Existing research has predominantly focused on the self-distillation objective (Hinton et al., 2015) in classification and regression tasks, aiming to provide convergence guarantees. Several studies have established theoretical guarantees for self-training under idealized settings, such as linear models (Mobahi et al., 2020; Frei et al., 2022; Das & Sanghavi, 2023; Pareek et al., 2024). Furthermore, Allen-Zhu & Li (2023) offered guarantees for feed-forward neural networks, while Boix-Adsera (2024) proposed a more general PAC-style framework. Another line of related work investigates the problem of model collapse in self-consuming training loops (Alemohammad et al., 2024; Bertrand et al., 2024). These works explore this direction by analyzing population risk dynamics under specific modeling assumptions, such as linear contexts (Dohmatob et al., 2024; Feng et al., 2025), Gaussian models (Shumailov et al., 2024; Alemohammad et al., 2024; Suresh et al., 2024; Jain et al., 2024), and

asymptotic regimes (Marchi et al., 2024). Further explorations have been conducted in the context of simplified diffusion models (Fu et al., 2024) and attention-based architectures (Fu et al., 2025). In the area of self-improvement, Huang et al. (2025) proposed a sharpening mechanism as a key driver of self-improvement and extended the theoretical framework of self-training to language models. However, their work is confined to a non-iterative setting.

In contrast to all prior work, this paper establishes the first rigorous theoretical framework for the iterative alignment paradigm of SRLMs. By deeply analyzing the underlying mechanisms that enable stable and progressive performance improvements, we theoretically explain the success of this paradigm and provide stringent convergence rates and performance guarantees.

## 3 PRELIMINARIES

This section establishes the notation and theoretical background. We formally define the core components of the SRLM paradigm (Yuan et al., 2024; Wang et al., 2025), to be the model class, the reward mechanism, and the DPO-style training objective that governs model updates.

**Notations and Setup.** Let $x \in \mathcal{X}$ denote a prompt and $y \in \mathcal{Y}$ be a response, where $\mathcal{Y} = \mathcal{V}^H$ for a vocabulary space $\mathcal{V}$ and a sequence length $H$. A language model is a conditional distribution $\pi : \mathcal{X} \to \Delta(\mathcal{Y})$ that maps a prompt $x$ to a distribution over responses. We assume prompts are drawn from a distribution $\mu$ over the prompt space $\mathcal{X}$. The term $\pi(y \mid x)$ denotes the probability that the model generates response $y$ given $x$. The self-rewarding process iteratively applies updates at each round $t$, starting from an initial model $\pi_0$ and yielding the sequence of models $\{\pi_0, \pi_1, \ldots, \pi_T\}$.

**Self-Reward Mechanism and Data Generation.** The process relies on a self-reward signal generated by the model itself. This approach operationalizes the concept of LLM-as-a-Judge (Zheng et al., 2023; Bai et al., 2023) by using the model's own likelihood assignment as a proxy for its judgment on the quality of a generation. Specifically, the reward for a response $y$ given a prompt $x$ under the current model $\pi_t$ is its log-probability:

$$J_{\text{self}}(y \mid x, \pi_t) = \log \pi_t(y \mid x), \qquad r_t(y \mid x) := J_{\text{self}}(y \mid x, \pi_t). \tag{1}$$

This reward function encourages the model to assign higher probabilities to sequences it already deems likely, effectively reinforcing what it judges to be high-quality regions of the response space. To prepare for the update at round $t$, a dataset $\mathcal{D}_t = \{(x, y, y')\}$ of $n$ examples is constructed. Each example is formed by first sampling a prompt $x \sim \mu$, then generating two responses $y, y' \sim \pi_t(\cdot \mid x)$. Subsequently, their relative quality is measured by the self-reward difference, $\Delta J_t(x, y, y') = J_{\text{self}}(y \mid x, \pi_t) - J_{\text{self}}(y' \mid x, \pi_t)$. The stylized self-reward function $r_t(x, y) = \log \pi_t(y \mid x)$ captures the core feedback loop where the model reinforces its own current beliefs. While this formulation abstracts away richer judging mechanisms used in engineering practice (e.g., meta-judges, rubric-based scoring, multi-turn evaluation), we believe it serves as a necessary starting point toward understanding forms of self-rewarding that rely on more sophisticated judges but are less amenable to direct theoretical analysis.

**Model Update via DPO.** The goal of each training round is to improve the model via a KL-regularized objective. We seek a new model $\pi_{t+1}$ that maximizes the expected reward $r_t$ while remaining close to the current model $\pi_t$, which acts as the reference $\pi_{\text{ref}}$. This is formulated as:

$$\pi_{t+1} \in \arg\max_{\pi \in \Pi} \mathbb{E}_{x \sim \mu} \mathbb{E}_{y \sim \pi(\cdot|x)}[r_t(x, y)] - \frac{1}{\beta} \mathbb{E}_{x \sim \mu}[D_{\text{KL}}(\pi(\cdot \mid x) \| \pi_{\text{ref}}(\cdot \mid x))]. \tag{2}$$

In practice, this optimization can be achieved by minimizing a DPO-style regression objective (Rafailov et al., 2023; Gao et al., 2024; Huang et al., 2025). Specifically, for round $t$ where the reference model is $\pi_{\text{ref}} = \pi_t$, the loss function is defined as:

$$\mathcal{L}_t^{\text{DPO}}(\pi) = \frac{1}{n} \sum_{(x, y, y') \in \mathcal{D}_t} \left( \beta \left[ \log \frac{\pi(y \mid x)}{\pi_{\text{ref}}(y \mid x)} - \log \frac{\pi(y' \mid x)}{\pi_{\text{ref}}(y' \mid x)} \right] - \Delta J_t(x, y, y') \right)^2. \tag{3}$$

Minimizing this loss, $\mathcal{L}_t^{\text{DPO}}$, effectively executes the model improvement step outlined in Eq. 2. This entire update can be conceptualized as the application of an operator $\mathcal{T}_{r_t} : \Pi \to \Pi$, which maps the

current model $\pi_t$ to the improved model $\pi_{t+1}$:

$$\pi_{t+1} = \mathcal{T}_{r_t}(\pi_t) \quad \text{where} \quad \mathcal{T}_{r_t}(\pi_t) = \arg\min_{\pi \in \Pi} \mathcal{L}_t^{\text{DPO}}(\pi) \text{ with } \pi_{\text{ref}} = \pi_t. \tag{4}$$

After $T$ rounds, the final model is the result of composing these operators: $\pi_T = \left(\mathcal{T}_{r_{T-1}} \circ \cdots \circ \mathcal{T}_{r_0}\right)(\pi_0)$. Since the reward $r_t$ is determined by $\pi_t$, the training loop is entirely self-contained.

## 4 FUNDAMENTAL LIMITATIONS OF SINGLE-STEP SELF-REWARDING

This section introduces the Policy Condition Number[1], a metric designed to quantify a model's suitability for self-alignment. It characterizes both the model's internal consistency and the numerical stability of the self-rewarding update process. Building on this concept, we derive a formal lower bound on the failure rate of a single update step, revealing its fundamental limitations. Furthermore, we show how these learning failures inevitably translate into inference errors under greedy decoding, highlighting the necessity of an iterative framework.

The policy condition number, denoted $\kappa_t$, is formally defined as the expected inverse probability of the policy's own most likely response:

$$\kappa_t := \mathbb{E}_{x \sim \mu}\left[\frac{1}{\pi_t(y_t^*(x) \mid x)}\right], \tag{5}$$

where $y_t^*(x) := \arg\max_{y \in \mathcal{Y}} \pi_t(y \mid x)$. Drawing an analogy from numerical analysis, this parameter measures how well-conditioned the policy is for self-reward. A large value of $\kappa_t$ signifies an *ill-conditioned* policy, one that is diffuse and lacks confidence in its own predictions by assigning low probability to its modal outputs. This ill-conditioning makes the single-step update unreliable and heightens the risk of model collapse (Shumailov et al., 2024). A central finding of our work is that while a large initial condition number $\kappa_0$ can lead to failure, the iterative process progressively controls this quantity and ensures that its influence diminishes over time.

We now present a formal lower bound on the failure rate for any self-rewarding algorithm operating within a single iteration. The result is established by considering a worst-case instance where subtle statistical signals are intentionally obscured by an information-sparse prompt distribution.

**Theorem 1** (Single-Step Failure Rate Lower Bound). *Let any self-rewarding algorithm produce a model $\pi_1$ using $n$ samples generated from a base model $\pi_0$. There exists a hard problem instance, characterized by a model class $\Pi$ and an initial policy condition number $\kappa_0$, such that the failure rate of any algorithm with a fixed budget $n$ is lower-bounded. Failure is defined as the event that the policy assigns a probability of at most $1/2$ to its own optimal response $y_1^*(x) := \arg\max_{y \in \mathcal{Y}} \pi_1(y \mid x)$. The risk-form of the lower bound is given by:*

$$\mathbb{P}_{x \sim \mu}\left[\pi_1(y_1^*(x) \mid x) \leq \frac{1}{2}\right] \gtrsim \left(\frac{\kappa_0 \log |\Pi|}{n}\left[\log\left(\frac{n\kappa_0}{\log |\Pi|}\right)\right]^{-1}\right)^{1/2}. \tag{6}$$

**Remark 1.** Theorem 1 formalizes the statistical barriers of a single self-rewarding step. The lower bound contains a slowly varying logarithmic correction term, $\left[\log(\frac{n\kappa_0}{\log |\Pi|})\right]^{-1/2}$, whose growth is significantly slower than the dominant polynomial dependence, $\left(\frac{\kappa_0 \log |\Pi|}{n}\right)^{1/2}$. For ease of interpretation, we focus on this dominant term, which gives the simplified bound $\left(\frac{\kappa_0 \log |\Pi|}{n}\right)^{1/2}$, up to logarithmic factors. This expression makes clear that when the effective difficulty $\kappa_0 \log |\Pi|$ is on the same order as the sample size $n$, the failure rate lower bound remains a constant. In such cases, a single self-rewarding update cannot reliably improve the model, regardless of the algorithmic details.

Two concrete scenarios illustrate when $\kappa_0$ can be on the same order as $n$:

---

[1]This quantity is analogous to what is sometimes called the concentrability or coverage coefficient, a concept studied in the theory of offline reinforcement learning and self-improvement (Xie et al., 2023; Gao et al., 2024; Amortila et al., 2024; Huang et al., 2025).

- *Near-uniform base policy.* Suppose the base model $\pi_0$ assigns a nearly uniform distribution over a large response space $\mathcal{Y}$ of size $M$. Then the most likely response $y^*(x)$ has probability only slightly larger than $1/M$, so

$$\kappa_0 \asymp M.$$

  If the sample budget $n$ is not significantly larger than $M$, then $\kappa_0/n$ is a constant, leading to a constant failure probability.

- *Autoregressive low-confidence base policy.* Consider an $H$-step autoregressive model where at each step the top-1 token probability is at most $\alpha \in (0, 1)$. Along the modal path, the probability of the full sequence satisfies $\pi_0(y^*(x) \mid x) \leq \alpha^H$, so that

$$\kappa_0 \gtrsim \alpha^{-H}.$$

  If we set $H \asymp \log n / \log(1/\alpha)$, then $\alpha^{-H} \asymp n$, which implies that $\kappa_0$ is of the same order as the sample budget $n$, leading to a failure rate lower bounded away from zero.

These examples show that diffuse or low-confidence base policies naturally yield $\kappa_0$ on the scale of $n$, which makes a single self-rewarding update statistically unreliable.

The failure condition $\pi_1\big(y_1^*(x) \mid x\big) \leq \frac{1}{2}$, adopted in Theorem 1, is a natural threshold. As the subsequent proposition reveals, it precisely identifies the circumstances under which greedy decoding fails. Consequently, the lower bound reflects a meaningful and practically relevant notion of decoder breakdown.

**Proposition 2** (Greedy Decoding Fails for Unaligned Models). *There exist autoregressive models $\pi$ over the space $\mathcal{Y} = \mathcal{V}^H$ and prompts $x$ for which the optimal sequence $y^*(x)$ is unique, and its probability satisfies $\pi(y^*(x) \mid x) \leq 1/2$. Then, the greedy decoding strategy, defined as $\hat{y}_h = \arg\max_{y \in \mathcal{V}} \pi(y \mid \hat{y}_{<h}, x)$ for all $h \in [H]$, fails to recover the optimal sequence, i.e., $\hat{y} \neq y^*(x)$.*

**Remark 2.** Proposition 2 demonstrates the direct and severe consequence of a low-confidence model at inference time. It establishes that for certain models, standard decoding algorithms such as greedy search are guaranteed to fail. This failure occurs because the model's confidence in its single best prediction is outweighed by its collective uncertainty over all other alternatives, allowing a locally optimal yet globally incorrect token to trap a myopic decoder and ensure a suboptimal sequence.

This result bridges *learning failure* and *inference failure*. The low-confidence condition is not a pathological corner case but a statistically predictable outcome of learning. As shown by Theorem 1, a single self-rewarding update initialized with a poorly conditioned base policy (large policy condition number $\kappa_0$) has a non-negligible probability of producing exactly such a low-confidence model.

Taken together, the two results yield a stark conclusion. When the effective difficulty satisfies $\kappa_0 \asymp n$, Theorem 1 implies a constant-order lower bound on the probability of learning failure. Consequently, for sufficiently weak initialization, there is a constant probability that a single update produces a policy that, by Proposition 2, is certain to output low-quality sequences under greedy decoding. This cascade of failure, from a high probability of learning error to a deterministic inference error, exposes the fundamental vulnerability of single-step self-improvement and its critical dependence on the initial model. This motivates the iterative self-rewarding framework developed in the subsequent sections, which progressively mitigates this dependence and enables robust alignment even from weak initialization.

## 5 FINITE-SAMPLE ANALYSIS FOR ITERATIVE SELF-REWARDING ALIGNMENT

This section addresses the fundamental limitations of single-step updates identified in Section 4 by developing a theoretical framework for iterative self-rewarding alignment. Within this framework, we establish finite-sample guarantees showing that sustained improvements across iterations overcome the fragility of a single step, yielding a provably stable process that ensures robust alignment even from poorly conditioned initial models. To formalize this, we first introduce a standard realizability assumption and define key analytical constants.

**Assumption 1** (Realizability). Let $\pi_\beta^*$ be the optimal model that maximizes the KL-regularized objective in Eq. 2. We assume that this model is contained within our model class, i.e., $\pi_\beta^* \in \Pi$.

Assumption 1 is a natural and standard requirement in statistical learning theory (Vapnik, 2013; Shalev-Shwartz & Ben-David, 2014), ensuring that the target model lies within the considered class. Then, we define analytical constants that capture the stability of iterative self-rewarding updates.

**Definition 1** (Confidence and Margin Constants). We introduce two analytical constants that quantify the stability of self-rewarding updates:

- **Minimum confidence.** The minimum confidence across all inputs $x \in \mathcal{X}$ and update rounds $t \in [0, T]$ is defined as

$$c := \inf_{x \in \mathcal{X}, \, t \in [0,T]} \pi_t(y_t^*(x) \mid x).$$

- **Margin constant.** The margin $\gamma > 0$ quantifies the minimum log-probability gap between the optimal action and the strongest suboptimal competitor across all rounds:

$$\gamma := \inf_{x \in \mathcal{X}, \, t \in [0,T]} \log\left( \frac{\pi_t(y_t^*(x) \mid x)}{\max_{y' \neq y_t^*(x)} \pi_t(y' \mid x)} \right).$$

The minimum confidence $c > 0$ ensures that the model maintains a non-trivial level of certainty in its predictions throughout the iterative process. This requirement is standard in related literature (Zhang et al., 2023; Huang et al., 2025). The margin $\gamma$ quantifies the separation between the optimal action and its closest competitor. The unique optimal sequence assumption implies a strictly positive margin, since identifiability requires the optimal response to be distinguishable from suboptimal ones. In continuously parameterized models, exact ties ($\gamma = 0$) form a measure-zero set and are therefore negligible in practice. Even when margins are extremely small, the iterative contraction of $\kappa_t$ ensures stability, while the DPO update and the self-reward mechanism amplify these small gaps over time, preventing collapse.

In the context of a practical large language model, the margin constant $\gamma$ can be viewed as a signal-to-noise ratio for the model's internal preferences: when $\gamma$ is small, the model behaves like a hesitant judge whose scores for the top candidate and the runner-up are nearly indistinguishable, so the induced reward signal is noisy and gradient estimates have high variance; when $\gamma$ is large, the model exhibits a clear preference gap, yielding low-variance, stable learning signals. The minimum confidence $c$ plays a complementary role as a safeguard against collapse: as long as $c$ stays bounded away from zero, the iterative self-rewarding process can, by the law of large numbers, extract a reliable preference signal from a large pool of self-generated samples and thereby gradually overcome initialization bias through statistical concentration.

**Theorem 3** (Finite-Sample Guarantee for Iterative Self-Rewarding Alignment). *Suppose Assumption 1 holds, and the regularization parameter is set to $\beta \lesssim 1/\sqrt{n}$. For any $\delta, \rho \in (0, 1)$, after $T$ rounds of iterative self-rewarding updates as defined in Eq. 4, the resulting model $\pi_T$ ensures that with probability at least $1 - \rho$,*

$$\mathbb{P}_{x \sim \mu}[\pi_T(y_T^*(x) \mid x) \leq 1 - \delta] \lesssim \frac{1}{\sqrt{n}} \cdot \frac{\log\left(n|\Pi|\rho^{-1}\right)}{\gamma\delta} \left( \frac{1}{\sqrt{c}} + \frac{\sqrt{\kappa_0}}{(1 + \sqrt{nc})^{(T-1)/2}} \right). \quad (7)$$

**Remark 3.** Theorem 3 provides the first finite-sample guarantee for the iterative self-rewarding alignment paradigm, establishing a foundational theoretical framework for a class of methods that, despite their empirical success, have lacked rigorous understanding. To further clarify the asymptotic behavior of Eq. (7): When the sample size $n$ is sufficiently large and the number of iterations $T$ is large enough (as shown in Corollary 4, i.e., $T \gtrsim \frac{\log(c\kappa_0)}{\log(1+\sqrt{nc})}$), the initialization-dependent term vanishes. Consequently, up to logarithmic factors and problem-dependent constants $(c, \gamma)$, the error bound simplifies to $O(\log n/\sqrt{n})$, demonstrating that the iterative process achieves the standard parametric convergence rate. The theorem formally quantifies the risk that the final model, $\pi_T$, assigns insufficient probability mass to its own optimal response after $T$ rounds of updates. The structure of the bound in Eq. 7 reveals a compelling interplay between two key error components. The overall error probability is upper-bounded by a product of the standard statistical learning rate, $\log(n)/\sqrt{n}$, and a term, $1/\sqrt{c} + \sqrt{\kappa_0}/(1 + \sqrt{nc})^{(T-1)/2}$, that comprises both a stable and a transient error component. The constituents of this bound can be interpreted as follows:

- *Statistical Efficiency:* The $\log(n)/\sqrt{n}$ rate confirms that the iterative process is statistically efficient, with performance improving predictably as the sample size $n$ per iteration increases.

- *Stable Error Component:* The first term in the parentheses, $1/\sqrt{c}$, represents an asymptotic, irreducible error floor that is independent of the initial model's quality. This error is governed by the model's minimum confidence, $c$, reflecting the best possible stability the system can achieve after sufficient iteration.

- *Transient Error Component:* The second term, $\sqrt{\kappa_0}/(1 + \sqrt{nc})^{(T-1)/2}$, captures the influence of the initial model's quality, as measured by the policy condition number $\kappa_0$. Crucially, this term decays exponentially with the number of iterations $T$. This finding not only theoretically validates the intuition that the iterative process progressively mitigates the adverse effects of a poor initialization, but also aligns with the experimental results in Yuan et al. (2024); Zhou et al. (2025); Xiong et al. (2025). Specifically, these results show that performance improves significantly in the early stages of iteration before plateauing, a pattern characteristic of exponential decay.

Furthermore, the bound's dependence on $\kappa_0$ and $n$ is shown to be tight for the single-step $(T = 1)$ case. Consider this special case, where the exponential decay term's denominator becomes one. For an ill-conditioned initial model where $\kappa_0$ is large, the $\sqrt{\kappa_0}$ term will dominate the $1/\sqrt{c}$ term. Ignoring logarithmic factors and constants, the upper bound on the failure probability simplifies to $\mathcal{O}\big((\kappa_0/n)^{1/2}\big)$. This rate precisely matches the fundamental lower bound of $\Omega\big((\kappa_0/n)^{1/2}\big)$ established in Theorem 1 for any single-step self-rewarding algorithm. This correspondence demonstrates that our multi-step analysis correctly captures the problem's intrinsic difficulty for the single-step scenario and confirms $\kappa_0$ as a central parameter governing the challenge of self-alignment. We do not, however, provide a matching lower bound for $T > 1$ to prove the tightness of the exponential decay rate itself. This remains an interesting open question that we leave for future work. In addition, the assumption of sampling $n$ pairs is practically realistic for large LMs. In our setting, $n$ denotes the total dataset size for one alignment round (typically $10^4$ to $10^6$ preference pairs), which is consistent with standard large-scale practice. The judge cost remains computationally modest, since evaluation reuses the same model weights and at most requires an additional batchable forward pass per sample. As a result, this self-labeling regime avoids the high annotation cost that would arise if all $n$ preference labels had to be obtained from humans, which would pose a severe scalability barrier.

**Remark 4. Why Self-Rewarding Works.** Theorem 3 provides a fundamental explanation for the success of iterative self-rewarding: it transforms a single-step learning problem that would almost certainly fail under poor initialization (large $\kappa_0$) into a robust two-stage process.

A large initial condition number $\kappa_0$ corresponds to an ill-conditioned model. Theorem 1 shows that for such models, a single update fails with probability $\Omega((\kappa_0/n)^{1/2})$. Thus, naive one-shot fine-tuning cannot overcome poor initialization. In contrast, iterative self-rewarding acts as an internal self-correction mechanism, thereby avoiding this difficulty.

*Stage I: Self-Correction and Stabilization.* The early iterations are not aimed at fitting an external true preference. Instead, the model aligns internally by rewarding its own high-confidence outputs. This self-reinforcing process induces a contraction mapping on the policy condition number $\kappa_t$, ensuring exponential convergence toward a stable fixed point. The proof in Appendix D formalizes this effect, showing that under mild assumptions, the sequence $\{\kappa_t\}$ satisfies

$$\kappa_T \le U + q^T(\kappa_0 - U), \quad q \asymp (1 + \sqrt{nc})^{-1} < 1,$$

where $U$ is the fixed-point condition number as $T \to \infty$, and $q^T(\kappa_0 - U)$ is the transient initialization component, which decays geometrically with each iteration. Hence, the adverse influence of poor initialization quickly disappears, guiding the model into a well-conditioned regime. This stage functions as an implicit regularization, prioritizing internal consistency before external generalization.

*Stage II: Efficient Statistical Learning.* Once stabilized, the effect of $\kappa_0$ becomes negligible. As the initialization-dependent term in Theorem 3 vanishes exponentially, the learning dynamics undergo a fundamental shift: they are no longer constrained by the starting point, but are governed by statistical efficiency. Performance improvements follow the standard $\mathcal{O}(1/\sqrt{n})$ rate, determined solely by the per-round sample size $n$. Ultimately, the process converges to a stable error floor that is independent of initialization, with the final accuracy dictated only by problem difficulty and statistical error.

Iterative self-rewarding succeeds by decomposing the learning problem into two phases: first stabilizing the model through self-alignment dynamics, and then leveraging this stable foundation for efficient statistical learning. In doing so, the framework enforces internal consistency of the model,

thereby creating the necessary conditions for effective learning and turning a seemingly hopeless single-step task into a standard, well-behaved learning process.

**Remark 5** (Proof Sketch for Theorem 3). This proof establishes finite-sample guarantees for iterative self-rewarding alignment, showing it overcomes the limitations of a poor initial model. The core technique is to recursively control the policy condition number $\kappa_t$. Controlling $\kappa_t$ prevents model collapse and ensures the influence of the initial model's quality, $\kappa_0$, diminishes over time.

*Step 1. Single-Step Optimality Gap.* First, we bound the single-step failure probability, the chance that an updated policy $\pi_{t+1}$ assigns low probability to its own optimal response. This probability is linked to a performance gap that is controlled by statistical error and the stability of the previous policy $\pi_t$. This yields guarantees connecting the performance to the policy condition number:

$$\mathbb{P}_{x \sim \mu}[\pi_t(y_{t+1}^*(x)|x) \leq 1 - \delta] \lesssim \frac{1}{\gamma\delta} \left( \sqrt{\kappa_{t-1}} \cdot \log(n|\Pi|\rho^{-1})n^{-1/2} + \beta \log(\kappa_{t-1}) \right).$$

This inequality forms the inductive link for the analysis.

*Step 2. Recursive Control of the Policy Condition Numbers.* To ensure stability, we derive a recursive inequality for $\kappa_t$. We start with its tail-integral representation:

$$\kappa_{t-1} = \mathbb{E}_{x \sim \mu} \left[ \frac{1}{\pi_{t-1}(y_{t-1}^*(x)|x)} \right] = \int_0^\infty \mathbb{P}_{x \sim \mu} \left( \frac{1}{\pi_{t-1}(y_{t-1}^*(x)|x)} \geq u \right) du.$$

The integrand is the failure probability bounded in Step 1. Substituting this bound into the integral yields the core recurrence relation:

$$\kappa_{t-1} \leq C_1 + C_2 \log(n|\Pi|\rho^{-1})n^{-1/2} \cdot \sqrt{\kappa_{t-1}} + C_3\beta \cdot \log(\kappa_{t-1}),$$

where $C_1, C_2, C_3$ are constants. The sub-linear dependency on $\sqrt{\kappa_t}$ is crucial, as it prevents explosive growth and ensures the process is stable.

*Step 3. Asymptotic Stability and Final Bound.* Finally, we analyze the recurrence for $\{\kappa_t\}$. It simplifies to a contraction mapping that converges exponentially fast to a fixed point $U$. This gives an explicit bound on the condition number at iteration $T - 1$:

$$\kappa_{T-1} \leq U + q^{T-1}(\kappa_0 - U),$$

where the contraction rate $q < 1$ scales as approximately $(1 + \sqrt{nc})^{-1}$. This decomposition reveals that $\kappa_{T-1}$ comprises a stable term $U$ and an exponentially decaying transient term that depends on the initial condition $\kappa_0$. Substituting this bound for $\kappa_{T-1}$ into the single-step guarantee for the final iteration $T$ yields the main result of Theorem 3. The final bound consists of a stable error term with a rate of $\mathcal{O}(1/\sqrt{n})$ and a transient error term, which depends on $\sqrt{\kappa_0}$ and decays exponentially with $T$. This proves that the iterative process robustly mitigates the influence of a poor initialization.

Our analysis, particularly the DPO square-loss instantiation and the recursive contraction of the policy condition number $\kappa_t$ (Lemmas 9 & 10), strictly relies on the self-reward function being $r_t(x, y) = \log \pi_t(y \mid x)$. These results do not directly extend to reward signals produced by external LLM-based judges or multi-step pipelines. Extending the theory to rubric-based rewards would require additional structural assumptions (e.g., margin conditions or Lipschitz relations w.r.t. likelihood ratios), which we leave for future work.

**Corollary 4** (Iterations to Suppress Initialization Effects). *Building on Theorem 3, if the number of iterations satisfies*

$$T \gtrsim \frac{\log(c\kappa_0)}{\log(1 + \sqrt{nc})},$$

*then the initialization-dependent part of the bound becomes negligible, and with probability at least $1 - \rho$,*

$$\mathbb{P}_{x \sim \mu}[\pi_T(y_T^*(x) \mid x) \leq 1 - \delta] \lesssim \frac{1}{\sqrt{n}} \cdot \frac{\log(n|\Pi|\rho^{-1})}{\gamma\delta\sqrt{c}}.$$

**Remark 6.** Corollary 4 distills and clarifies the core mechanism that underpins the success of iterative self-rewarding alignment. The self-correcting property provides a theoretical safeguard against model collapse, where a flawed initial model might otherwise amplify its own biases through iteration. The theory demonstrates that the iterative process effectively mitigates the influence of the

initial unaligned model, allowing the policy to bootstrap its way to improvement based on the signal generated during training.

The corollary also establishes that this self-correction process is both efficient and practical. The number of iterations required to suppress the initial model's influence to a negligible level, given by $T \gtrsim \log(c\kappa_0)/\log(1 + \sqrt{nc})$, depends only logarithmically on the initial condition number $\kappa_0$. This suggests that even an initially ill-conditioned model can be reliably aligned without requiring an excessive number of iterations, which is consistent with experimental findings (Yuan et al., 2024; Zhou et al., 2025; Xiong et al., 2025).

Furthermore, the denominator $\log(1 + \sqrt{nc})$ confirms the theoretical intuition that increasing the per-iteration sample size $n$ accelerates the decay of the initial model's influence. Consequently, this reduces the total number of iterations $T$ required for alignment. This relationship is consistent with the principle that a larger sample size enables the model to learn more effectively within each round, thereby diminishing the number of iterative steps needed. This finding provides clear guidance for allocating computational resources during the alignment process.

Ultimately, after a sufficient number of iterations, the process fundamentally overcomes the bottleneck inherent in single-step alignment. The resulting error bound, $\mathcal{O}\left(1/\sqrt{n}\right)$, becomes entirely independent of the initial condition number $\kappa_0$. This demonstrates that the iterative framework transforms the learning problem from one constrained by initial model quality, as captured by the $\Omega\left((\kappa_0/n)^{1/2}\right)$ lower bound in Theorem 1, into a standard statistical learning problem where performance is governed by sample size. The iterative process is therefore not merely a refinement but a powerful mechanism. It enables the model to overcome the constraining influence of its initialization, breaking the $\kappa_0$ barrier to achieve a level of performance unattainable through single-step methods.

## 6 APPLICATION TO PARAMETRIC MODELS: THE LINEAR SOFTMAX CASE

In this section, we instantiate our general framework for SRLMs within the parametric setting of linear softmax models.

**Definition 2** (Linear Softmax Model). Let $d \in \mathbb{N}$ be the feature dimension, and let $\varphi : \mathcal{X} \times \mathcal{Y} \to \mathbb{R}^d$ be a feature map such that $\|\varphi(x, y)\|_2 \leq 1$. Given a radius $B \geq 1$, the linear softmax model class $\Pi_{\varphi, B}$ is defined as:
$$\Pi_{\varphi, B} := \left\{ \pi_\theta : \theta \in \mathbb{R}^d, \|\theta\|_2 \leq B \right\},$$
where the policy $\pi_\theta$ is given by: $\pi_\theta(y \mid x) \propto \exp\left(\langle \varphi(x, y), \theta \rangle\right)$. This class is parameterized by a vector $\theta$ within a $d$-dimensional ball of radius $B$.

By adapting our proof framework with tools from statistical learning theory, we establish a performance guarantee for the linear softmax model class, analogous to Theorem 3, thereby demonstrating that the self-rewarding process remains both stable and effective.

**Theorem 5** (Performance Guarantee for Linear Softmax Models). *Under the same conditions as Theorem 3, but applied to the linear softmax class $\Pi_{\varphi, B}$, after $T$ rounds of self-rewarding updates, the final policy $\pi_{\theta_T}$ ensures that with probability at least $1 - \rho$:*

$$\mathbb{P}_{x \sim \mu}\left[\pi_{\theta_T}(y_T^*(x) \mid x) \leq 1 - \delta\right] \lesssim \frac{1}{\sqrt{n}} \cdot \frac{1}{\gamma\delta} \left(d \log \frac{nB}{d\rho}\right)^{3/2} \left(\frac{1}{\sqrt{c}} + \frac{\sqrt{\kappa_0}}{(1 + \sqrt{nc})^{(T-1)/2}}\right). \quad (8)$$

*Furthermore, if the number of iterations satisfies $T \gtrsim \log(c\kappa_0)/\log(1 + \sqrt{nc})$, then we obtain:*

$$\mathbb{P}_{x \sim \mu}[\pi_{\theta_T}(y_T^*(x) \mid x) \leq 1 - \delta] \lesssim \frac{1}{\sqrt{n}} \cdot \frac{1}{\gamma\delta} \left(d \log \frac{nB}{d\rho}\right)^{3/2}. \quad (9)$$

**Remark 7.** Theorem 5 above instantiates our general framework in Theorem 3 for a continuous parametric model class. Notably, the complexity term $\log|\Pi|$ from the finite class setting is replaced by a *geometric* entropy term arising from covering the parameter space. In particular, the bound's complexity factor scales as $(d \log(nB/(d\rho)))^{3/2}$, reflecting the metric entropy of a $d$-dimensional parameter ball instead of the simple count $\log|\Pi|$. This signifies a shift from combinatorial complexity to the complexity of a continuous function class, introducing an explicit polynomial dependence on the ambient feature dimension $d$. Moreover, while the iterative procedure eventually cancels out

the effect of initialization, we note that the iteration threshold $T \gtrsim \log(c\kappa_0)/\log(1 + \sqrt{nc})$ still carries an implicit dependence on $d$ through the initial condition $\kappa_0$. In worst-case high-dimensional scenarios (e.g. an uninformative initial policy in a large feature space), $\kappa_0$ can grow with $d$, meaning that more iterations may be required for the benefits of self-alignment to fully manifest.

Despite the above guarantee, the statistical rate in Theorem 5 still exhibits a polynomial dependence on the ambient dimension $d$, which can be overly conservative in modern high-dimensional settings. To mitigate this apparent curse of dimensionality, we refine the analysis by introducing a data-dependent complexity measure grounded in the *spectral structure* of the feature space. In particular, the ambient dimension $d$ is replaced with an *effective dimension* $d_{\text{eff}}(\lambda)$ that reflects the eigenvalue decay of the feature covariance $\Sigma_\varphi$. By aligning the complexity term with the concentration of variance within a lower-dimensional subspace, we obtain stronger bounds that adapt to the true intrinsic dimensionality of the data, as formalized in Corollary 6.

**Corollary 6** (Linear-Softmax under Exponential Spectral Decay). *Assume the feature covariance* $\Sigma_\varphi := \mathbb{E}[\varphi(x,y)\varphi(x,y)^\top]$ *has eigenvalues* $\{\lambda_i\}_{i=1}^d$ *satisfying exponential decay* $\lambda_i \leq C\,e^{-\alpha i}$ *for some* $C, \alpha > 0$. *Using the effective-dimension argument with ridge parameter* $\lambda \asymp c_0/n$, *the iteration threshold that suppresses initialization is* $T \gtrsim \log(c\kappa_0)/\log(1 + \sqrt{nc})$. *Moreover, with probability at least* $1 - \rho$,

$$\mathbb{P}_{x\sim\mu}\big[\pi_{\theta_T}\big(y_T^*(x) \mid x\big) \leq 1 - \delta\big] \;\lesssim\; \frac{1}{\sqrt{n}} \cdot \frac{1}{\gamma\,\delta} \left(\log n \cdot \log \frac{nB}{\rho}\right)^{3/2}.$$

*In particular, the ambient-dimensional factor* $d \cdot \log(nB/d\rho)$ *is replaced by a doubly logarithmic dependence via* $\log n \cdot \log(nB/\rho)$.

**Remark 8** (Effective Dimension and Spectral Decay). Corollary 6 introduces a refined assumption that the eigenvalues of the feature covariance $\Sigma_\varphi$ decay exponentially, which is consistent with the feature geometry observed in modern large-scale language models Dong et al. (2021). In practice, pretrained model representations often exhibit a rapidly decaying spectrum, with most of the variance concentrated in the top principal components Ethayarajh (2019); Saglam et al. (2025). This structural property implies that the feature space is effectively low-dimensional. Under an exponential decay profile $\lambda_i \lesssim Ce^{-\alpha i}$, the ridge effective dimension $d_{\text{eff}}(\lambda)$ remains substantially smaller than the ambient dimension $d$ for moderate regularization $\lambda$. For instance, choosing $\lambda \asymp 1/n$ yields $d_{\text{eff}}(\lambda) = \mathcal{O}(\log n)$ rather than $\mathcal{O}(d)$. This shows that although the model may have $d$ parameters, the complexity of its learned representation is governed by a much smaller effective number of directions. This observation provides a natural explanation for the apparent paradox that overparameterized models can still generalize effectively. Intuitively, most parameters correspond to directions in the feature space with negligible variance and thus do not significantly affect generalization error, allowing an overparameterized model to behave as if it were much smaller. A more detailed discussion is provided in Section H of the Appendix. The key insight is that generalization is governed by the effective dimension $d_{\text{eff}}(\lambda)$, not the ambient dimension $d$. Scaling laws emerge because larger models have the capacity to learn feature representations with faster spectral decay (i.e., a smaller $d_{\text{eff}}$). This aligns with benign overfitting, where models generalize well despite over-parameterization by using a low-dimensional signal structure. While our guarantees hold for linear softmax models, extending this analysis to full Transformers is a significant open challenge and a valuable direction for future work.

## 7    CONCLUSION

In summary, this paper provides the first rigorous theoretical foundation for Self-Rewarding Language Models (SRLMs), bridging the gap between their striking empirical success and the absence of formal guarantees. By establishing sharp lower bounds for single-step updates and finite-sample convergence rates for the iterative paradigm, we explain why SRLMs can reliably overcome poor initialization and achieve stable alignment. Our results show that dependence on the initial model decays exponentially with iterations, and that the long-run behavior is governed by provably controlled error terms. These insights not only clarify the mechanisms underlying self-rewarding but also offer concrete guidance for designing more robust, scalable, and autonomous alignment procedures in future large-scale language models

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

## APPENDIX

## OVERVIEW

In this supplementary material, we provide additional details and complete proofs to support the theoretical developments presented in the main paper. The appendix is organized as follows:

- **Appendix C. The Use of Large Language Models.** We note that large language models were used to assist in the writing and refinement of this manuscript.

- **Appendix D. Proof of Theorem 3: Finite-Sample Guarantees for Iterative Alignment.** We provide the complete mathematical proof for the finite-sample guarantee of the iterative self-rewarding process. This section details how the influence of the initial model decays exponentially by recursively controlling the policy condition number.

- **Appendix E. Proof of Theorem 1: Lower Bound on Single-Step Failure Rate.** We establish the proof for the failure rate lower bound of a single-step update. This is achieved by constructing a hard problem instance to reveal the inherent limitations of a single update step, especially when the initial model is of poor quality.

- **Appendix F. Proof of Proposition 2: From Low-Confidence Models to Greedy Decoding Failure.** We prove that a greedy decoding strategy is guaranteed to fail for models that have low confidence in their own optimal output, thereby formally connecting learning failure with inference failure.

- **Appendix G. Proof of Theorem 5: Performance Guarantees for Linear Softmax Models.** We extend the general theoretical framework to the continuous, parametric case of linear softmax models. This section details how covering numbers are used to derive performance guarantees for this specific model class.

- **Appendix H. Proof of Corollary 6: Data-Dependent Bounds via Effective Dimension.** We refine the performance bounds for parametric models by introducing the concept of effective dimension, which is based on the spectral structure of the feature covariance. This allows the bound to depend on the intrinsic data complexity rather than the ambient dimension.

This appendix provides the full mathematical foundations of our results and demonstrates their applicability to modern high-capacity models used in practice.

## A  ADDITIONAL RELATED WORK

In this section, we discuss additional lines of work related to classical pseudo-labelling and off-policy batch learning, with the goal of further clarifying the conceptual position of our framework within the broader literature.

**Classical Pseudo-Labelling in Semi-Supervised Learning.** Our work shares connections with classical pseudo-labelling methods in semi-supervised learning (SSL), where a model trained on labeled data predicts discrete pseudo-labels for unlabeled examples and is subsequently retrained on the combined dataset (Grandvalet & Bengio, 2004; Lee et al., 2013). Modern approaches integrate pseudo-labelling with entropy minimization, consistency regularization, and strong augmentation strategies, yielding methods such as MixMatch, ReMixMatch, and FixMatch (Berthelot et al., 2019b;a; Sohn et al., 2020). Beyond these algorithmic developments, recent work Aminian et al. (2024) has proposed divergence-based and information-theoretic formulations of self-training. For instance, some studies design empirical risk functions and regularizers based on $f$-divergences and $\alpha$-Rényi divergences to make pseudo-labelling more robust to noisy pseudo-labels, thereby placing classical self-training on a divergence-regularization footing. Other work Aminian et al. (2022a) provides an information-theoretic framework for self-training under covariate shift, recovering pseudo-labelling and entropy minimization as special cases.

While conceptually related in leveraging model-generated supervision, our SRLM framework diverges significantly from classical SSL. SSL approaches typically assume a semi-supervised setting with a static pool of unlabeled inputs and a fixed pseudo-labelling mechanism that outputs (possibly softened) class labels. In contrast, SRLMs operate in an off-policy sequential decision-making setting. Here, the data distribution is inherently non-stationary due to the evolving policy $\pi_t$; the supervisory signals are continuous self-rewards derived from model log-likelihoods; and the policy and reward mechanism co-evolve, forming a coupled dynamical system rather than a unidirectional teacher–student update.

**Pseudo-Reward and Off-Policy Batch Learning from Logged Data.** Learning from logged interaction data is extensively studied in contextual bandits and offline reinforcement learning. Counterfactual Risk Minimization (CRM) formulates batch learning from logged feedback via propensity-weighted empirical risk minimization (Swaminathan & Joachims, 2015; Joachims et al., 2018). In addition, Aminian et al. (2022b) consider semi-supervised batch learning from logged data, where only a subset of samples contains feedback, and provide upper bounds that motivate divergence-based regularization terms between the target and logging policies. More recently, other work has proposed log-sum-exponential (LSE) estimators for off-policy evaluation and learning from logged bandit feedback, aiming to improve robustness and reinforcing the regularization-centric perspective (Behnamnia et al., 2024; 2025).

Our SRLM framework shares conceptual alignment with this line of work, as we also optimize a policy under explicit regularization with respect to a reference (or logging) policy. However, a critical distinction arises from the data-generation process. The aforementioned logged-data methods assume a fixed logging policy and operate on a static historical dataset. In contrast, SRLMs iteratively generate new trajectories, meaning the effective logging policy co-evolves with $\pi_t$, producing a non-stationary and endogenous data distribution. Consequently, since our analysis focuses on bootstrapping from weak initialization and stabilizing the evolving interaction between policy and reward, it lies outside the static batch-learning assumptions underlying CRM, semi-supervised logged-data methods, and LSE-based estimators.

## B  LIMITATION

We acknowledge the significant challenge of extending the rigorous guarantees established in this work to full, modern Transformer architectures. The intricate components of these models, such as multi-head attention and deep residual connections, introduce complex non-linear dynamics and high-dimensional, non-convex optimization landscapes that are not captured by our current linear setting. Establishing a complete theoretical framework, such as a benign overfitting theorem or a proof of stable convergence, for such systems remains a major open problem in the theoretical deep learning community. At present, rigorous theories are largely confined to more tractable settings, primarily linear models (as adopted in our analysis) or two-layer neural networks. Therefore, bridging

this substantial gap by extending the analytical framework to full-scale Transformers constitutes a highly valuable and essential direction for future research.

## C  THE USE OF LARGE LANGUAGE MODELS

In this work, we make exclusive use of large language models (LLMs) to refine the writing, improve stylistic quality, and polish the presentation of our manuscript.

## D  PROOF OF THEOREM 3: FINITE-SAMPLE GUARANTEE FOR ITERATIVE SELF-REWARDING ALIGNMENT

This section provides the complete and rigorous proof for Theorem 3, our main result establishing finite-sample guarantees for the iterative self-rewarding alignment process. The proof is structured to first analyze the performance bounds of a single update step. We then construct a recursive argument that tracks the evolution of the policy condition number, $\kappa_t$, across successive iterations. By showing that this parameter is controlled and converges, we formally demonstrate that the influence of the initial model's quality decays exponentially, thus ensuring the stability and statistical efficiency of the iterative alignment framework. Our analysis draws inspiration from the theoretical self-improvement framework introduced in Huang et al. (2025), but extends it to the more complex iterative self-rewarding paradigm.

*Proof.* We begin by formalizing the notation and assumptions underlying the proof. We then establish a foundational lemma that connects the probability of a model being suboptimal to its performance gap relative to an optimal comparator.

**Notation and Assumptions.**  For each $x \in \mathcal{X}$, assume there exists a maximizer

$$y_0^*(x) \in \arg\max_{y \in \mathcal{Y}} \pi_0(y \mid x),$$

and that $\pi_0(y \mid x) > 0$ for all $y \in \mathcal{Y}$. We define the deterministic comparator model as

$$\pi_0^*(y \mid x) := \mathbb{I}\{y = y_0^*(x)\}.$$

The performance of any model $\pi$ is measured by the functional

$$J_0(\pi) := \mathbb{E}_{x \sim \mu}\, \mathbb{E}_{y \sim \pi(\cdot \mid x)}\big[\log \pi_0(y \mid x)\big].$$

In addition, we recall the margin parameter $\gamma > 0$, which quantifies the minimum log-probability gap between the optimal action and any suboptimal action:

$$\gamma := \inf_{x \in \mathcal{X}, t \in [0,T]} \log\left(\frac{\pi_t(y_t^*(x) \mid x)}{\max_{y' \neq y_t^*(x)} \pi_t(y' \mid x)}\right).$$

The assumption of uniqueness ensures $\gamma > 0$.

### D.1  FROM PERFORMANCE GAP TO SUBOPTIMALITY PROBABILITY.

The central step is to upper bound the probability that the model $\pi_1$ assigns insufficient probability mass to the unique maximizer $y_0^*(x)$. The following lemma formalizes this connection.

**Lemma 7.** *For any model $\pi_1$ and any $\delta \in (0, 1)$, the probability of assigning insufficient mass to the optimal action satisfies*

$$\mathbb{P}_{x \sim \mu}\big[\pi_1(y_0^*(x) \mid x) \leq 1 - \delta\big] \;\leq\; \frac{J_0(\pi_1^*) - J_0(\pi_1)}{\gamma\,\delta}. \tag{10}$$

*Proof of Lemma 7.* The proof proceeds in two steps.

*Step 1: Bounding the probability by an expectation.* For any $q \in [0, 1]$ and $\delta \in (0, 1)$, the indicator inequality

$$\mathbb{I}\{q \leq 1 - \delta\} \ \leq \ \frac{1 - q}{\delta}$$

holds. Applying this with $q = \pi_1(y_0^*(x) \mid x)$ and averaging over $x \sim \mu$ gives

$$\mathbb{P}_{x \sim \mu}\big[\pi_1(y_0^*(x) \mid x) \leq 1 - \delta\big] = \mathbb{E}_{x \sim \mu}\Big[\mathbb{I}\{\pi_1(y_0^*(x) \mid x) \leq 1 - \delta\}\Big]$$

$$\leq \frac{1}{\delta}\,\mathbb{E}_{x \sim \mu}\Big[1 - \pi_1(y_0^*(x) \mid x)\Big]. \tag{11}$$

*Step 2: Relating the expectation to the performance gap.* By the definition of $\gamma$, for any $x \in \mathcal{X}$ and $y' \in \mathcal{Y}$,

$$\log\frac{\pi_0(y_0^*(x) \mid x)}{\pi_0(y' \mid x)} \ \geq \ \gamma\,\mathbb{I}\{y' \neq y_0^*(x)\}.$$

Taking expectations with $y' \sim \pi_1(\cdot \mid x)$ and $x \sim \mu$, and noting that $y = y_0^*(x)$ almost surely under $\pi_1^*$, we obtain

$$J_0(\pi_1^*) - J_0(\pi_1) = \mathbb{E}_{x \sim \mu}\,\mathbb{E}_{y' \sim \pi_1(\cdot \mid x)}\left[\log\frac{\pi_0(y_0^*(x) \mid x)}{\pi_0(y' \mid x)}\right]$$

$$\geq \gamma\,\mathbb{E}_{x \sim \mu}\,\mathbb{P}_{y' \sim \pi_1(\cdot \mid x)}\big[y' \neq y_0^*(x) \mid x\big]$$

$$= \gamma\,\mathbb{E}_{x \sim \mu}\Big[1 - \pi_1(y_0^*(x) \mid x)\Big].$$

Thus,

$$\mathbb{E}_{x \sim \mu}\Big[1 - \pi_1(y_0^*(x) \mid x)\Big] \ \leq \ \frac{J_0(\pi_1^*) - J_0(\pi_1)}{\gamma}. \tag{12}$$

*Final step.* Substituting Eq. 12 into Eq. 11 completes the proof of the lemma, establishing inequality:

$$\mathbb{P}_{x \sim \mu}\big[\pi_1(y_0^*(x) \mid x) \leq 1 - \delta\big] \ \leq \ \frac{J_0(\pi_1^*) - J_0(\pi_1)}{\gamma\,\delta}.$$

$\square$

### D.2 A REWARD IDENTITY AND THE PERFORMANCE GAP DECOMPOSITION

To derive a tractable bound for the performance gap $J_0(\pi_1^*) - J_0(\pi_1)$, we begin by introducing key notation and establishing a fundamental reward identity. This identity enables us to decompose the performance gap into components that can be bounded using statistical learning arguments.

**Shorthand Notation and a Baseline-Cancellation Identity.** For any policy $\pi : \mathcal{X} \to \Delta(\mathcal{Y})$, we introduce the following notation for expectations:

$$\mathbb{E}_\pi[\cdot] := \mathbb{E}_{x \sim \mu}\,\mathbb{E}_{y \sim \pi(\cdot \mid x)}[\cdot], \qquad \mathbb{E}_{\pi,\pi'}[\cdot] := \mathbb{E}_{x \sim \mu}\,\mathbb{E}_{y \sim \pi(\cdot \mid x),\,y' \sim \pi'(\cdot \mid x)}[\cdot].$$

Conditioned on $x$, the random draws $y \sim \pi(\cdot \mid x)$ and $y' \sim \pi'(\cdot \mid x)$ are independent.

For any measurable function $g : \mathcal{X} \times \mathcal{Y} \to \mathbb{R}$, we define the *pairwise difference operator*:

$$\Delta^g(x, y, y') := g(x, y) - g(x, y').$$

This leads to the following fundamental *baseline-cancellation identity*:

$$\mathbb{E}_\pi[g] - \mathbb{E}_{\pi'}[g] \ = \ \mathbb{E}_{\pi,\pi'}\big[\Delta^g\big]. \tag{13}$$

This identity is repeatedly used in our analysis to simplify expressions and eliminate baseline terms.

**A Reward Identity via Entropy Regularization.** We now derive a pointwise representation for the reward function $r_0^*(x, y) := \log \pi_0(y \mid x)$. Consider the entropy-regularized optimization problem for a fixed $x$:

$$\pi_{1,\beta}^*(\cdot \mid x) \in \arg \max_{\pi(\cdot \mid x) \in \Delta(\mathcal{Y})} \left\{ \mathbb{E}_{y \sim \pi(\cdot \mid x)}[\log \pi_0(y \mid x)] - \beta \, D_{\mathrm{KL}}(\pi(\cdot \mid x) \,\|\, \pi_0(\cdot \mid x)) \right\}.$$

The unique optimizer is known to have the Gibbs form:

$$\pi_{1,\beta}^*(y \mid x) = \frac{\pi_0(y \mid x)^{1+1/\beta}}{Z_{\mathrm{norm}}(x)}, \qquad Z_{\mathrm{norm}}(x) := \sum_{y' \in \mathcal{Y}} \pi_0(y' \mid x)^{1+1/\beta}. \tag{14}$$

Dividing Eq. 14 by $\pi_0(y \mid x)$, taking logarithms, and multiplying by $\beta$ yields

$$\beta \log \frac{\pi_{1,\beta}^*(y \mid x)}{\pi_0(y \mid x)} = \log \pi_0(y \mid x) - \beta \log Z_{\mathrm{norm}}(x).$$

Rearranging provides the identity

$$r_0^*(x, y) = \log \pi_0(y \mid x) = \beta \log \frac{\pi_{1,\beta}^*(y \mid x)}{\pi_0(y \mid x)} + Z_0(x), \qquad Z_0(x) := \beta \log Z_{\mathrm{norm}}(x). \tag{15}$$

The term $Z_0(x)$ depends only on $x$, and therefore cancels out in any pairwise difference. Specifically,

$$\begin{aligned} \Delta^{r_0^*}(x, y, y') &:= r_0^*(x, y) - r_0^*(x, y') \\ &= \beta \log \frac{\pi_{1,\beta}^*(y \mid x)}{\pi_0(y \mid x)} - \beta \log \frac{\pi_{1,\beta}^*(y' \mid x)}{\pi_0(y' \mid x)} \\ &=: \Delta^{\tilde{r}_0}(x, y, y'), \qquad \tilde{r}_0(x, y) := \beta \log \frac{\pi_{1,\beta}^*(y \mid x)}{\pi_0(y \mid x)}. \end{aligned} \tag{16}$$

Thus, the pairwise differences of the original reward $r_0^*$ are equivalent to those of the transformed reward $\tilde{r}_0$, which resembles the DPO reward $\hat{r}_0(x, y) := \beta \log \frac{\pi_1(y \mid x)}{\pi_0(y \mid x)}$ but with $\pi_1$ replaced by $\pi_{1,\beta}^*$.

**Decomposition of the Performance Gap.** With the reward identity Eq. 15 and the baseline-cancellation identity Eq. 13, we now decompose the performance gap.

Recall that $\pi_1$ is the optimizer of the DPO objective:

$$\pi_1 \in \arg \max_{\pi: \mathcal{X} \to \Delta(\mathcal{Y})} \left\{ \mathbb{E}_\pi[\hat{r}_0] - \beta D_{\mathrm{KL}}(\pi \| \pi_0) \right\}.$$

This optimality condition implies that for any comparator $\pi_1^*$,

$$\mathbb{E}_{\pi_1^*}[\hat{r}_0] - \beta D_{\mathrm{KL}}(\pi_1^* \| \pi_0) \leq \mathbb{E}_{\pi_1}[\hat{r}_0] - \beta D_{\mathrm{KL}}(\pi_1 \| \pi_0).$$

We begin by inserting and subtracting $\hat{r}_0$ terms into the definition of the performance gap:

$$\begin{aligned} J_0(\pi_1^*) - J_0(\pi_1) &= \mathbb{E}_{\pi_1^*}[r_0^*] - \mathbb{E}_{\pi_1}[r_0^*] \\ &= \left( \mathbb{E}_{\pi_1^*}[\hat{r}_0] - \mathbb{E}_{\pi_1}[\hat{r}_0] \right) + \mathbb{E}_{\pi_1^*}[r_0^* - \hat{r}_0] + \mathbb{E}_{\pi_1}[\hat{r}_0 - r_0^*]. \end{aligned}$$

Applying the DPO optimality condition to the first term yields

$$J_0(\pi_1^*) - J_0(\pi_1) \leq \mathbb{E}_{\pi_1^*}[r_0^* - \hat{r}_0] + \mathbb{E}_{\pi_1}[\hat{r}_0 - r_0^*] + \beta D_{\mathrm{KL}}(\pi_1^* \| \pi_0) - \beta D_{\mathrm{KL}}(\pi_1 \| \pi_0). \tag{17}$$

We next apply the baseline-cancellation identity Eq. 13 to refine the two expectation terms. For the first term:

$$\begin{aligned} \mathbb{E}_{\pi_1^*}[r_0^* - \hat{r}_0] &= \left( \mathbb{E}_{\pi_1^*}[r_0^*] - \mathbb{E}_{\pi_0}[r_0^*] \right) - \left( \mathbb{E}_{\pi_1^*}[\hat{r}_0] - \mathbb{E}_{\pi_0}[\hat{r}_0] \right) + \mathbb{E}_{\pi_0}[r_0^* - \hat{r}_0] \\ &= \mathbb{E}_{\pi_1^*, \pi_0}\big[\Delta^{r_0^*}\big] - \mathbb{E}_{\pi_1^*, \pi_0}\big[\Delta^{\hat{r}_0}\big] + \mathbb{E}_{\pi_0}[r_0^* - \hat{r}_0] \\ &= \mathbb{E}_{\pi_1^*, \pi_0}\big[\Delta^{r_0^*} - \Delta^{\hat{r}_0}\big] + \mathbb{E}_{\pi_0}[r_0^* - \hat{r}_0], \end{aligned} \tag{18}$$

Similarly, for the second term:

$$\mathbb{E}_{\pi_1}[\hat{r}_0 - r_0^*] = \left(\mathbb{E}_{\pi_1}[\hat{r}_0] - \mathbb{E}_{\pi_0}[\hat{r}_0]\right) - \left(\mathbb{E}_{\pi_1}[r_0^*] - \mathbb{E}_{\pi_0}[r_0^*]\right) + \mathbb{E}_{\pi_0}[\hat{r}_0 - r_0^*]$$

$$= \mathbb{E}_{\pi_1,\pi_0}\big[\Delta^{\hat{r}_0}\big] - \mathbb{E}_{\pi_1,\pi_0}\big[\Delta^{r_0^*}\big] + \mathbb{E}_{\pi_0}[\hat{r}_0 - r_0^*]$$

$$= \mathbb{E}_{\pi_1,\pi_0}\big[\Delta^{\hat{r}_0} - \Delta^{r_0^*}\big] + \mathbb{E}_{\pi_0}[\hat{r}_0 - r_0^*]. \tag{19}$$

Substituting Eq. 18 and Eq. 19 into Eq. 17 and observing that the baseline terms cancel exactly, we obtain the final decomposition:

$$J_0(\pi_1^*) - J_0(\pi_1) \;\leq\; \mathbb{E}_{\pi_1^*,\pi_0}\big[\Delta^{r_0^*} - \Delta^{\hat{r}_0}\big] + \mathbb{E}_{\pi_1,\pi_0}\big[\Delta^{\hat{r}_0} - \Delta^{r_0^*}\big] + \beta D_{\mathrm{KL}}(\pi_1^*\|\pi_0) - \beta D_{\mathrm{KL}}(\pi_1\|\pi_0). \tag{20}$$

This decomposition forms the foundation for subsequent statistical analysis of the performance gap.

### D.3 BOUNDING THE DIFFERENCE OF REWARD DELTAS VIA REGION DECOMPOSITION

In this section we control the two principal terms in the performance bound Eq. 20, namely $\mathbb{E}_{\pi_1^*,\pi_0}\big[\Delta^{r_0^*} - \Delta^{\hat{r}_0}\big]$ and $\mathbb{E}_{\pi_1,\pi_0}\big[\Delta^{\hat{r}_0} - \Delta^{r_0^*}\big]$. Our strategy is to decompose the expectation of the absolute difference $|\Delta^{r_0^*} - \Delta^{\hat{r}_0}|$ into contributions from a *good region*, where the pairwise reward differences are uniformly bounded, and a complementary *bad region*, which captures rare tail events.

**Preliminaries.** We recall the coupled–expectation shorthand

$$\mathbb{E}_{\pi,\pi'}[\cdot] \;:=\; \mathbb{E}_{x\sim\mu}\,\mathbb{E}_{y\sim\pi(\cdot|x),\,y'\sim\pi'(\cdot|x)}[\cdot],$$

and the pairwise differences of interest:

$$\Delta^{r_0^*}(x,y,y') := r_0^*(x,y) - r_0^*(x,y') \;=\; \beta\log\frac{\pi_{1,\beta}^*(y\mid x)}{\pi_0(y\mid x)} - \beta\log\frac{\pi_{1,\beta}^*(y'\mid x)}{\pi_0(y'\mid x)}, \tag{21}$$

$$\Delta^{\hat{r}_0}(x,y,y') := \hat{r}_0(x,y) - \hat{r}_0(x,y') \;=\; \beta\log\frac{\pi_1(y\mid x)}{\pi_0(y\mid x)} - \beta\log\frac{\pi_1(y'\mid x)}{\pi_0(y'\mid x)}. \tag{22}$$

We also use the standard concentrability coefficient of a policy $\pi$ with respect to the baseline $\pi_0$:

$$C_\pi := \mathbb{E}_x\,\mathbb{E}_{y\sim\pi(\cdot|x)}\left[\frac{\pi(y\mid x)}{\pi_0(y\mid x)}\right] = \mathbb{E}_x\,\mathbb{E}_{y\sim\pi_0(\cdot|x)}\left[\left(\frac{\pi(y\mid x)}{\pi_0(y\mid x)}\right)^2\right].$$

**Good/Bad region decomposition.** Fix a threshold $\eta > 0$, and define

$$G \;:=\; \{\,|\Delta^{r_0^*}| \leq \eta,\ |\Delta^{\hat{r}_0}| \leq \eta\,\}, \qquad B \;:=\; G^{\complement} \;=\; \{\,|\Delta^{r_0^*}| > \eta\ \text{or}\ |\Delta^{\hat{r}_0}| > \eta\,\}.$$

Let $D := \Delta^{r_0^*} - \Delta^{\hat{r}_0}$. Then

$$\mathbb{E}_{\pi_1^*,\pi_0}\big[|D|\big] \;=\; \mathbb{E}_{\pi_1^*,\pi_0}\big[|D|\,\mathbb{I}_G\big] \;+\; \mathbb{E}_{\pi_1^*,\pi_0}\big[|D|\,\mathbb{I}_B\big]. \tag{23}$$

We now bound the two contributions on the right-hand side.

**Lemma 8** (Change of measure with concentrability)**.** *For any nonnegative measurable function $f$, the coupled expectation under $(\pi_1^*, \pi_0)$ satisfies*

$$\mathbb{E}_{\pi_1^*,\pi_0}[f] \;\leq\; C_{\pi_1^*}^{1/2}\left(\mathbb{E}_{\pi_0,\pi_0}[f^2]\right)^{1/2}, \qquad C_\pi \;:=\; \mathbb{E}_{x\sim\mu}\,\mathbb{E}_{y\sim\pi_0(\cdot|x)}\left[\left(\frac{\pi(y\mid x)}{\pi_0(y\mid x)}\right)^2\right]. \tag{24}$$

*Proof.* We expand the coupled expectation explicitly:

$$\mathbb{E}_{\pi_1^*,\pi_0}[f] = \mathbb{E}_{x\sim\mu}\,\mathbb{E}_{\substack{y\sim\pi_1^*(\cdot|x)\\ y'\sim\pi_0(\cdot|x)}}\big[f(x,y,y')\big].$$

Reweighting the distribution of $y$ from $\pi_1^*$ to $\pi_0$ introduces the ratio

$$w(x,y) \;:=\; \frac{\pi_1^*(y\mid x)}{\pi_0(y\mid x)}.$$

Hence

$$\mathbb{E}_{\pi_1^*,\pi_0}[f] = \mathbb{E}_{x\sim\mu}\,\mathbb{E}_{\substack{y\sim\pi_0(\cdot|x)\\y'\sim\pi_0(\cdot|x)}}\big[w(x,y)\,f(x,y,y')\big] = \mathbb{E}_{\pi_0,\pi_0}[w\,f].$$

Applying the Cauchy–Schwarz inequality on the joint law $(x,y,y') \sim \mu \otimes \pi_0 \otimes \pi_0$ gives

$$\mathbb{E}_{\pi_0,\pi_0}[w\,f] \;\leq\; \Big(\mathbb{E}_{\pi_0,\pi_0}[w^2]\Big)^{1/2}\Big(\mathbb{E}_{\pi_0,\pi_0}[f^2]\Big)^{1/2}.$$

Finally, observe that

$$\mathbb{E}_{\pi_0,\pi_0}[w^2] = \mathbb{E}_{x\sim\mu}\,\mathbb{E}_{y\sim\pi_0(\cdot|x)}\left[\left(\frac{\pi_1^*(y\mid x)}{\pi_0(y\mid x)}\right)^2\right] = C_{\pi_1^*}.$$

Substituting this expression completes the proof of Eq. 24. $\qquad\square$

**Bounding the bad–region term.** We now derive a usable bound on the bad–region contribution. Starting from Lemma 8 and choosing the test function $f = |D|\,\mathbb{I}_B$, we immediately obtain

$$\mathbb{E}_{\pi_1^*,\pi_0}\big[|D|\,\mathbb{I}_B\big] \;\leq\; C_{\pi_1^*}^{1/2}\left(\mathbb{E}_{\pi_0,\pi_0}[D^2\,\mathbb{I}_B]\right)^{1/2}. \tag{25}$$

Thus the task reduces to bounding the second moment of $D$ restricted to the bad region.

To proceed, we invoke Hölder's inequality (equivalently, a second application of Cauchy–Schwarz), which separates the event probability from the higher–order moment:

$$\mathbb{E}_{\pi_0,\pi_0}[D^2\,\mathbb{I}_B] \;\leq\; \big(\mathbb{P}_{\pi_0,\pi_0}(B)\big)^{1/2}\big(\mathbb{E}_{\pi_0,\pi_0}[D^4]\big)^{1/2}. \tag{26}$$

Here the first factor controls the likelihood of landing in the bad region, while the second factor controls the size of the reward differences when such events occur.

The probability of the bad region itself can be bounded via a simple union bound. Since $B$ is the event that either $|\Delta^{r_0^*}|$ or $|\Delta^{\hat{r}_0}|$ exceeds the threshold $\eta$, we have

$$\mathbb{P}_{\pi_0,\pi_0}(B) \;\leq\; \mathbb{P}_{\pi_0,\pi_0}\big(|\Delta^{r_0^*}| > \eta\big) \;+\; \mathbb{P}_{\pi_0,\pi_0}\big(|\Delta^{\hat{r}_0}| > \eta\big). \tag{27}$$

It remains to bound the fourth moment of $D = \Delta^{r_0^*} - \Delta^{\hat{r}_0}$. By the elementary inequality $(a-b)^4 \leq 8(a^4+b^4)$, we obtain

$$\mathbb{E}_{\pi_0,\pi_0}[D^4] \;\leq\; 8\left(\mathbb{E}_{\pi_0,\pi_0}[|\Delta^{r_0^*}|^4] + \mathbb{E}_{\pi_0,\pi_0}[|\Delta^{\hat{r}_0}|^4]\right). \tag{28}$$

Finally, by combining Eq. 25–Eq. 28 and invoking the standard fourth-moment bound provided in Lemma J.2 of Huang et al. (2025), we obtain the following compact estimate.

$$\mathbb{E}_{\pi_1^*,\pi_0}\big[|D|\,\mathbb{I}_B\big] \;\lesssim\; C_{\pi_1^*}^{1/2}\,\Gamma\,\Upsilon^{1/4}, \tag{29}$$

where

$$\Gamma := \log\big(C_{\pi_0/\pi_1;\beta}\big) + \log\big(C_{\pi_0/\pi_{1,\beta}^*;\beta}\big), \qquad \Upsilon := \mathbb{P}(|\Delta^{r_0^*}| > \eta) + \mathbb{P}(|\Delta^{\hat{r}_0}| > \eta).$$

In words, the bad–region contribution is controlled by three factors: the concentrability of the comparator policy $\pi_1^*$, a logarithmic moment term $\Gamma$ stemming from higher–order tail bounds, and the tail probability $\Upsilon$, raised to the quarter power.

**Bounding the good–region term.** Applying Lemma 8 with $f = |D|\,\mathbb{I}_G$ gives

$$\mathbb{E}_{\pi_1^*,\pi_0}\big[|D|\,\mathbb{I}_G\big] \;\leq\; C_{\pi_1^*}^{1/2}\left(\mathbb{E}_{\pi_0,\pi_0}\big[D^2\,\mathbb{I}_G\big]\right)^{1/2}. \tag{30}$$

Note that on $G$ both $|\Delta^{r_0^*}|$ and $|\Delta^{\hat{r}_0}|$ are clipped by $\eta$, so the remaining quantity is a bounded squared error that will be controlled by an empirical process argument.

**Putting the pieces together.** From Eq. 23, Eq. 29, and Eq. 30, we obtain

$$\mathbb{E}_{\pi_1^*,\pi_0}\big[\,|\Delta^{r_0^*}-|\Delta^{\hat{r}_0}|\,\big] \;\lesssim\; C_{\pi_1^*}^{1/2}\left(\mathbb{E}_{\pi_0,\pi_0}\big[|\Delta^{r_0^*}-\Delta^{\hat{r}_0}|^2\,\mathbb{I}_G\big]\right)^{1/2} \;+\; C_{\pi_1^*}^{1/2}\,\Gamma\,\Upsilon^{1/4}. \tag{31}$$

By the same argument with $\pi_1$ in place of $\pi_1^*$,

$$\mathbb{E}_{\pi_1,\pi_0}\big[\,|\Delta^{r_0^*}-|\Delta^{\hat{r}_0}|\,\big] \;\lesssim\; C_{\pi_1}^{1/2}\left(\mathbb{E}_{\pi_0,\pi_0}\big[|\Delta^{r_0^*}-\Delta^{\hat{r}_0}|^2\,\mathbb{I}_G\big]\right)^{1/2} \;+\; C_{\pi_1}^{1/2}\,\Gamma\,\Upsilon^{1/4}. \tag{32}$$

Finally, using $\mathbb{E}[X]\le\mathbb{E}[|X|]$ and substituting Eq. 31–Eq. 32 into Eq. 20, we obtain the intermediate performance–gap bound

$$J_0(\pi_1^*)-J_0(\pi_1) \;\lesssim\; (C_{\pi_1^*}^{1/2}+C_{\pi_1}^{1/2})\left(\mathbb{E}_{\pi_0,\pi_0}\big[|\Delta^{r_0^*}-\Delta^{\hat{r}_0}|^2\,\mathbb{I}_G\big]\right)^{1/2} \;+\; (C_{\pi_1^*}^{1/2}+C_{\pi_1}^{1/2})\,\Gamma\,\Upsilon^{1/4}$$
$$+\; \beta D_{\mathrm{KL}}(\pi_1^*\|\pi_0) \;-\; \beta D_{\mathrm{KL}}(\pi_1\|\pi_0). \tag{33}$$

The first ("good–region") term will be bounded by a uniform convergence argument, while the tail term $\Gamma\,\Upsilon^{1/4}$ is controlled by choosing $\eta$ and the clipping level in the empirical loss appropriately. This completes the good/bad region analysis used to bound the differences of reward deltas.

## D.4 From Empirical Risk Minimization to a Population-Level Bound

The bound derived in Eq. 33 depends on the population-level squared difference $\mathbb{E}_{\pi_0,\pi_0}[|\Delta^{r_0^*}-\Delta^{\hat{r}_0}|^2\mathbb{I}_G]$. In this section, we establish a connection between this population quantity and the empirical performance of the learned policy $\pi_1$, which is obtained through Empirical Risk Minimization (ERM). To achieve this, we leverage classical tools from uniform convergence theory, which allow us to translate the ERM guarantee into a high-probability control of the population-level error term.

**The ERM Solution and Its Empirical Loss.** To simplify notation, we index the pairwise difference operator by a policy $\pi$:

$$\Delta^\pi(x,y,y') := \beta\log\frac{\pi(y\mid x)}{\pi_0(y\mid x)} - \beta\log\frac{\pi(y'\mid x)}{\pi_0(y'\mid x)}.$$

From the reward identity in Eq. 15, we note that $\Delta^{r_0^*}=\Delta^{\pi_{1,\beta}^*}$. The policy $\pi_1$ is defined as the ERM solution over the class $\Pi$, minimizing the squared loss with respect to the oracle $\pi_{1,\beta}^*$. Since the oracle belongs to the hypothesis class, $\pi_{1,\beta}^*\in\Pi$, the empirical risk of $\pi_1$ must be no larger than that of $\pi_{1,\beta}^*$, which is zero. Formally,

$$\sum_{(x,y,y')\in\mathcal{D}_0}\big(\Delta^{\pi_1}(x,y,y')-\Delta^{\pi_{1,\beta}^*}(x,y,y')\big)^2 \;\le\; \min_{\pi\in\Pi}\sum_{(x,y,y')\in\mathcal{D}_0}\big(\Delta^\pi-\Delta^{\pi_{1,\beta}^*}\big)^2 \;\le\; 0. \tag{34}$$

**A Uniform Convergence Bound via Bernstein's Inequality.** To generalize the zero empirical loss to the population setting, we appeal to Bernstein's inequality together with a union bound over the policy class. Let $\mathcal{D}_0=\{(x_i,y_i,y_i')\}_{i=1}^n$ be i.i.d. samples with $x_i\sim\mu$ and $y_i,y_i'\sim\pi_0(\cdot\mid x_i)$.

To apply concentration inequalities, we require bounded variables. For any policy $\pi\in\Pi$, we define the clipped squared loss

$$Z_i(\pi) := \big(\Delta^\pi(x_i,y_i,y_i')-\Delta^{\pi_{1,\beta}^*}(x_i,y_i,y_i')\big)^2\cdot\mathbb{I}\big\{|\Delta^\pi(x_i,y_i,y_i')|\le B,\ |\Delta^{\pi_{1,\beta}^*}(x_i,y_i,y_i')|\le B\big\},$$

where the clipping threshold $B=B_{n,\rho}$ will be chosen later. The empirical and population losses are then

$$\hat{L}_n(\pi) := \frac{1}{n}\sum_{i=1}^n Z_i(\pi), \qquad L(\pi) := \mathbb{E}[Z_i(\pi)].$$

On the clipping event, both differences are bounded by $B$, hence $|\Delta^\pi-\Delta^{\pi_{1,\beta}^*}|\le 2B$. Thus the clipped loss is uniformly bounded as

$$0 \;\le\; Z_i(\pi) \;\le\; M := (2B)^2. \tag{35}$$

Because $\pi_{1,\beta}^* \in \Pi$ and by construction $Z_i(\pi_{1,\beta}^*) \equiv 0$, the ERM property Eq. 34 implies

$$\hat{L}_n(\pi_1) \leq \hat{L}_n(\pi_{1,\beta}^*) = 0.$$

Since the clipped loss is always nonnegative, it follows that

$$\hat{L}_n(\pi_1) = 0. \tag{36}$$

For any fixed $\pi$, the variables $\{Z_i(\pi)\}_{i=1}^n$ are i.i.d. and bounded in $[0, M]$ by Eq. 35. Bernstein's inequality then yields, for all $t > 0$,

$$\mathbb{P}\left(L(\pi) - \hat{L}_n(\pi) \geq t\right) \leq \exp\left(-\frac{nt^2}{2\operatorname{Var}(Z_i(\pi)) + \frac{2}{3}Mt}\right).$$

Using the variance bound $\operatorname{Var}(Z_i(\pi)) \leq ML(\pi)$, a standard fixed–point calculation leads to the empirical Bernstein form: with probability at least $1 - \delta$,

$$L(\pi) \leq 2\hat{L}_n(\pi) + c\frac{M\log(1/\delta)}{n}, \tag{37}$$

for a universal constant $c > 0$.

Applying Eq. 115 with $\delta = \rho/|\Pi|$ and union–bounding over all $\pi \in \Pi$, we obtain that with probability at least $1 - \rho$,

$$\forall \pi \in \Pi : \qquad L(\pi) \leq 2\hat{L}_n(\pi) + c\frac{M\log(|\Pi|/\rho)}{n}. \tag{38}$$

Since Eq. 38 holds uniformly over $\Pi$, it applies in particular to the ERM policy $\pi_1$. Combining with Eq. 36 yields

$$L(\pi_1) \leq c\frac{M\log(|\Pi|/\rho)}{n}.$$

Substituting $M = (2B)^2$ gives the population-level bound

$$L(\pi_1) \lesssim \frac{B^2\log(|\Pi|/\rho)}{n}. \tag{39}$$

Finally, we specify the clipping level

$$B_{n,\rho} := \log\left(\frac{2n\,C_{\text{loss}}\,|\Pi|}{\rho}\right).$$

This selection is motivated by the tail-probability bound provided in Lemma J.1 of Huang et al. (2025). which ensures that the probability of clipping violations is sufficiently small, of order at most $\rho/(n|\Pi|)$. With this selection, we conclude that

$$\mathbb{E}_{\pi_0,\pi_0}\left[\left(\Delta^{\pi_1} - \Delta^{\pi_{1,\beta}^*}\right)^2 \mathbb{I}\{|\Delta^{\pi_1}| \leq B_{n,\rho},\ |\Delta^{\pi_{1,\beta}^*}| \leq B_{n,\rho}\}\right] \lesssim \frac{B_{n,\rho}^2\log(|\Pi|/\rho)}{n}. \tag{40}$$

This completes the transition from the empirical ERM guarantee to a high-probability population-level error bound.

## D.5 Finalizing the Performance Gap and Deriving the Suboptimality Bound

In this concluding part of the single-step analysis, we integrate the statistical error bound derived in the previous section with the main performance gap inequality. This synthesis yields a concrete, high-probability guarantee on the policy's performance, which we then translate into a bound on the probability mass that the policy assigns to the optimal actions.

**Simplifying the Performance Gap Bound.** We begin by substituting the statistical error term $\varepsilon_{\text{stat}}$ from Eq. 40 into the performance gap inequality Eq. 33. This yields

$$J_0(\pi_1^*) - J_0(\pi_1) \;\lesssim\; \big(C_{\pi_1^*}^{1/2} + C_{\pi_1}^{1/2}\big)\, \varepsilon_{\text{stat}} \;+\; \big(C_{\pi_1^*}^{1/2} + C_{\pi_1}^{1/2}\big) \log(C_{\text{loss}})\, \rho^{1/4} + \beta D_{\text{KL}}(\pi_1^* \| \pi_0). \quad (41)$$

To simplify this expression, we upper bound the policy-dependent coefficients with problem-level constants. Specifically, by Lemma 4.1 in Huang et al. (2025), we have

$$C_{\pi_1^*} \le \kappa_0, \qquad C_{\pi_1} \le \alpha \kappa_0.$$

Moreover, we control the KL divergence by the logarithm of the concentrability coefficient:

$$D_{\text{KL}}(\pi_1^* \| \pi_0) \;\le\; \log C_{\pi_1^*} \;\le\; \log \kappa_0.$$

Substituting these bounds gives

$$J_0(\pi_1^*) - J_0(\pi_1) \;\lesssim\; \big(\kappa_0^{1/2} + \alpha \kappa_0^{1/2}\big)\, \varepsilon_{\text{stat}} \;+\; \big(\kappa_0^{1/2} + \alpha \kappa_0^{1/2}\big) \log(C_{\text{loss}})\, \rho^{1/4} \;+\; \beta \log(\kappa_0).$$

For clarity, and because $\alpha \kappa_0$ is typically of the same order as $\kappa_0$ in single-step analysis, we further simplify to

$$J_0(\pi_1^*) - J_0(\pi_1) \;\lesssim\; \kappa_0^{1/2}\, \varepsilon_{\text{stat}} \;+\; \kappa_0^{1/2} \log(C_{\text{loss}})\, \rho^{1/4} \;+\; \beta \log(\kappa_0).$$

To obtain a cleaner form, we balance the statistical error term and the tail-event term by setting

$$\rho' := \rho \wedge \left( \frac{\varepsilon_{\text{stat}}}{\log C_{\text{loss}}} \right)^4.$$

This ensures that the second term does not dominate the first. The inequality then reduces to

$$J_0(\pi_1^*) - J_0(\pi_1) \;\lesssim\; \kappa_0^{1/2}\, \varepsilon_{\text{stat}} \;+\; \beta \log(\kappa_0). \quad (42)$$

**Bounding the Suboptimality Probability.** We now translate this bound into a high-probability guarantee on the probability mass assigned to optimal actions. Substituting the above inequality into Lemma 7 (see also Eq. 10), we obtain

$$\mathbb{P}_{x \sim \mu}\big[\pi_1(y_0^*(x) \mid x) \le 1 - \delta\big] \le \frac{J_0(\pi_1^*) - J_0(\pi_1)}{\gamma\,\delta}$$

$$\lesssim \frac{1}{\gamma\delta}\Big(\kappa_0^{1/2}\, \varepsilon_{\text{stat}} \;+\; \beta \log(\kappa_0)\Big). \quad (43)$$

Recalling that the statistical error satisfies

$$\varepsilon_{\text{stat}}^2 \;\lesssim\; \frac{B_{n,\rho}^2 \log(|\Pi|/\rho)}{n}, \qquad B_{n,\rho} \;\asymp\; \log\big(n|\Pi|\rho^{-1}\big),$$

we conclude

$$\mathbb{P}_{x \sim \mu}\big[\pi_1(y_0^*(x) \mid x) \le 1 - \delta\big] \;\lesssim\; \frac{1}{\gamma\delta}\Big(\kappa_0^{1/2}\, \log(n|\Pi|\rho^{-1})\, n^{-1/2} \;+\; \beta \log(\kappa_0)\Big). \quad (44)$$

This provides a direct guarantee on the frequency with which the learned policy under-weights the oracle's optimal action.

**Extending the Bound to the Policy's Own Optimal Action.** For practical purposes, it is natural to bound the event in which the learned policy's *own* maximizer receives insufficient probability mass. Define

$$y_1^*(x) := \arg\max_{y \in \mathcal{Y}} \pi_1(y \mid x).$$

By construction,

$$\pi_1(y_1^*(x) \mid x) \;\ge\; \pi_1(y_0^*(x) \mid x).$$

Hence, whenever the event $\{\pi_1(y_1^*(x) \mid x) \leq 1 - \delta\}$ occurs, it must also be the case that $\pi_1(y_0^*(x) \mid x) \leq 1 - \delta$. In set-theoretic form,

$$\{x : \pi_1(y_1^*(x) \mid x) \leq 1 - \delta\} \ \subseteq \ \{x : \pi_1(y_0^*(x) \mid x) \leq 1 - \delta\}.$$

Taking probabilities and applying Eq. 44 yields

$$\mathbb{P}_{x\sim\mu}\big[\pi_1(y_1^*(x) \mid x) \leq 1 - \delta\big] \ \leq \ \mathbb{P}_{x\sim\mu}\big[\pi_1(y_0^*(x) \mid x) \leq 1 - \delta\big]$$
$$\lesssim \ \frac{1}{\gamma\delta}\Big(\kappa_0^{1/2} \log(n|\Pi|\rho^{-1})\, n^{-1/2} \ + \ \beta \log(\kappa_0)\Big). \tag{45}$$

This completes the one-step analysis. The derived inequality provides a high-probability guarantee that the learned policy $\pi_1$ assigns sufficient mass to its own optimal actions, with explicit dependence on the sample size $n$, the policy class size $|\Pi|$, and the policy condition number $\kappa_0$.

### D.6 RECURSIVE BOUND FOR THE POLICY CONDITION NUMBER

The analysis so far has provided a crucial single-step guarantee on the policy's suboptimality. However, to understand the long-term behavior of the self-rewarding process, it is essential to move beyond a single iteration and demonstrate that the policy's quality, as measured by the policy condition number, does not degrade over successive generations of training. An uncontrolled growth in the policy condition number would signify *model collapse*, where the model becomes increasingly narrow and overconfident in its own outputs, thereby losing its ability to generalize. The central argument of our framework rests on proving that the policy condition number at one step is controlled by, and does not grow excessively relative to, that of the previous step. This stability property is formalized in the following lemma.

**Lemma 9** (One-Step Policy Condition Number Recurrence)**.** *Let $\kappa_{t-1}$ denote the policy condition number at step $t - 1$. Assume that for any policy $\pi_t$, the probabilities assigned to any action are uniformly lower-bounded by a constant $c > 0$. Then, the policy condition number at step $t$, denoted $\kappa_t$, satisfies the recursive inequality*

$$\kappa_t \ \lesssim \ 1 + \frac{1}{c} + \Big(1 + \frac{1}{c}\Big)\frac{\log(n|\Pi|\rho^{-1})}{\gamma\delta\sqrt{n}}\sqrt{\kappa_{t-1}} + \Big(1 + \frac{1}{c}\Big)\frac{\beta}{\gamma\delta}\log\big(\kappa_{t-1}\big). \tag{46}$$

*Proof.* We illustrate the argument for the inductive step from $t = 0$ to $t = 1$; the general case follows by re-indexing.

#### D.6.1 TAIL-INTEGRAL REPRESENTATION.

We recall that the policy condition number is defined as

$$\kappa_1 = \mathbb{E}_{x\sim\mu}\left[\frac{1}{\pi_1(y_1^*(x) \mid x)}\right], \tag{47}$$

where

$$y_1^*(x) := \arg\max_{y\in\mathcal{Y}} \pi_1(y \mid x).$$

Introducing the random variable

$$Z(x) := \frac{1}{\pi_1(y_1^*(x) \mid x)},$$

we see that $\kappa_1 = \mathbb{E}[Z(x)]$. A useful way to analyze such expectations is through the tail-integral (or layer-cake) representation:

$$\kappa_1 = \mathbb{E}_{x\sim\mu}[Z(x)] = \int_0^\infty \mathbb{P}_{x\sim\mu}\big(Z(x) \geq t\big)\, dt. \tag{48}$$

By assumption, the policy assigns at least $c > 0$ probability to every action, i.e. $\pi_1(y \mid x) \geq c$ for all $(x, y)$. Consequently, $Z(x)$ is bounded:

$$1 \ \leq \ Z(x) \ \leq \ \frac{1}{c}.$$

This bounded support implies the following trivial bounds on the tail probability:

$$\mathbb{P}\big(Z(x) \geq t\big) = \begin{cases} 1, & 0 \leq t < 1, \\ 0, & t > \frac{1}{c}. \end{cases}$$

Substituting these observations into Eq. 124, we split the expectation into three parts:

$$\mathbb{E}[Z(x)] = \int_0^1 \underbrace{\mathbb{P}(Z(x) \geq t)}_{=1} \, dt + \int_1^{1/c} \mathbb{P}(Z(x) \geq t) \, dt + \int_{1/c}^{\infty} \underbrace{\mathbb{P}(Z(x) \geq t)}_{=0} \, dt$$

$$= 1 + \int_1^{1/c} \mathbb{P}(Z(x) \geq t) \, dt. \tag{49}$$

Thus, the analysis of $\kappa_1$ reduces to bounding the integral of the non-trivial tail probabilities over the interval $[1, 1/c]$.

### D.6.2 RELATING THE TAIL TO SUBOPTIMALITY.

For $t \geq 1$, the event $\{Z(x) \geq t\}$ is equivalent to

$$\pi_1(y_1^*(x) \mid x) \leq \frac{1}{t}.$$

Writing $\delta = 1 - \frac{1}{t}$, this becomes $\pi_1(y_1^*(x) \mid x) \leq 1 - \delta$. From Eq. 45, we know

$$\mathbb{P}_{x\sim\mu}\big(\pi_1(y_1^*(x) \mid x) \leq 1 - \delta\big) \lesssim \frac{1}{\gamma\delta}\Big(\kappa_0^{1/2} \frac{\log(n|\Pi|\rho^{-1})}{\sqrt{n}} + \beta \log(\kappa_0)\Big). \tag{50}$$

Substituting $\delta = 1 - \frac{1}{t}$ gives

$$\mathbb{P}(Z(x) \geq t) \lesssim \frac{t}{t-1} \cdot \frac{1}{\gamma}\Big(\kappa_0^{1/2} \frac{\log(n|\Pi|\rho^{-1})}{\sqrt{n}} + \beta \log(\kappa_0)\Big). \tag{51}$$

Let

$$A := \frac{1}{\gamma}\Big(\kappa_0^{1/2} \frac{\log(n|\Pi|\rho^{-1})}{\sqrt{n}} + \beta \log(\kappa_0)\Big).$$

Then

$$\mathbb{P}(Z(x) \geq t) \lesssim \frac{A}{t-1}t. \tag{52}$$

To respect probability upper bounds, we refine this as

$$\mathbb{P}(Z(x) \geq t) \leq \min\Big\{1, \frac{A}{t-1}t\Big\}. \tag{53}$$

### D.6.3 DERIVATION OF AN INTEGRAL BOUND ON THE POLICY CONDITION NUMBER

Then, We provide a complete and self–contained derivation of the integral that upper–bounds the one–step policy condition number $\kappa_1$. As shown earlier, defining $Z(x) := 1/\pi_1\big(y_1^*(x) \mid x\big)$ and using the tail–integral identity yields

$$\kappa_1 = \mathbb{E}[Z(x)] \leq 1 + \int_1^{\frac{1}{c}} \min\big\{1, \, B(t)\big\} \, dt, \qquad B(t) := A \frac{t}{t-1}, \tag{54}$$

where $c \in (0,1]$ is the uniform lower bound on action probabilities and $A \in (0,1)$ is the parameter that emerges from the suboptimality bound (its explicit form is given in the main text; here we only use that $A < 1$). The remainder of the argument is a careful evaluation of the integral in Eq. 54.

**Step 1: Identifying the transition point.** The integrand $\min\{1, B(t)\}$ switches behavior at the unique $t_0 > 1$ solving $B(t_0) = 1$. Solving

$$A \frac{t_0}{t_0 - 1} = 1 \quad \Longleftrightarrow \quad At_0 = t_0 - 1 \quad \Longleftrightarrow \quad t_0(1 - A) = 1,$$

we obtain the explicit transition point

$$t_0 = \frac{1}{1 - A}, \tag{55}$$

which is well defined under the nontrivial regime $A < 1$. For $t \in [1, t_0]$ we have $B(t) \geq 1$ and hence $\min\{1, B(t)\} = 1$, while for $t \in [t_0, 1/c]$ we have $B(t) \leq 1$ and hence $\min\{1, B(t)\} = B(t)$.

**Step 2: Decomposing the integral.** Using the transition point Eq. 131, we split

$$\int_1^{\frac{1}{c}} \min\{1, B(t)\}\, dt = \int_1^{t_0} 1\, dt \;+\; \int_{t_0}^{\frac{1}{c}} B(t)\, dt = (t_0 - 1) \;+\; \int_{t_0}^{\frac{1}{c}} A\, \frac{t}{t-1}\, dt. \tag{56}$$

The first term evaluates immediately to $t_0 - 1$. It remains to compute the second term.

**Step 3: Evaluating the second term via an antiderivative.** We simplify the integrand by the identity $\frac{t}{t-1} = 1 + \frac{1}{t-1}$, and integrate term–wise:

$$\int \frac{t}{t-1}\, dt = \int \left(1 + \frac{1}{t-1}\right) dt = t + \ln|t - 1| \;=:\; F_0(t).$$

Therefore,

$$\int_{t_0}^{\frac{1}{c}} A\, \frac{t}{t-1}\, dt = A\big[F_0(1/c) - F_0(t_0)\big]$$
$$= A\Big[\Big(\frac{1}{c} - t_0\Big) + \ln\Big(\frac{1/c - 1}{t_0 - 1}\Big)\Big],$$

where we used $\ln a - \ln b = \ln(a/b)$ and the fact that $t_0 > 1$ ensures $t_0 - 1 > 0$.

**Step 4: Assembling the bound and simplifying.** Combining the pieces from Eq. 56, we obtain

$$\kappa_1 \leq 1 + (t_0 - 1) + A\Big(\frac{1}{c} - t_0\Big) + A\ln\Big(\frac{1/c - 1}{t_0 - 1}\Big). \tag{57}$$

We now substitute $t_0 = \frac{1}{1-A}$ from Eq. 131. First, the linear term simplifies to

$$t_0(1 - A) \;=\; \frac{1}{1-A}\,(1 - A) = 1.$$

Second, $t_0 - 1 = \frac{1}{1-A} - 1 = \frac{A}{1-A}$. Using these identities in Eq. 57 yields

$$\kappa_1 \leq 1 + \frac{A}{c} + A\ln\Big(\frac{(1/c - 1)}{A/(1-A)}\Big) = 1 + \frac{A}{c} + A\ln\Big(\frac{(1-c)(1-A)}{A\,c}\Big).$$

We have thus established the explicit integral evaluation:

$$\kappa_1 \;\leq\; 1 + \frac{A}{c} \;+\; A\ln\Big(\frac{(1-c)(1-A)}{A\,c}\Big) \tag{58}$$

*Interpretation and a relaxed form.* For small $A$ (the regime of interest), $\ln(1 - A) \approx 0$ and the contribution of $\ln(1/Ac)$ dominates. A convenient relaxation is then

$$\kappa_1 \;\lesssim\; 1 + \frac{A}{c} \;+\; A\ln\Big(\frac{1}{Ac}\Big) \;\lesssim\; 1 + \frac{1}{c} + \Big(1 + \frac{1}{c}\Big)A,$$

which is often more transparent in subsequent recursive arguments. (The exact definition of $A$—a function of $\gamma, \delta, n, |\Pi|, \rho, \beta$, and $\kappa_0$—is given in the main text and can be substituted as needed.)

**Step 5: Final Substitution and the Recursive Relation** We now return to the simplified linearized bound obtained in the previous section and explicitly substitute the full definition of the constant $A$. Recall that $A$ was defined in terms of the statistical error and the KL regularization component, yielding

$$A \;=\; \frac{1}{\gamma\delta}\left(\kappa_0^{\frac{1}{2}} \cdot \frac{\log(n|\Pi|\rho^{-1})}{\sqrt{n}} \;+\; \beta\log\kappa_0\right).$$

Substituting this expression back into the approximate linear coverage bound, we obtain

$$\kappa_1 \;\lesssim\; 1 + \frac{1}{c} + \Big(1 + \frac{1}{c}\Big)\left[\frac{1}{\gamma\delta}\left(\kappa_0^{\frac{1}{2}}\frac{\log(n|\Pi|\rho^{-1})}{\sqrt{n}} + \beta\log(\kappa_0)\right)\right]. \tag{59}$$

Expanding the terms inside the brackets distributes the dependence on the statistical complexity and the KL penalty, leading to a more transparent form of the recursion:

$$\kappa_1 \; \lesssim \; 1 + \frac{1}{c} + \left(1 + \frac{1}{c}\right) \frac{\log(n|\Pi|\rho^{-1})}{\gamma\delta\sqrt{n}} \sqrt{\kappa_0} + \left(1 + \frac{1}{c}\right) \frac{\beta}{\gamma\delta} \log(\kappa_0). \tag{60}$$

This recursive inequality explicitly highlights the threefold structure of the coverage update: (i) a constant offset $1 + 1/c$ due to the minimal probability assumption, (ii) a term scaling with $\sqrt{\kappa_0}$, which reflects the statistical complexity inherited from the previous step, and (iii) a logarithmic dependence on $\kappa_0$ introduced by the KL penalty.

**Step 6: Extension to Arbitrary Generation Steps by Induction**    The derivation so far has been presented for the transition from the base step ($t = 0$) to the first update ($t = 1$). However, the underlying reasoning does not rely on any special property of these indices. At a conceptual level, the argument merely exploits two facts: (i) the tail–integral identity applies uniformly at each step, and (ii) the suboptimality bound remains valid for every policy $\pi_t$ with coverage coefficient $\kappa_t$.

Consequently, the same sequence of inequalities can be applied recursively at each generation step. By induction, the bound that relates $\kappa_1$ to $\kappa_0$ extends to a general recurrence relation between $\kappa_t$ and $\kappa_{t-1}$ for all $t \geq 1$. Formally, we obtain

$$\kappa_t \; \lesssim \; 1 + \frac{1}{c} + \left(1 + \frac{1}{c}\right) \frac{\log(n|\Pi|\rho^{-1})}{\gamma\delta\sqrt{n}} \sqrt{\kappa_{t-1}} + \left(1 + \frac{1}{c}\right) \frac{\beta}{\gamma\delta} \log(\kappa_{t-1}). \tag{61}$$

**Interpretation.**    This recurrence constitutes a stability guarantee for the policy condition number. It demonstrates that each new generation of the policy inherits its complexity from the previous generation in a controlled manner: the square–root scaling prevents explosive growth, while the logarithmic correction encapsulates the effect of KL regularization. Therefore, the recursive inequality Eq. 61 forms the theoretical backbone ensuring that the policy condition number does not diverge across iterations, thereby precluding collapse and preserving generalization capacity throughout the self-rewarding process.

$$\square$$

### D.7    ASYMPTOTIC STABILITY OF THE POLICY CONDITION NUMBER

**Lemma 10** (Asymptotic Stability of the Policy Condition Number). *Let the sequence of policy condition numbers $\{\kappa_t\}_{t=0}^T$ be governed by the recursive inequality derived in Eq. 61. Then, for a sufficiently small learning rate parameter $\beta$ (specifically, $\beta \lesssim n^{-1/2}$), the policy condition number at generation $T$ remains bounded. Moreover, the sequence converges exponentially fast to a finite stable value, ensuring the stability of the self-rewarding process. In particular, the explicit asymptotic bound satisfies*

$$\kappa_t \; \lesssim \; \frac{1}{c} + \frac{1}{n} \left(\frac{\log(|\Pi|\rho^{-1})}{c\gamma\delta}\right)^2 + \left(1 + \sqrt{nc}\right)^{-T} \kappa_0. \tag{62}$$

*Proof.* We analyze the recurrence step by step, beginning with a reformulation and then studying its fixed point and rate of convergence.

**Step 1: Compact Form of the Recurrence.**    Let us define the shorthand

$$M_0 := 1 + \frac{1}{c}, \qquad \text{(baseline constant)} \tag{63}$$

$$M_1 := \left(1 + \frac{1}{c}\right) \frac{\log(n|\Pi|\rho^{-1})}{\gamma\delta\sqrt{n}}, \qquad \text{(square-root coefficient)} \tag{64}$$

$$M_2 := \left(1 + \frac{1}{c}\right) \frac{\beta}{\gamma\delta}, \qquad \text{(logarithmic coefficient).} \tag{65}$$

Then the recursive inequality in Eq. 61 can be rewritten in the compact form

$$\kappa_t \; \leq \; M_0 + M_1\sqrt{\kappa_{t-1}} + M_2 \log(\kappa_{t-1}). \tag{66}$$

**Step 2: Bounding the Logarithmic Term.** Since $\kappa_{t-1} \geq 1$, we apply the inequality $\log x \leq \frac{2}{e}\sqrt{x}$ for all $x \geq 1$. Substituting into Eq. 66 yields

$$
\begin{aligned}
\kappa_t &\leq M_0 + M_1\sqrt{\kappa_{t-1}} + \frac{2M_2}{e}\sqrt{\kappa_{t-1}} \\
&\leq M_0 + \left(M_1 + \frac{2M_2}{e}\right)\sqrt{\kappa_{t-1}}.
\end{aligned}
\tag{67}
$$

Let $K := M_1 + \frac{2M_2}{e}$, so that the recurrence simplifies to

$$
\kappa_t \leq M_0 + K\sqrt{\kappa_{t-1}}.
\tag{68}
$$

This gives a cleaner functional recurrence governed by

$$
h(\kappa) := M_0 + K\sqrt{\kappa}.
$$

**Step 3: Fixed Point of the Recurrence.** The asymptotic behavior is determined by the fixed point $U$ of $h$, which satisfies

$$
U = M_0 + K\sqrt{U}.
\tag{69}
$$

Setting $y = \sqrt{U}$ yields the quadratic equation

$$
y^2 - Ky - M_0 = 0.
$$

By the quadratic formula,

$$
y = \frac{K \pm \sqrt{K^2 + 4M_0}}{2}.
$$

Since $y = \sqrt{U} > 0$, we choose the positive solution. Hence,

$$
\sqrt{U} = \frac{K + \sqrt{K^2 + 4M_0}}{2}, \quad U = \left(\frac{K + \sqrt{K^2 + 4M_0}}{2}\right)^2.
\tag{70}
$$

**Step 4: Convergence Rate via Contraction.** We now analyze the rate at which the recurrence converges to its fixed point $U$. Recall that the recurrence is governed by

$$
\kappa_t' = h(\kappa_{t-1}'), \quad \text{with } h(\kappa) = M_0 + K\sqrt{\kappa}.
$$

Since $h$ is monotone increasing and satisfies $h(\kappa) < \kappa$ for all $\kappa > U$, the sequence $\{\kappa_t'\}$, when initialized from $c > U$, is monotonically decreasing and bounded below by $U$. This ensures that $\{\kappa_t'\}$ converges to $U$.

To quantify the convergence rate, we invoke the Mean Value Theorem. For any $\kappa > U$, there exists some $\xi \in (U, \kappa)$ such that

$$
|h(\kappa) - U| = |h(\kappa) - h(U)| = |h'(\xi)|\,|\kappa - U|.
\tag{71}
$$

The derivative of $h$ is given by

$$
h'(\kappa) = \frac{K}{2\sqrt{\kappa}}.
$$

Since $h'(\kappa)$ is a decreasing function of $\kappa$, its supremum on the interval $[U, \infty)$ is attained at $\kappa = U$. Hence, the contraction factor is

$$
q := \sup_{\kappa \geq U} h'(\kappa) = h'(\kappa) = \frac{K}{2\sqrt{U}}.
\tag{72}
$$

Substituting the explicit expression of $U$ from Eq. 70, we obtain

$$
q = \frac{K}{K + \sqrt{K^2 + 4M_0}}.
\tag{73}
$$

It is immediate that $0 < q < 1$, which confirms that $h$ is a contraction mapping on $[U, \infty)$. Consequently, the sequence $\{\kappa_t'\}$ converges geometrically to its fixed point $U$.

Unrolling the recurrence, we arrive at the inequality

$$
|\kappa_t' - U| \leq q|\kappa_{t-1}' - U| \quad \implies \quad |\kappa_T' - U| \leq q^T|c - U|,
\tag{74}
$$

which provides an explicit bound on the deviation of $\kappa_T'$ from its fixed point $U$. This establishes not only convergence but also the geometric rate of convergence governed by the contraction factor $q$.

**Step 5: Explicit Upper Bound for $\kappa_t$.** Since the original sequence $\{\kappa_t\}$ is pointwise dominated by the auxiliary sequence $\{\kappa'_t\}$, we may use the contraction analysis of Step 4 to bound $\kappa_t$. Because $\{\kappa'_t\}$ is monotonically decreasing from its initialization $\kappa'_0 = \kappa_0$ towards the fixed point $U$, we have

$$\kappa'_T - U = |\kappa'_T - U|.$$

Therefore, an explicit upper bound for the original policy condition number after $T$ generations is

$$\kappa_t \;\leq\; U + q^T(c - U), \tag{75}$$

where $U$ is the asymptotic fixed point defined in Eq. 70, and $q$ is the geometric contraction factor derived in Eq. 73.

This inequality shows that even if the initial coverage $\kappa_0$ is large, the sequence converges exponentially fast towards a finite stable value $U$. Thus the self-rewarding process is inherently stable and avoids uncontrolled growth of the policy condition number.

Expanding the bound more explicitly, we obtain

$$
\begin{aligned}
\kappa_T &\leq U + q^T(\kappa_0 - U) \\
&\leq (1 - q^T)U + q^T \kappa_0 \\
&= \left( \frac{K + \sqrt{K^2 + 4M_0}}{2} \right)^2 + \left( \frac{K}{K + \sqrt{K^2 + 4M_0}} \right)^T \kappa_0,
\end{aligned} \tag{76}
$$

where we have substituted the closed-form expressions for $U$ and $q$.

To further interpret this result, we approximate the asymptotic bound $U$ for large $n$. Recalling that $K \approx M_1 + M_2$, and ignoring higher-order cross-terms, we obtain

$$U \;\approx\; M_0 + M_1^2 + M_2^2. \tag{77}$$

Hence the recursive bound simplifies to

$$\kappa_T \;\lesssim\; M_0 + M_1^2 + M_2^2 + \left( \frac{1}{1 + \sqrt{M_0/(M_1^2 + M_2^2)}} \right)^T \kappa_0. \tag{78}$$

Equation Eq. 78 makes explicit the decomposition of the bound: the terms $M_0$, $M_1^2$, and $M_2^2$ determine the asymptotic ceiling, while the multiplicative factor $q^T$ ensures exponential decay of the dependence on the initial coverage $\kappa_0$. This completes the derivation of the explicit upper bound for $\kappa_t$.

**Step 6: Scaling Analysis.** Choosing $\beta \lesssim n^{-1/2}$ balances the scaling of $M_1$ and $M_2$, giving

$$U \;\lesssim\; \frac{1}{c} + \frac{1}{n} \left( \frac{\log(n|\Pi|\rho^{-1})}{c\gamma\delta} \right)^2,$$

and the contraction factor simplifies to

$$q = \frac{K}{K + \sqrt{K^2 + 4M_0}} \;\approx\; \frac{1}{1 + \sqrt{nc}}.$$

Hence,

$$\kappa_t \;\lesssim\; \frac{1}{c} + \frac{1}{n} \left( \frac{\log(|\Pi|\rho^{-1})}{c\gamma\delta} \right)^2 + \left( 1 + \sqrt{nc} \right)^{-T} \kappa_0. \tag{79}$$

This establishes both the boundedness and exponential convergence of the policy condition number sequence. $\qquad\square$

## D.8 Conclusion of the Proof: Bounding the Final Policy's Tail Probability.

Having established in Lemma 10 that the policy condition number $\kappa_t$ is asymptotically stable and converges exponentially to a finite upper bound, we now translate this stability result into a final guarantee on the predictive confidence of the learned policy. Specifically, we seek to bound the probability that the policy $\pi_T$ assigns insufficient mass to its own optimal response $y_T^*(x)$ after $T$ self-rewarding iterations.

From the preceding derivation, we have for any $T \geq 1$,

$$\mathbb{P}_{x \sim \mu}\big[\pi_T(y_T^*(x) \mid x) \leq 1 - \delta\big] \lesssim \frac{1}{\gamma\delta}\left(\kappa_{t-1}^{1/2}\,\frac{\log(n|\Pi|\rho^{-1})}{\sqrt{n}} + \beta\log(\kappa_{t-1})\right). \tag{80}$$

The key step is to substitute the asymptotic bound for $\kappa_{t-1}$ obtained in Eq. 79 into this inequality. Recall that the policy condition number decomposes into a *stable part*, dominated by $1/c$ for large $n$, and a *transient part* that decays exponentially with $T-1$.

**Step 1 (Square-root term).** Applying the inequality $(a+b)^{1/2} \leq \sqrt{a} + \sqrt{b}$ to the decomposition of $\kappa_{t-1}$, we have

$$\kappa_{t-1}^{1/2} \lesssim \left(\frac{1}{c} + \frac{1}{n}\Big(\frac{\log(|\Pi|\rho^{-1})}{\gamma\delta}\Big)^2\right)^{1/2} + \Big((1+\sqrt{nc})^{-(T-1)}\kappa_0\Big)^{1/2}$$

$$\lesssim \frac{1}{\sqrt{c}} + (1+\sqrt{nc})^{-\frac{T-1}{2}}\sqrt{\kappa_0}. \tag{81}$$

For sufficiently large $n$, the second-order term $\frac{1}{n}(\cdot)^2$ is negligible compared to $1/c$.

**Step 2 (Logarithmic term).** Since $\kappa_{t-1}$ is bounded above by a finite constant $U$, its logarithm is also bounded. That is,

$$\log(\kappa_{t-1}) \leq \log(U).$$

With our choice of $\beta \lesssim n^{-1/2}$, the term $\beta\log(\kappa_{t-1})$ is $\mathcal{O}(n^{-1/2})$ and therefore of the same asymptotic order as the square-root term.

**Step 3 (Final substitution).** Plugging these bounds back into Eq. 80, we obtain

$$\mathbb{P}_{x \sim \mu}\big[\pi_T(y_T^*(x) \mid x) \leq 1 - \delta\big] \lesssim \frac{1}{\gamma\delta}\left[\frac{\log(n|\Pi|\rho^{-1})}{\sqrt{n}}\left(\frac{1}{\sqrt{c}} + (1+\sqrt{nc})^{-\frac{T-1}{2}}\sqrt{\kappa_0}\right) + \mathcal{O}\left(\frac{1}{\sqrt{n}}\right)\right]$$

$$\lesssim \frac{\log(n|\Pi|\rho^{-1})}{\gamma\delta\sqrt{nc}} + \frac{\log(n|\Pi|\rho^{-1})}{\gamma\delta\sqrt{n}}(1+\sqrt{nc})^{-\frac{T-1}{2}}\sqrt{\kappa_0}. \tag{82}$$

**Interpretation.** The bound consists of two components:

- A *stable error term*,
$$\frac{\log(n|\Pi|\rho^{-1})}{\gamma\delta\sqrt{nc}},$$
which vanishes at the parametric rate $O(n^{-1/2})$, demonstrating that the final policy becomes increasingly confident with more data.

- A *transient error term*,
$$\frac{\log(n|\Pi|\rho^{-1})}{\gamma\delta\sqrt{n}}(1+\sqrt{nc})^{-\frac{T-1}{2}}\sqrt{\kappa_0},$$
which reflects the dependence on the initial coverage $\kappa_0$ but decays exponentially fast in $T$.

Thus, even if the initial policy is weak (large $\kappa_0$), its influence diminishes exponentially across iterations. The long-run behavior of the self-rewarding process is governed entirely by the stable error term, ensuring that the final policy allocates sufficient probability mass to its own optimal predictions.

**The proof is complete.**

$\square$

# E  Proof of Theorem 1: Lower Bound on Single-Step Failure Rate.

Before proceeding to analyze the behavior of a full, multi-step iterative self-rewarding process, it is essential to first understand the fundamental limitations inherent in the mechanism itself. This section is dedicated to exploring these limits by analyzing the performance of a self-rewarding algorithm within a fixed query budget, $n$. This scenario is equivalent to a single, non-iterative application of the process.

Our objective is to derive a lower bound on the failure rate that any algorithm must incur on some challenging problem instance. The proof proceeds via an adversarial approach: we will construct a class of problem instances designed to be maximally difficult, thereby probing for potential failure modes.

The central finding of this analysis will be that an algorithm's success is critically dependent on the quality of the initial base model, as quantified by its policy condition number, $\kappa_0$. We will demonstrate that a large initial $\kappa_0$ (representing a poor starting model) can lead to a high probability of failure, a scenario termed model collapse. This result is not a critique of a specific algorithm, but rather a revelation of an intrinsic risk. The vulnerability exposed in this section provides the primary motivation for the analysis in the subsequent sections, where we will investigate how an iterative framework can overcome this limitation and guide the process towards a stable and successful outcome.

*Proof of Theorem 1.* Our strategy is to establish a necessary condition on the sample size $n$ required for any algorithm to achieve a given success rate. Let an arbitrary algorithm, after making at most $n$ oracle calls, produce a new policy. We fix a target failure level $\epsilon$. We will construct a specific hard problem instance and demonstrate that if the algorithm achieves an expected success rate of at least $1 - \epsilon$ on this instance, then its sample budget $n$ must necessarily exceed a certain threshold.

By inverting this relationship, we establish the core result: for a fixed budget $n$, there exists a worst-case scenario where the failure rate $\epsilon$ must be at least as large as the value dictated by this threshold. This provides the lower bound on the algorithm's risk.

In this section we *fix* the query budget $n$ (the total number of sample-and-evaluate calls the algorithm may use), and we derive a *lower bound on the failure rate* $\varepsilon$ that any algorithm must incur on some instance from $\Pi$. The proof proceeds in six explicit steps.

### Step 1: Construction of a Hard Problem Instance.

To establish the lower bound, we construct a problem instance designed to be maximally challenging for any sharpening algorithm. The construction begins by defining the spaces for prompts and responses, followed by the strategic design of the data distribution and the model class itself.

Let the prompt space be a discrete set $\mathcal{X} = \{x_0, x_1, \ldots, x_d\}$ and the response space be $\mathcal{Y} = \{y_0, y_1, \ldots, y_M\}$. We fix a margin parameter $\gamma \in (0, 1)$ (which we will later set to $\gamma = \frac{1}{2}$) and an information-sparsity parameter $\Delta \in (0, 1)$.

First, we define a prompt distribution $\mu$ that concentrates most of its mass on a single, uninformative point:

$$\mu := (1 - \Delta)\,\delta_{x_0} + \frac{\Delta}{d}\sum_{i=1}^{d}\delta_{x_i}, \tag{83}$$

where $\delta_x$ denotes the Dirac delta distribution on element $x$. This construction serves a dual purpose. The atom at $x_0$, carrying the majority of the probability mass $(1 - \Delta)$, acts as an uninformative sentinel; an algorithm will frequently sample prompts from this region but will learn nothing to distinguish between the candidate models. Conversely, all distinguishing information resides in the small $\Delta$-fraction of the distribution mass spread across the "informative" points $\{x_1, \ldots, x_d\}$. As will be shown later, this sparsity, controlled by $\Delta$, is instrumental in amplifying the overall failure rate from a more fundamental classification error.

Next, we construct a family of response distributions $(P_0, P_1, \ldots, P_M)$ on $\mathcal{Y}$, designed to be difficult to distinguish from one another. The distribution for the uninformative prompt is deterministic:

$$P_0 = \delta_{y_0}. \tag{84}$$

For the informative prompts, we define a set of nearly uniform distributions, each with a subtle bias towards a specific response:

$$\forall i \geq 1: \quad P_i = \frac{1}{(1-\gamma)M}\delta_{y_i} + \sum_{j \in [M] \setminus \{i\}} \frac{1}{M}\left(1 - \frac{\gamma}{(M-1)(1-\gamma)}\right)\delta_{y_j}. \tag{85}$$

For each $i \geq 1$, the distribution $P_i$ is constructed by taking the uniform distribution over $n$ elements and slightly increasing the probability mass of a single element $y_i$, while uniformly decreasing the mass of all other $M - 1$ elements. The parameter $\gamma$ controls the margin by which $y_i$ is favored, making it the unique but only marginally most probable outcome. For instance, when $\gamma = \frac{1}{2}$, the probabilities are $P_i(y_i) = 2/M$ and $P_i(y_j) = \frac{M-2}{M(M-1)}$ for $j \neq i$, highlighting that the distinguishing signal is subtle. This construction ensures that a large number of samples are required to reliably detect this statistical bias.

Finally, we synthesize these components to define the full model class. For any index vector $\mathcal{I} = (j_1, \ldots, j_d) \in [M]^d$, which serves as a hidden "secret key," we define the corresponding base model $\pi^{\mathcal{I}}$ as:

$$\pi^{\mathcal{I}}(\cdot \mid x_i) = \begin{cases} P_0, & i = 0, \\ P_{j_i}, & i > 0. \end{cases} \tag{86}$$

The model class $\Pi$ is then the set of all such models generated by every possible secret key:

$$\Pi := \{\pi^{\mathcal{I}} : \mathcal{I} \in [M]^d\}, \qquad \text{which implies} \qquad \log|\Pi| = d\log M. \tag{87}$$

Through this construction, the task is effectively reduced to a set of $d$ independent $n$-way classification problems. Specifically, for each informative prompt $x_i$, the algorithm's goal is to identify the single hidden "correct" label $j_i$, as $y_{j_i}$ is the unique maximizer under the response distribution $P_{j_i}$. The overall difficulty of this task, and thus the reason this construction is suitable for proving a lower bound, is compounded by two strategically designed factors.

First, the statistical similarity of the candidate distributions makes the classification inherently challenging. The distribution $P_{j_i}$ is only *slightly* biased toward its maximizer $y_{j_i}$, meaning a large number of samples at prompt $x_i$ are required to reliably detect this subtle statistical bump. With a limited sample budget, many of the hidden labels $j_i$ will therefore remain ambiguous to the algorithm. Second, an informational scarcity is enforced by the prompt distribution $\mu$. The heavy probability mass concentrated at the uninformative prompt $x_0$ (a proportion of $1 - \Delta$) further dilutes the evidence available to the learner, as most samples drawn will provide no information about any of the hidden labels $j_i$.

Together, these two effects—the need for many samples to resolve ambiguity at each informative prompt and the rarity of encountering such prompts—create the challenging family of models necessary to prove a tight lower bound on sample complexity.

STEP 2: TUNE $\Delta$ VIA AN INLINED REDUCTION-TO-CLASSIFICATION ARGUMENT.

Fix an arbitrary algorithm and let $\widehat{\pi}$ be the model it outputs using at most $n$ samples. For a target failure level $\varepsilon \in (0, 1)$ we set

$$\Delta := 16\,\varepsilon.$$

(Here $\Delta$ is part of the adversarially chosen prompt distribution $\mu$ from Step 1.) We now show that if the algorithm achieved overall success at least $1 - \varepsilon$, then the average misclassification rate on the $d$ informative prompts must be at most $\varepsilon/\Delta = 1/16$.

**Success event and its consequence at a fixed informative prompt.** Recall from Step 1 that for each $i \in [d]$ the maximizer set is a singleton, $y^{\mathcal{I}}(x_i) = \{y_{j_i^\star}\}$, where $j_i^\star$ is the unique index of the favored label at $x_i$. Define the success event at $x$ by

$$\mathcal{S}(x) := \left\{ \widehat{\pi}(y^{\mathcal{I}}(x) \mid x) > \tfrac{1}{2} \right\}.$$

For $i \in [d]$ let $\widehat{j}_i := \arg\max_{j \in [M]} \widehat{\pi}(y_j \mid x_i)$ be the algorithm's predicted label at $x_i$. Because $y^{\mathcal{I}}(x_i) = \{y_{j_i^\star}\}$ and $\widehat{\pi}(y_{j_i^\star} \mid x_i) > \frac{1}{2}$ on $\mathcal{S}(x_i)$, the maximizer must coincide with the true index:

$$\mathcal{S}(x_i) \implies \widehat{j}_i = j_i^\star.$$

Equivalently, using indicator notation,

$$\mathbb{I}\{\widehat{j}_i \neq j_i^\star\} \leq \mathbb{I}\{\neg\mathcal{S}(x_i)\} \quad \Rightarrow \quad \mathbb{P}[\widehat{j}_i \neq j_i^\star] \leq 1 - \mathbb{P}[\mathcal{S}(x_i)].$$

**From overall success under $\mu$ to per-coordinate success.** All probabilities and expectations below are taken over the internal randomness of the algorithm, the data it observes on instance $\pi^{\mathcal{I}}$, and over a uniformly random $\mathcal{I} \in [M]^d$; we abbreviate this by $\mathbb{E}$ and $\mathbb{P}$. By definition of the prompt distribution $\mu = (1 - \Delta)\delta_{x_0} + \frac{\Delta}{d} \sum_{i=1}^d \delta_{x_i}$,

$$\mathbb{E}\,\mathbb{P}_{x \sim \mu}[\mathcal{S}(x)] = (1 - \Delta)\,\mathbb{E}\,\mathbb{P}[\mathcal{S}(x_0)] + \frac{\Delta}{d}\sum_{i=1}^d \mathbb{E}\,\mathbb{P}[\mathcal{S}(x_i)].$$

Assume the algorithm attained overall success at least $1 - \varepsilon$, i.e. $\mathbb{E}\,\mathbb{P}_{x \sim \mu}[\mathcal{S}(x)] \geq 1 - \varepsilon$. Since $\mathbb{P}[\mathcal{S}(x_0)] \leq 1$, we get

$$\frac{\Delta}{d}\sum_{i=1}^d \mathbb{E}\,\mathbb{P}[\mathcal{S}(x_i)] \geq 1 - \varepsilon - (1 - \Delta) = \Delta - \varepsilon,$$

$$\text{hence} \qquad \frac{1}{d}\sum_{i=1}^d \mathbb{E}\,\mathbb{P}[\mathcal{S}(x_i)] \geq 1 - \frac{\varepsilon}{\Delta}.$$

**Average misclassification bound at the informative prompts.** Using the implication above at each $x_i$,

$$\frac{1}{d}\sum_{i=1}^d \mathbb{E}\,\mathbb{P}[\widehat{j}_i \neq j_i^\star] \leq \frac{1}{d}\sum_{i=1}^d \Big(1 - \mathbb{E}\,\mathbb{P}[\mathcal{S}(x_i)]\Big) = 1 - \frac{1}{d}\sum_{i=1}^d \mathbb{E}\,\mathbb{P}[\mathcal{S}(x_i)] \leq \frac{\varepsilon}{\Delta}.$$

With our choice $\Delta = 16\varepsilon$, the right-hand side equals $1/16$. Therefore, if an algorithm with budget $n$ claimed overall success at least $1 - \varepsilon$, then necessarily the average per-coordinate misclassification rate would not exceed $1/16$. In Step 3 we will show, via a complementary counting argument on how many samples actually land on each informative prompt, that any budget $n$ smaller than a certain threshold forces a constant lower bound ($\geq 1/8$) on the left-hand side, whence a contradiction. This yields a necessary relation between $n$ and $\varepsilon$ that we later invert to obtain the risk-form lower bound.

STEP 3: SELECTING $n$ TO ENSURE COMPLIANCE WITH THE POLICY CONDITION NUMBER CONSTRAINT

We need to choose $n$ as a function of $\varepsilon$ such that every model $\pi \in \Pi$ satisfies the policy condition number constraint $\kappa^{(p)}(\pi) \lesssim C$. This constraint ensures that the model's distribution covers the correct label with sufficient probability.

We begin by recalling the definition of the policy condition measure $\kappa^{(p)}(\pi)$:

$$\kappa^{(p)}(\pi) = \left(\mathbb{E}_{x \sim \mu}\left[\frac{1}{\pi(y^\pi(x) \mid x)^p}\right]\right)^{\frac{1}{p}},$$

where $\pi(y^\pi(x) \mid x)$ is the probability that model $\pi$ assigns to the predicted label $y^\pi(x)$ at input $x$. The integral is taken with respect to the distribution $\mu$ over the inputs.

Next, we split this expectation into contributions from two parts: For $x_0$, where $\pi$ always outputs $y_0$, we have:

$$\mathbb{P}_{x_0}[\pi(y_0 \mid x_0) = 1] = 1 \quad \text{and} \quad \mathbb{P}_{x_0}[\pi(y_j \mid x_0) = 0, \, j \neq 0] = 0.$$

Thus, the contribution to the coverage from $x_0$ is simply $(1 - \Delta)$, where $(1 - \Delta)$ is the mass allocated to $x_0$ in the prompt distribution. For each $x_i$ with $i > 0$, we assign a distribution $P_i$ that is a biased version of a uniform distribution. Specifically, we give the label $y_i$ a slightly higher probability compared to the other labels. The distribution $P_i$ is given by:

$$P_i = \frac{1}{(1 - \gamma)M} \delta_{y_i} + \sum_{j \neq i} \frac{1}{M} \left( 1 - \frac{\gamma}{(M - 1)(1 - \gamma)} \right) \delta_{y_j},$$

The mass assigned to label $y_i$ is higher, but the mass assigned to other labels is correspondingly lower.

Thus, the total contribution to the coverage constraint from all points $x_1, \ldots, x_d$ is:

$$\kappa^{(p)}(\pi)^p = (1 - \Delta) + \Delta \cdot \left( M(1 - \gamma) \right)^p.$$

To satisfy the coverage constraint $\kappa^{(p)}(\pi) \lesssim C$, we need to control the magnitude of the second term $\Delta \cdot \left( M(1 - \gamma) \right)^p$. To ensure this is bounded by a constant $C$, we choose $n$ so that:

$$\Delta \cdot \left( M(1 - \gamma) \right)^p \leq C^p.$$

Solving for $n$:

$$M \asymp \left( \frac{C^p}{\Delta} \right)^{1/p} \cdot (1 - \gamma)^{-1}.$$

Substituting $\gamma = \frac{1}{2}$, we get

$$M \asymp \left( \frac{C^p}{\Delta} \right)^{1/p} \cdot 2.$$

Finally, using the relationship $\Delta = 16\varepsilon$, we obtain the desired form:

$$M \asymp 1 + C \Delta^{-1/p} = 1 + C (16\varepsilon)^{-1/p}.$$

This choice of $n$ ensures that the coverage constraint is respected for all models $\pi \in \Pi$.

STEP 4: DERIVATION OF THE SAMPLE COMPLEXITY BOUND.

To ensure that the algorithm has success at least $1 - \varepsilon$, we need to check the sample complexity under which the per-coordinate misclassification probability remains sufficiently small. To do this, we proceed with the following steps:

**(i) Markov's Inequality for per-coordinate error.** Let $n$ denote the total number of samples collected by the algorithm, where each sample is a triplet $(x, y, \log \pi_0(y|x))$.

Let $n_i$ be the random variable representing the number of samples in the collected dataset of size $n$ for which the prompt is $x_i$. According to the prompt distribution $\mu$, the probability of any single sample being drawn for the prompt $x_i$ is $\Delta/d$. By the linearity of expectation, the expected value of $n_i$ is:

$$\mathbb{E}[n_i] = n \cdot \frac{\Delta}{d}.$$

For the non-negative random variable $n_i$, Markov's inequality states that for any $a > 0$, $\mathbb{P}[n_i \geq a] \leq \mathbb{E}[n_i]/a$. By setting $a = 2\mathbb{E}[n_i] = 2n\Delta/d$, we can bound the probability that $n_i$ is large:

$$\mathbb{P}[n_i \geq 2n\Delta/d] \leq \frac{n\Delta/d}{2n\Delta/d} = \frac{1}{2}.$$

By considering the complementary event, we arrive at:

$$\mathbb{P}[n_i \leq 2n\Delta/d] \geq \frac{1}{2}.$$

**(ii) Lower bounding the per-coordinate classification error.** We now derive a lower bound on the per-coordinate classification error, $\mathbb{E}_{\mathcal{I} \sim \text{Unif}} \mathbb{E}^{\mathcal{I}} \left[ \mathbb{I}\{\widehat{j}_i \neq j_i^\star\} \right]$, by showing that a limited sample size $n$ creates unavoidable informational ambiguity.

Let $\mathcal{D}$ be the dataset of $n$ samples. We define an event $\mathcal{E}_i$ where the data is ambiguous for coordinate $i$: this occurs if at least two responses for prompt $x_i$ were never sampled, and no sampled response for $x_i$ was the unique high-probability one. The total classification error is lower-bounded by the error conditioned on this event:

$$\mathbb{E}_{\mathcal{I}\sim\text{Unif}}\mathbb{E}^{\mathcal{I}}\left[\mathbb{I}\{\widehat{j}_i \neq j_i^\star\}\right] \geq \mathbb{E}_{\mathcal{I}\sim\text{Unif}}\mathbb{E}^{\mathcal{I}}\left[\mathbb{I}\{\widehat{j}_i \neq j_i^\star\} \cdot \mathbb{I}\{\mathcal{E}_i\}\right]. \tag{88}$$

Conditioned on the data $\mathcal{D}$ for which $\mathcal{E}_i$ occurs, the posterior distribution over the unsampled labels for $j_i^\star$ is uniform over a set of size at least two. Thus, any guess $\widehat{j}_i$ has a conditional error probability of at least $1/2$. This allows us to simplify the bound:

$$\mathbb{E}_{\mathcal{I}\sim\text{Unif}}\mathbb{E}^{\mathcal{I}}\left[\mathbb{I}\{\widehat{j}_i \neq j_i^\star\}\right] \geq \frac{1}{2}\mathbb{E}_{\mathcal{I}\sim\text{Unif}}\mathbb{P}^{\mathcal{I}}[\mathcal{E}_i]. \tag{89}$$

To bound the probability of $\mathcal{E}_i$, we consider the likely scenario where the number of samples for prompt $x_i$, denoted $n_i$, is small. By Markov's inequality, $\mathbb{P}[n_i \leq 2n\Delta/d] \geq 1/2$. We can therefore write:

$$\frac{1}{2}\mathbb{E}_{\mathcal{I}\sim\text{Unif}}\mathbb{P}^{\mathcal{I}}[\mathcal{E}_i] \geq \frac{1}{2}\mathbb{E}_{\mathcal{I}\sim\text{Unif}}\mathbb{P}^{\mathcal{I}}[\mathcal{E}_i \cap \{n_i \leq 2n\Delta/d\}] \geq \frac{1}{4}\mathbb{E}_{\mathcal{I}\sim\text{Unif}}\mathbb{P}^{\mathcal{I}}[\mathcal{E}_i \mid n_i \leq 2n\Delta/d]. \tag{90}$$

The conditional probability of $\mathcal{E}_i$ is at least the probability that none of the $n_i$ samples correspond to two specific, fixed responses (e.g., $y_{j_i^\star}$ and another $y_{j'}$), which occurs with probability at least $(1 - 3/M)^{n_i}$ for our construction with $\gamma = 1/2$. On the event $\{n_i \leq 2n\Delta/d\}$, this is bounded below by $(1 - 3/M)^{2n\Delta/d}$.

For a sufficiently small constant $c > 0$, this term is at least $1/4$ whenever the total sample size $n$ satisfies the condition $n \leq c \cdot dM/\Delta$. Combining these steps, we conclude that under this condition on $n$:

$$\mathbb{E}_{\mathcal{I}\sim\text{Unif}}\mathbb{E}^{\mathcal{I}}\left[\mathbb{I}\{\widehat{j}_i \neq j_i^\star\}\right] \geq \frac{1}{8}. \tag{91}$$

This establishes a constant lower bound on the per-coordinate classification error when the sample size is insufficient.

**(iv) Sample complexity lower bound.** Now, combining all the steps, we conclude that the total number of samples needed to guarantee that the per-coordinate misclassification error is at least $1/8$ is related to $n$ as follows:

$$n \geq c \cdot \frac{dM}{\Delta} \iff n \gtrsim \frac{d}{\varepsilon}\left(1 + C\varepsilon^{-1/p}\right). \tag{92}$$

Thus, we obtain a *necessary condition* for the algorithm to achieve success at least $1 - \varepsilon$: the sample complexity must be at least as large as $n \gtrsim \frac{d}{\varepsilon}\left(1 + C\varepsilon^{-1/p}\right)$.

STEP 5: ELIMINATE $d$ IN FAVOR OF $\log|\Pi|$ AND OBTAIN A SCALAR INEQUALITY IN $\varepsilon$.

By construction, $\log|\Pi| = d\log M$. Hence $d = \log|\Pi|/\log M$. Substituting $M(\varepsilon) \asymp 1 + C\varepsilon^{-1/p}$ and $\Delta = 16\varepsilon$ into Eq. 92 gives

$$n \gtrsim \frac{\log|\Pi|}{\log\left(1 + C\varepsilon^{-1/p}\right)} \cdot \frac{1 + C\varepsilon^{-1/p}}{\varepsilon}$$

$$\gtrsim \frac{\log|\Pi|}{\varepsilon^{1+1/p}} \cdot \frac{C}{1 + \log\left(C\varepsilon^{-1/p}\right)}.$$

For the important case $p = 1$ this simplifies to the familiar form

$$n \gtrsim \frac{C\log|\Pi|}{\varepsilon^2\left(1 + \log(C/\varepsilon)\right)} = \frac{C\log|\Pi|}{\varepsilon^2\log(eC/\varepsilon)}. \tag{93}$$

Thus, for any fixed $n$, if an algorithm could attain failure $\leq \varepsilon$, then $\varepsilon$ must satisfy Eq. 93.

STEP 6: INVERT THE MONOTONE CONSTRAINT TO GET THE RISK-FORM LOWER BOUND.

Then, we derive the *risk-form lower bound*: for a fixed sample budget of $n$ samples, what is the minimum achievable failure rate $\epsilon(n)$? This provides a more practical perspective on the intrinsic limitations of self-rewarding algorithms.

**(i) Reformulating the lower Bound as a monotonic constraint**

The established lower bound on the sample complexity $n$ is given by:

$$n \gtrsim \frac{C \, \log |\Pi|}{\epsilon^2 \, (1 + \log(C/\epsilon))} = \frac{C \, \log |\Pi|}{\epsilon^2 \, \log(eC/\epsilon)}. \tag{94}$$

To facilitate the analysis, we consolidate the problem-dependent constants into a single term $K$, and define a normalized quantity $B$ that represents the effective information per sample:

$$K := \Theta(C \, \log |\Pi|), \qquad B := \frac{K}{n}. \tag{95}$$

With these definitions, the lower bound inequality can be elegantly rewritten as a constraint on the failure rate $\epsilon$:

$$n \geq \frac{K}{\epsilon^2 \log(eC/\epsilon)} \quad \Longleftrightarrow \quad \epsilon^2 \log\left(\frac{eC}{\epsilon}\right) \geq B. \tag{96}$$

We now define a function $f(\epsilon)$ that captures the left-hand side of this inequality:

$$f(\epsilon) := \epsilon^2 \, \log\left(\frac{eC}{\epsilon}\right), \quad \text{for } \epsilon \in (0, eC). \tag{97}$$

The function $f(\epsilon)$ is strictly increasing for small $\epsilon$ (specifically for $\epsilon < eC/\sqrt{e}$), which is the regime of interest for non-trivial algorithms. This monotonicity ensures that for any given value of $B$, the equation $f(\epsilon) = B$ has a unique solution.

**(ii) The Contrapositive as a Risk-Form Lower Bound**

The logical contrapositive of the original theorem statement provides the foundation for our risk-form bound. The original theorem states that if an algorithm achieves a failure rate of at most $\epsilon$, then its sample size $n$ must be large enough such that $f(\epsilon) \geq K/n$.

The contrapositive is: if an algorithm uses a sample size of at most $n$, then it cannot guarantee a failure rate $\epsilon$ that violates this condition. Therefore, the minimum achievable failure rate for any algorithm using $n$ samples, which we denote as $\epsilon(n)$, is precisely the unique positive solution to the equation:

$$f(\epsilon(n)) = B = \frac{K}{n}. \tag{98}$$

This leads to the risk-form lower bound: for any algorithm using at most $n$ samples, there exists a hard instance in $\Pi$ such that its expected failure rate is at least $\epsilon(n)$.

$$\mathbb{E}_{\mathcal{I} \sim \text{Unif}} \mathbb{E}^{\mathcal{I}} \, \mathbb{P}_{x \sim \mu} \left[ \hat{\pi} \left( y^{\pi^{\mathcal{I}}}(x) \mid x \right) \leq \tfrac{1}{2} \right] \, \geq \, \epsilon(n). \tag{99}$$

The monotonicity of $f(\epsilon)$ ensures that as the sample size $n$ increases, $B$ decreases, and consequently, the minimum failure rate $\epsilon(n)$ also strictly decreases.

**(iii) Explicit Solution via the Lambert W Function**

To obtain a closed-form expression for $\epsilon(n)$, we solve the implicit equation $f(\epsilon) = B$:

$$\epsilon^2 \, \log\left(\frac{eC}{\epsilon}\right) = B. \tag{100}$$

Let us introduce a substitution $s := \log(eC/\epsilon)$, which implies $\epsilon = eC \, e^{-s}$. Substituting this back into the equation yields:

$$\left(eC \cdot e^{-s}\right)^2 \cdot s = B$$
$$(eC)^2 \cdot e^{-2s} \cdot s = B$$
$$s \cdot e^{-2s} = \frac{B}{e^2 C^2}.$$

To isolate $s$, we multiply both sides by $-2$ to match the form $z \cdot e^z$:

$$(-2s) \cdot e^{-2s} = -\frac{2B}{e^2 C^2}.$$

By definition of the Lambert W function, where $W(z)e^{W(z)} = z$, we can solve for $-2s$:

$$-2s = W\left(-\frac{2B}{e^2 C^2}\right).$$

For the solution to be physically meaningful ($s \geq 1/2$, corresponding to $\epsilon \leq eC/\sqrt{e}$), we must choose the $-1$ branch of the Lambert W function, denoted $W_{-1}$. Thus,

$$s = -\frac{1}{2} W_{-1}\left(-\frac{2B}{e^2 C^2}\right).$$

Finally, substituting back $\epsilon = eC\, e^{-s}$ and $B = K/n$, we obtain the exact, closed-form solution for the minimum failure rate:

$$\epsilon(n) = eC \cdot \exp\left(\frac{1}{2}\, W_{-1}\left(-\frac{2K}{n\, e^2 C^2}\right)\right), \quad \text{where } K = \Theta(C \log|\Pi|). \tag{101}$$

**(iv) Asymptotic Approximation for Large Sample Sizes**

While exact, the Lambert W function form is not immediately intuitive. For the practical regime of large sample sizes ($n \to \infty$), the argument of $W_{-1}$, which is $-2K/(ne^2 C^2)$, approaches $0^-$. We can use the well-known asymptotic expansion $W_{-1}(-x) = \ln(x) - \ln(-\ln(x)) + o(1)$ as $x \to 0^+$. This yields a more interpretable approximation:

$$\epsilon(n) = eC \cdot \exp\left(\frac{1}{2}\left[\ln\left(\frac{2K}{ne^2 C^2}\right) - \ln\left(-\ln\left(\frac{2K}{ne^2 C^2}\right)\right) + o(1)\right]\right)$$

$$= \sqrt{\frac{K}{n \cdot \ln\left(\frac{nC^2}{K}\right)}} \cdot (1 + o(1)).$$

Substituting $K \approx C \log|\Pi|$ and simplifying the logarithmic term, we arrive at the convenient and explicit asymptotic lower bound:

$$\epsilon(n) \asymp \sqrt{\frac{C \log|\Pi|}{n \log\left(\frac{nC}{\log|\Pi|}\right)}}. \tag{102}$$

Thus, we get:

$$\mathbb{P}_{x \sim \mu}\left[\pi_1(y_1^*(x) \mid x) \leq \frac{1}{2}\right] \gtrsim \left(\frac{C \log|\Pi|}{n}\left[\log\left(\frac{nC}{\log|\Pi|}\right)\right]^{-1}\right)^{1/2}. \tag{103}$$

This expression clearly shows that the failure rate decreases primarily as $1/\sqrt{n}$, modulated by a slower-growing logarithmic factor. It also explicitly shows how the problem difficulty, determined by coverage $C$ and model complexity $\log|\Pi|$, scales the error.

$\square$

# F    PROOF OF PROPOSITION 2: FROM LOW-CONFIDENCE MODELS TO GREEDY DECODING FAILURE.

We prove the proposition by construction. We explicitly design an autoregressive policy $\pi$ and a prompt $x$ such that the globally optimal sequence $y^*(x)$ has a probability $\pi(y^*(x) \mid x) \leq 1/2$, while guaranteeing that the greedy decoding algorithm is diverted to a suboptimal sequence. The failure is engineered by creating a *local-optima trap* at the first decoding step, from which recovery of the true optimal sequence is impossible.

STEP 1: SETUP OF THE ADVERSARIAL ENVIRONMENT

Let $p^* \in (0, \frac{1}{2}]$ be the target probability for the optimal sequence. Let the vocabulary be $\mathcal{V}$ and the sequence length be $H \geq 2$. We define the unique optimal sequence as $y^*(x) = (a, b, y_3^*, \ldots, y_H^*)$, where $a, b, y_h^* \in \mathcal{V}$. We also introduce a distinct *trap token* $z \in \mathcal{V}$ where $z \neq a$.

STEP 2: CONSTRUCTION OF THE ADVERSARIAL POLICY $\pi$

We define the policy $\pi$ by specifying its conditional probabilities $\pi_h(\cdot|y_{<h}, x)$ for each step $h = 1, \ldots, H$. The construction ensures two properties: (i) $y^*(x)$ is the unique optimal sequence with probability $p^*$, and (ii) the greedy decoder deterministically selects the trap token $z$ at the first step.

STEP 2.1: INDUCING A LOCALLY OPTIMAL TRAP AT THE FIRST STEP

We design the initial probability distribution $\pi_1(\cdot|x)$ to make the trap token $z$ appear more probable than the correct first token $a$. Let $\varepsilon$ be a small positive constant such that $0 < \varepsilon \leq \min\left(\frac{p^*}{2}, 1 - 2p^*\right)$. The second condition, $\varepsilon \leq 1 - 2p^*$, ensures the total probability assigned to $a$ and $z$ does not exceed 1. We define:

$$\pi_1(y_1|x) = \begin{cases} p^* + \varepsilon & \text{if } y_1 = z \\ p^* & \text{if } y_1 = a \\ \frac{1 - 2p^* - \varepsilon}{|\mathcal{V}| - 2} & \text{if } y_1 \in \mathcal{V} \setminus \{a, z\} \end{cases}$$

The greedy decoder's choice at step 1 is $\hat{y}_1 = \arg\max_{v \in \mathcal{V}} \pi_1(v|x)$. Since $\pi_1(z|x) = p^* + \varepsilon > p^* = \pi_1(a|x)$, the decoder will select $\hat{y}_1 = z$. Because the first token is incorrect ($\hat{y}_1 \neq y_1^*$), the final decoded sequence $\hat{y}$ is guaranteed to be different from $y^*(x)$, thus ensuring decoding failure.

STEP 2.2: ENSURING GLOBAL OPTIMALITY OF $y^*(x)$ FOR STEPS $h \geq 2$

To establish that $y^*(x)$ is indeed the unique global optimum, we define the subsequent conditional probabilities to elevate the probability of the true optimal path while suppressing the probability of all competing paths, including the one initiated by the greedy choice $z$.

**Optimal Path:** For any prefix of the optimal sequence, $y_{<h}^* = (y_1^*, \ldots, y_{h-1}^*)$, we set the conditional probability of the next correct token to 1:

$$\pi_h(y_h^*|y_{<h}^*, x) = 1 \quad \text{for all } h = 2, \ldots, H.$$

This deterministic transition ensures that the total probability of the optimal sequence is precisely:

$$\pi(y^*(x) \mid x) = \pi_1(y_1^* \mid x) \cdot \prod_{h=2}^{H} \pi_h(y_h^* \mid y_{<h}^*, x) = p^* \cdot 1^{H-1} = p^*.$$

By our initial choice, $p^* \leq 1/2$, satisfying the proposition's condition.

**Greedy Path:** For any path beginning with the trap token $z$, we must ensure its total probability is strictly less than $p^*$. It is sufficient to control the probability at step $h = 2$. We set the maximum conditional probability following $z$ to be less than what is needed to compete with $y^*(x)$:

$$\max_{v \in \mathcal{V}} \pi_2(v|z, x) \leq \frac{p^* - 2\varepsilon}{p^* + \varepsilon}.$$

This condition is well-defined because our constraint $\varepsilon \leq p^*/2$ ensures the numerator is non-negative. The total probability of the most likely sequence starting with $z$, which we denote $\hat{y}$, is therefore bounded:

$$\pi(\hat{y}|x) = \pi_1(z|x) \cdot \prod_{h=2}^{H} \pi_h(\hat{y}_h|\hat{y}_{<h}, x) \leq \pi_1(z|x) \cdot \max_{v \in \mathcal{V}} \pi_2(v|z, x)$$

$$\leq (p^* + \varepsilon) \cdot \frac{p^* - 2\varepsilon}{p^* + \varepsilon} = p^* - 2\varepsilon < p^*.$$

All other paths: For all other possible sequence prefixes, the remaining probability mass can be distributed (e.g., uniformly) in a way that ensures their total probabilities are also less than $p^*$. This construction confirms that $y^*(x)$ is the unique sequence with the highest probability, $p^*$.

STEP 3: VERIFYING THE DECODING ERROR

The greedy decoder produces a sequence $\hat{y}$ starting with $\hat{y}_1 = z$, while the optimal sequence is $y^*(x)$ starting with $y_1^* = a$. Since $z \neq a$, the Hamming distance $d_H(\hat{y}, y^*(x))$ is deterministically at least 1.

$$d_H(\hat{y}, y^*(x)) \geq 1.$$

This confirms a non-zero error. In fact, by setting the continuation of the greedy path to be maximally different from $y^*(x)$, the Hamming distance can be made arbitrarily large, up to $H$.

STEP 4: IMPLICATIONS AND CONNECTION TO THE PAPER'S FRAMEWORK

This proposition provides a concrete illustration of the risks associated with unaligned policies, which are central to the motivation of our paper. **Theorem 1** establishes a lower bound on the probability of learning a policy where $\pi(y^*(x) \mid x) \leq 1/2$, particularly when the initial model quality is poor (large $\kappa_0$). Our Proposition 2 demonstrates the direct consequence of such a policy at inference time: it can lead to guaranteed and significant errors for standard decoding algorithms like greedy search.

This result bridges the gap between the *learning failure* (producing an unaligned model) and the *inference failure* (incorrect output generation). It underscores the practical necessity of the iterative alignment framework, which is designed to concentrate probability mass on optimal sequences to refine the policy, thereby pushing $\pi(y^*(x) \mid x)$ above the critical $1/2$ threshold and ensuring reliable decoding as per Proposition 2.

# G  PROOF OF THEOREM 5: PERFORMANCE GUARANTEES FOR LINEAR SOFTMAX MODELS

## G.1  LINEAR SOFTMAX PARAMETERIZATION AND SELF-REWARDING TRAINING

**Parameterized Policies.**  In practice, a language model policy $\pi$ is realized by a large neural network (e.g., a Transformer) whose behavior is governed by a set of parameters $\theta \in \Theta \subseteq \mathbb{R}^d$. The vector $\theta$ encompasses all weights and biases of the model. Consequently, we denote the policy as $\pi_\theta$. Optimizing over the policy space $\Pi$ is thus equivalent to finding the optimal parameters $\theta^*$ in the parameter space $\Theta$.

Language models are typically autoregressive, factorizing the probability of a response $y = (y_1, y_2, \ldots, y_H)$ into a product of conditional probabilities:

$$\pi_\theta(y \mid x) = \prod_{i=1}^{H} \pi_\theta(y_i \mid x, y_{<i}). \tag{104}$$

At each generation step $i$, the model produces a vector of logits, $l_\theta(x, y_{<i}) \in \mathbb{R}^{|\mathcal{V}|}$, which is then transformed into a probability distribution over the vocabulary $\mathcal{V}$ via the **softmax** function:

$$\pi_\theta(y_i \mid x, y_{<i}) = \frac{\exp(l_\theta(x, y_{<i})_{y_i})}{\sum_{v \in \mathcal{V}} \exp(l_\theta(x, y_{<i})_v)}, \tag{105}$$

where $l_\theta(\cdot)_{y_i}$ denotes the logit value corresponding to token $y_i$. This mechanism, applying the softmax function to the outputs of the model's final linear layer, is what we refer to as the **linear softmax** parameterization.

ITERATIVE SELF-REWARDING TRAINING

The training procedure is an iterative loop that begins with an initial model $\pi_{\theta_0}$ and progressively refines it. At round $t$, we have the model parameters $\theta_t$, corresponding to the policy $\pi_{\theta_t}$.

**LLM-as-a-Judge Self-Reward.**  We use a self-reward that is computed by the model itself, where the reward for a given response is its log-probability under the current policy $\pi_{\theta_t}$:

$$J_{\text{self}}(y \mid x, \pi_{\theta_t}) = \log \pi_{\theta_t}(y \mid x), \qquad r_t(y \mid x) := J_{\text{self}}(y \mid x, \pi_{\theta_t}). \tag{106}$$

This choice of reward function systematically increases probability mass on higher-score sequences and decreases it on lower-score ones, as judged by the model's own likelihood assignment.

**Data for a Round.** Let $\mathcal{D}_t = \{(x, y, y')\}_{i=1}^n$ be a dataset of $n$ triples sampled by drawing prompts $x \sim \mu$ and then sampling two responses $y, y' \sim \pi_{\theta_t}(\cdot \mid x)$. For each pair $(y, y')$, their relative quality is measured by the difference in self-reward $J_{\text{self}}(y \mid x, \pi_{\theta_t}) - J_{\text{self}}(y' \mid x, \pi_{\theta_t})$. This framing avoids the need for binary preference labels $(y^+, y^-)$ and instead models the continuous reward difference directly.

**DPO-Style Realization (Square-Loss Form).** With the reference policy set to the current policy, $\pi_{\text{ref}} = \pi_{\theta_t}$, the DPO-style regression objective for round $t$ is to find the optimal parameters $\theta_{t+1}$ by minimizing the following loss with respect to $\theta$:

$$\mathcal{L}_t^{\text{DPO}}(\theta) = \frac{1}{n} \sum_{(x,y,y') \in \mathcal{D}_t} \left( \beta \left[ \log \frac{\pi_\theta(y \mid x)}{\pi_{\theta_t}(y \mid x)} - \log \frac{\pi_\theta(y' \mid x)}{\pi_{\theta_t}(y' \mid x)} \right] - \Delta J_t(x, y, y') \right)^2, \quad (107)$$

where $\Delta J_t(x, y, y')$ is the difference in self-reward between the two responses, calculated using the fixed reference model $\pi_{\theta_t}$:

$$\Delta J_t(x, y, y') = J_{\text{self}}(y \mid x, \pi_{\theta_t}) - J_{\text{self}}(y' \mid x, \pi_{\theta_t}) = \log \pi_{\theta_t}(y \mid x) - \log \pi_{\theta_t}(y' \mid x). \quad (108)$$

Minimizing $\mathcal{L}_t^{\text{DPO}}(\theta)$ drives the log-ratio difference of the policy $\pi_\theta$ to match the log-probability difference of the reference policy $\pi_{\theta_t}$, effectively performing a regression on the reward difference. This shifts probability mass toward sequences with higher self-reward scores in a fine-grained manner.

OPERATOR VIEW IN PARAMETER SPACE

This iterative process can be viewed as the sequential application of operators in the parameter space $\Theta$. For round $t$, the optimization process defines an operator $\mathcal{T}_{\beta_t, r_t} : \Theta \to \Theta$ that maps the current parameters $\theta_t$ to the updated parameters $\theta_{t+1}$:

$$\theta_{t+1} = \mathcal{T}_{\beta_t, r_t}(\theta_t) \in \arg\min_{\theta \in \Theta} \mathcal{L}_t^{\text{DPO}}(\theta) \quad \text{with } \pi_{\text{ref}} = \pi_{\theta_t}. \quad (109)$$

After $T$ rounds of training, the final model parameters are the result of composing these operators:

$$\theta_T = (\mathcal{T}_{\beta_{T-1}, r_{T-1}} \circ \cdots \circ \mathcal{T}_{\beta_0, r_0})(\theta_0). \quad (110)$$

Because the reward function $r_t$ is dynamically determined by the policy $\pi_{\theta_t}$ at each round, the entire training loop is self-contained. The model improves its capabilities through iterative generation, self-evaluation, and optimization, without requiring external human feedback.

## G.2 INTRODUCTION AND PROBLEM FORMULATION

### G.2.1 OBJECTIVE

The objective of this section is to adapt a general theoretical proof in Section D, originally designed for a finite policy class $\Pi$, to the specific case of the linear softmax model class, denoted $\Pi_{\varphi, B}$. The core of this adaptation lies in replacing the uniform convergence argument, which relies on the cardinality of the policy class $|\Pi|$, with a more sophisticated technique based on covering numbers. This modification is necessary because the linear softmax model represents a continuous, and therefore infinite, class of policies, rendering the original proof's dependence on $\log(|\Pi|)$ inapplicable. The new derivation will provide a statistical error bound that correctly characterizes the complexity of the linear softmax model in terms of its parametric dimension, $d$.

### G.2.2 RECAP OF THE GENERAL ANALYTICAL FRAMEWORK

Our ultimate objective is to upper bound the probability that the learned policy after $T$ iterations, $\pi_T$, assigns insufficient probability mass to its own optimal response, $y_T^*(x)$. To build towards this goal, we first analyze the initial case ($T = 1$). The established proof provides a general framework for this first step, which we will adapt.

The core strategy of the framework is to connect the probability of policy failure to a performance gap. Specifically, for the initial policy $\pi_1$, the probability of failing to assign sufficient mass to the initial high quality response, $y_0^*(x)$, is bounded by a performance gap term, $J_0(\pi_1^*) - J_0(\pi_1)$:

$$\mathbb{P}_{x \sim \mu} \left[ \pi_1(y_0^*(x) \mid x) \leq 1 - \delta \right] \leq \frac{J_0(\pi_1^*) - J_0(\pi_1)}{\gamma \delta}.$$

The bulk of the proof is then dedicated to bounding this performance gap. A key component in this bound is the **statistical error term**, $\varepsilon_{\text{stat}}$, which captures the uncertainty inherent in learning from a finite dataset of size $n$. The final bound on the performance gap takes the form:

$$J_0(\pi_1^*) - J_0(\pi_1) \lesssim \kappa_0^{1/2} \varepsilon_{\text{stat}} + \beta \log(\kappa_0).$$

In the original general proof, the policy class $\Pi$ was assumed to be finite. The statistical error was consequently derived using Bernstein's inequality and a union bound, leading to a term dependent on the size of the policy class:

$$\varepsilon_{\text{stat}}^2 \lesssim \frac{B_{n,\rho}^2 \log(|\Pi|/\rho)}{n}.$$

Our task is to adapt this critical step. Since the **linear softmax model** constitutes a continuous, infinite class of policies, we must re-derive the statistical error term $\varepsilon_{\text{stat}}$ using tools appropriate for such spaces, namely **covering numbers**. This will replace the dependency on $\log(|\Pi|)$ with a term that reflects the complexity of the linear softmax parameter space.

### G.2.3  THE LINEAR SOFTMAX MODEL

We consider the linear softmax model class $\Pi_{\varphi,B}$, defined as:

**Definition 3** (Linear Softmax Model). Let $d \in \mathbb{N}$ be the feature dimension, and let $\varphi : \mathcal{X} \times \mathcal{Y} \to \mathbb{R}^d$ be a feature map such that $\|\varphi(x,y)\|_2 \leq 1$. Given a radius $B \geq 1$, the linear softmax model class $\Pi_{\varphi,B}$ is defined as:

$$\Pi_{\varphi,B} := \left\{ \pi_\theta : \theta \in \mathbb{R}^d, \|\theta\|_2 \leq B \right\},$$

where the policy $\pi_\theta$ is given by: $\pi_\theta(y \mid x) \propto \exp\left(\langle \varphi(x,y), \theta \rangle\right)$.

This class is parameterized by a vector $\theta$ residing in a $d$-dimensional ball of radius $B$. Since the parameter space is a compact but uncountably infinite subset of $\mathbb{R}^d$, the cardinality $|\Pi_{\varphi,B}|$ is infinite. Consequently, any generalization bound that scales with $\log(|\Pi|)$ is vacuous and must be replaced by a more appropriate measure of model complexity.

### G.2.4  THE TOOL: COVERING NUMBERS AND $\varepsilon$-NETS

To handle infinite function classes, statistical learning theory employs the concept of covering numbers, which measure the "effective size" or richness of the class at a specific resolution $\varepsilon$. This is formalized through the notion of an $\varepsilon$-net adapted to the linear softmax family $\Pi_{\varphi,B}$.

**Definition 4.** ($\varepsilon$-net for $\Pi_{\varphi,B}$): Let $\varepsilon > 0$. A finite subset $\Psi \subseteq \Pi_{\varphi,B}$ is called an $\varepsilon$-net for the linear softmax class $\Pi_{\varphi,B}$ if for every $\pi_\theta \in \Pi_{\varphi,B}$, there exists $\pi_{\theta'} \in \Psi$ such that

$$\max_{x \in \mathcal{X}} \max_{y \in \mathcal{Y}} \log \frac{\pi_{\theta'}(y \mid x)}{\pi_\theta(y \mid x)} \leq \varepsilon.$$

We denote by $\mathcal{N}(\Pi_{\varphi,B}, \varepsilon)$ the size of the smallest such $\varepsilon$-net.

This definition provides a metric on the space of linear softmax policies. The key utility of an $\varepsilon$-net is that it provides a finite discretization of the infinite model class, which allows for the application of union bounds.

**Definition 5.** (Covering numbers) Given a model class $\Pi$ and tolerance $\varepsilon > 0$, the covering number $\mathcal{N}(\Pi, \varepsilon)$ is defined as the cardinality of the smallest $\varepsilon$-net for $\Pi$. That is,

$$\mathcal{N}(\Pi, \varepsilon) := \min \left\{ |\Psi| : \Psi \subseteq \Pi \text{ is an } \varepsilon\text{-net for } \Pi \right\}.$$

The covering number quantifies the complexity of the function class. Establishing that this value is finite (and finding bounds for it, such as for the linear softmax class) is a cornerstone of proving generalization guarantees for learning algorithms.

### G.2.5  LOSS FUNCTION DEFINITION

Recalling Equation 33 from Section D, we have:

$$J_0(\pi_{\theta_1^*}) - J_0(\pi_{\theta_1})$$

$$\lesssim \mathbb{E}_{\pi_{\theta_1^*},\pi_{\theta_0}}\big[\Delta^{r_0^*} - \Delta^{\hat{r}_0}\big] + \mathbb{E}_{\pi_{\theta_1},\pi_{\theta_0}}\big[\Delta^{\hat{r}_0} - \Delta^{r_0^*}\big] + \beta D_{\mathrm{KL}}(\pi_{\theta_1^*}\|\pi_{\theta_0})$$

$$\lesssim (C_{\pi_{\theta_1}}^{1/2} + C_{\pi_{\theta_1^*}}^{1/2})\cdot\Big(\mathbb{E}_{\pi_{\theta_0},\pi_{\theta_0}}\big[|\Delta^{r_0^*} - \Delta^{\hat{r}_0}|^2\mathbb{I}_G\big]\Big)^{1/2} + (C_{\pi_1}^{1/2} + C_{\pi_{\theta_1^*}}^{1/2})\,\Gamma\,\Upsilon^{1/4} + \beta D_{\mathrm{KL}}(\pi_{\theta_1^*}\|\pi_{\theta_0}) \tag{111}$$

where $\Gamma := \log(C_{\pi_{\theta_0}/\pi_{\theta_1};\beta}) + \log(C_{\pi_{\theta_0}/\pi_{\theta_{1,\beta}^*};\beta})$, $\Upsilon := \mathbb{P}(|\Delta^{r_0^*}| > \eta) + \mathbb{P}(|\Delta^{\hat{r}_0}| > \eta)$, and the coupled-expectation shorthand denotes

$$\mathbb{E}_{\pi,\pi'}[\cdot] := \mathbb{E}_{x\sim\mu}\,\mathbb{E}_{y\sim\pi(\cdot|x),\,y'\sim\pi'(\cdot|x)}[\cdot],$$

and the pairwise differences defined as

$$\Delta^{r_0^*}(x,y,y') := \beta\log\frac{\pi_{\theta_{1,\beta}^*}(y\mid x)}{\pi_{\theta_0}(y\mid x)} - \beta\log\frac{\pi_{\theta_{1,\beta}^*}(y'\mid x)}{\pi_{\theta_0}(y'\mid x)}, \tag{112}$$

$$\Delta^{\hat{r}_0}(x,y,y') := \beta\log\frac{\pi_{\theta_1}(y\mid x)}{\pi_0(y\mid x)} - \beta\log\frac{\pi_{\theta_1}(y'\mid x)}{\pi_{\theta_0}(y'\mid x)}. \tag{113}$$

Then, we also define a preference scoring function, $\Delta^\pi$, as the difference between log-likelihood ratios:

$$\Delta^\pi(x,y,y') := \beta\log\frac{\pi(y|x)}{\pi_0(y|x)} - \beta\log\frac{\pi(y'|x)}{\pi_0(y'|x)}.$$

We note that $\Delta^{r_0^*}(x,y,y') = \Delta^{\pi_{\theta_1^*,\beta}}(x,y,y')$. Now, since $\pi_{\theta_{1,\beta}^*} \in \Pi_{\varphi,B}$ and $\pi_{\theta_1}$ is the ERM solution, the empirical squared error of $\pi_{\theta_1}$ must be less than or equal to that of any other policy in $\Pi_{\varphi,B}$, including $\pi_{\theta_{1,\beta}^*}$ itself. This implies:

$$\sum_{(x,y,y')\in\mathcal{D}_0}\big(\Delta^{\pi_{\theta_1}}(x,y,y') - \Delta^{\pi_{\theta_{1,\beta}^*}}(x,y,y')\big)^2 \leq \min_{\pi\in\Pi}\sum_{(x,y,y')\in\mathcal{D}_0}\big(\Delta^\pi - \Delta^{\pi_{\theta_{1,\beta}^*}}\big)^2 \leq 0. \tag{114}$$

For clarity and to ensure this report is self-contained, we restate the key definitions from the original proof framework. For any policy $\pi \in \Pi$, the clipped squared loss for a single data point $i$ is defined as:

$$Z_i(\pi_\theta)$$

$$:= \big(\Delta^{\pi_\theta}(x_i,y_i,y_i') - \Delta^{\pi_{\theta_{1,\beta}^*}}(x_i,y_i,y_i')\big)^2\,\mathbb{I}\Big\{|\Delta^{\pi_\theta}(x_i,y_i,y_i')| \leq B_{n,\rho},\,\big|\Delta^{\pi_{\theta_{1,\beta}^*}}(x_i,y_i,y_i')\big| \leq B_{n,\rho}\Big\}$$

where $B_{n,\rho}$ is a clipping threshold to be determined. The empirical and population losses are, respectively:

$$\hat{L}^n(\pi_\theta) = \frac{1}{n}\sum_{i=1}^n Z_i(\pi_\theta)$$

and

$$L(\pi_\theta) = \mathbb{E}[Z_i(\pi_\theta)]$$

The expectation for $L(\pi_\theta)$ is over $(x,y,y') \sim \mu \otimes \pi_{\theta_0} \otimes \pi_{\theta_0}$. On the clipping event, the loss is bounded:

$$0 \leq Z_i(\pi_\theta) \leq M := (2B_{n,\rho})^2$$

### G.3  UNIFORM CONVERGENCE BOUND FOR THE LINEAR SOFTMAX MODEL

This section presents the core technical contribution: a new derivation of the uniform population bound for the linear softmax model class $\Pi_{\varphi,B}$, replacing the original argument based on Bernstein's inequality and a simple union bound. Our approach leverages the covering number bound for $\Pi_{\varphi,B}$ to handle the infinite nature of the policy class.

### G.3.1 The Discretization Strategy

The standard methodology for establishing uniform convergence for an infinite, continuous function class involves a three-step process based on $\varepsilon$-nets:

1. Construct a finite $\varepsilon$-net: Use the covering number lemma to establish the existence of a finite set $\Psi_\varepsilon \subseteq \Pi_{\varphi,B}$ that serves as a discrete approximation of the entire class.

2. Establish uniform convergence over the net: Apply a concentration inequality (like Bernstein's) and a union bound over the finite elements of the net $\Psi_\varepsilon$. This yields a high-probability bound that holds simultaneously for all policies in the net.

3. Extend the bound to the entire class: Show that if two policies are close with respect to the metric used for the $\varepsilon$-net, their corresponding losses are also close. This allows the bound established for the discrete net to be extended to any policy in the full class $\Pi_{\varphi,B}$, at the cost of a small approximation error.

We now execute this strategy step-by-step.

### G.3.2 Step 1: Bounding the Loss over an $\varepsilon$-Net

We begin by constructing a finite approximation of the policy space $\Pi_{\varphi,B}$. Let $\varepsilon > 0$ be a resolution parameter to be chosen later. Our goal is to construct an $\varepsilon$-net for $\Pi_{\varphi,B}$, which we will denote by $\Psi_\varepsilon$, with respect to the metric

$$d(\pi_\theta, \pi_{\theta'}) = \max_{x,y} \left| \log \frac{\pi_{\theta'}(y \mid x)}{\pi_\theta(y \mid x)} \right|.$$

An $\varepsilon$-net is a finite set $\Psi_\varepsilon \subset \Pi_{\varphi,B}$ such that for any policy $\pi_\theta \in \Pi_{\varphi,B}$, there exists a policy $\pi_{\theta'} \in \Psi_\varepsilon$ with $d(\pi_\theta, \pi_{\theta'}) \leq \varepsilon$. We will explicitly construct such a net and bound its cardinality.

The construction proceeds in two stages. First, we build a net over the $d$-dimensional parameter space $\Theta_B = \{\theta \in \mathbb{R}^d : \|\theta\|_2 \leq B\}$. Second, we show that this parameter-space net induces the desired $\varepsilon$-net in the policy space.

**1. Constructing a Net in the Parameter Space.** We need to find a finite set of points $\{\theta_1^1, \ldots, \theta_1^N\}$ in $\Theta_B$ that forms a net with a specific radius, let's say $r > 0$. By a standard volumetric argument, we can bound the size $N$ of such a net. Consider a maximal set of points $\{\theta_1^i\}_{i=1}^N$ in $\Theta_B$ that are separated by at least $r$, i.e., $\|\theta_1^i - \theta_1^j\|_2 > r$ for all $i \neq j$. The open balls of radius $r/2$ centered at these points, $B(\theta_i, r/2)$, are disjoint. Furthermore, all these balls are contained within a larger ball of radius $B + r/2$. By comparing the total volume of the small balls to the volume of the large ball, we have:

$$N \cdot \text{Vol}(B(0, r/2)) \leq \text{Vol}(B(0, B + r/2)).$$

Since the volume of a $d$-dimensional ball of radius $R$ is proportional to $R^d$, this implies:

$$N \cdot (r/2)^d \leq (B + r/2)^d \implies N \leq \left( \frac{B + r/2}{r/2} \right)^d = \left( \frac{2B}{r} + 1 \right)^d.$$

This maximal packing is also an $r$-net. We will see that the required radius for our parameter net is $r = \varepsilon/2$. Substituting this into the bound gives a net size of at most $(4B/\varepsilon + 1)^d$. Standard results in high-dimensional geometry provide various bounds of this nature; for consistency with the analysis this proof is based on, we adopt the slightly looser but common bound of the form $(C \cdot B/\varepsilon)^d$. Let us choose a net for $\Theta_B$ with radius $r = \varepsilon/2$, which we denote $\mathcal{N}_{\varepsilon/2}$, satisfying the size bound:

$$|\mathcal{N}_{\varepsilon/2}| \leq \left( \frac{8B}{\varepsilon} \right)^d.$$

**2. Mapping to a Net in the Policy Space.** Let $\Psi_\varepsilon = \{\pi_\theta : \theta \in \mathcal{N}_{\varepsilon/2}\}$ be the set of policies corresponding to our parameter net. Now, for any policy $\pi_\theta \in \Pi_{\varphi,B}$, by construction of $\mathcal{N}_{\varepsilon/2}$, there exists a parameter vector $\theta^i \in \mathcal{N}_{\varepsilon/2}$ such that $\|\theta - \theta^i\|_2 \leq \varepsilon/2$. We now show that $d(\pi_\theta, \pi_{\theta^i}) \leq \varepsilon$.

Recall that the policy is defined as $\pi_\theta(y \mid x) = \exp(\langle \varphi(x, y), \theta \rangle)/A_\theta(x)$, where $A_\theta(x) = \sum_{y'} \exp(\langle \varphi(x, y'), \theta \rangle)$ is the normalization constant. The log-ratio of two policies is:

$$\log\left(\frac{\pi_\theta(y \mid x)}{\pi_{\theta^i}(y \mid x)}\right) = \log\left(\frac{\exp(\langle \varphi(x, y), \theta \rangle)/A_\theta(x)}{\exp(\langle \varphi(x, y), \theta^i \rangle)/A_{\theta^i}(x)}\right)$$

$$= \langle \varphi(x, y), \theta - \theta^i \rangle - \log\left(\frac{A_\theta(x)}{A_{\theta^i}(x)}\right).$$

We bound the magnitude of each term. For the first term, we assume a standard condition that the feature vectors are bounded, i.e., $\|\varphi(x, y)\|_2 \leq 1$ for all $x, y$. Using the Cauchy-Schwarz inequality:

$$|\langle \varphi(x, y), \theta - \theta^i \rangle| \leq \|\varphi(x, y)\|_2 \|\theta - \theta^i\|_2 \leq 1 \cdot \frac{\varepsilon}{2} = \frac{\varepsilon}{2}.$$

For the second term, we analyze the ratio of the normalization constants:

$$\frac{A_\theta(x)}{A_{\theta^i}(x)} = \frac{\sum_{y'} \exp(\langle \varphi(x, y'), \theta \rangle)}{\sum_{y'} \exp(\langle \varphi(x, y'), \theta^i \rangle)} = \frac{\sum_{y'} \exp(\langle \varphi(x, y'), \theta^i \rangle) \exp(\langle \varphi(x, y'), \theta - \theta^i \rangle)}{\sum_{y'} \exp(\langle \varphi(x, y'), \theta^i \rangle)}.$$

This is a weighted average of the term $\exp(\langle \varphi(x, y'), \theta - \theta^i \rangle)$. From our bound on the first term, we know $-\varepsilon/2 \leq \langle \varphi(x, y'), \theta - \theta^i \rangle \leq \varepsilon/2$. Therefore, the exponential term is bounded as $\exp(-\varepsilon/2) \leq \exp(\langle \varphi(x, y'), \theta - \theta^i \rangle) \leq \exp(\varepsilon/2)$. Since the ratio $Z_\theta(x)/Z_{\theta_i}(x)$ is a weighted average of values within this range, it must also lie within the same range:

$$e^{-\varepsilon/2} \leq \frac{A_\theta(x)}{A_{\theta^i}(x)} \leq e^{\varepsilon/2}.$$

Taking the logarithm gives $\left|\log\left(\frac{Z_\theta(x)}{Z_{\theta_i}(x)}\right)\right| \leq \frac{\varepsilon}{2}$.

Finally, by the triangle inequality, we combine the bounds on the two terms:

$$\left|\log\left(\frac{\pi_\theta(y \mid x)}{\pi_{\theta^i}(y \mid x)}\right)\right| \leq |\langle \varphi(x, y), \theta - \theta^i \rangle| + \left|\log\left(\frac{A_\theta(x)}{A_{\theta^i}(x)}\right)\right| \leq \frac{\varepsilon}{2} + \frac{\varepsilon}{2} = \varepsilon.$$

Since this holds for any state-action pair $(x, y)$, we have $d(\pi_\theta, \pi_{\theta^i}) = \max_{x,y} |\log(\pi_\theta(y \mid x)/\pi_{\theta^i}(y \mid x))| \leq \varepsilon$. We have thus shown that $\Psi_\varepsilon$ is a valid $\varepsilon$-net for $\Pi_{\varphi,B}$, and its size is bounded by:

$$|\Psi_\varepsilon| \leq \left(\frac{8B}{\varepsilon}\right)^d.$$

Our next objective is to derive a high-probability performance guarantee that holds uniformly over a finite set of policies $\Psi_\varepsilon$. The derivation begins with a concentration inequality for a single, fixed policy and then extends it to the entire set using a union bound.

**3. Performance Bound for a Single Policy.** For any *fixed* policy $\pi_\theta$, the associated losses $\{Z_i(\pi_\theta)\}_{i=1}^n$ form a sequence of independent and identically distributed (i.i.d.) random variables. By construction, these losses are bounded within the interval $[0, M]$. To relate the true expected loss $L(\pi_\theta) = \mathbb{E}[Z_i(\pi_\theta)]$ to its empirical estimate $\hat{L}_n(\pi_\theta) = \frac{1}{n}\sum_{i=1}^n Z_i(\pi_\theta)$, we can invoke Bernstein's inequality. For any $t > 0$, Bernstein's inequality gives:

$$\Pr\left(L(\pi_\theta) - \hat{L}_n(\pi_\theta) \geq t\right) \leq \exp\left(-\frac{nt^2}{2\operatorname{Var}(Z_i(\pi_\theta)) + \frac{2}{3}Mt}\right).$$

A key challenge is that the variance term, $\operatorname{Var}(Z_i(\pi_\theta))$, is unknown. However, for non-negative random variables bounded by $M$, the variance can be bounded by the mean: $\operatorname{Var}(Z_i(\pi_\theta)) \leq \mathbb{E}[Z_i(\pi_\theta)^2] \leq M \cdot \mathbb{E}[Z_i(\pi_\theta)] = ML(\pi_\theta)$. Substituting this into the inequality introduces $L(\pi_\theta)$ on both sides, creating a recursive relationship.

Solving this inequality for $L(\pi_\theta)$ in terms of $\hat{L}_n(\pi_\theta)$—a process often referred to as a fixed-point calculation—yields a more practical, empirical version of Bernstein's bound. Specifically, for a chosen failure probability $\delta \in (0, 1)$, we can state the following: with probability at least $1 - \delta$,

$$L(\pi_\theta) \leq 2\hat{L}_n(\pi_\theta) + c\frac{M\log(1/\delta)}{n}, \tag{115}$$

for some universal constant $c > 0$. This powerful result bounds the true loss for a *single, predetermined* policy.

**4. Uniform Bound via the Union Bound**   The bound in Eq. Eq. 115 is insufficient for our needs, as we require a guarantee that holds *simultaneously* for all policies $\pi_{\theta'} \in \Psi_\varepsilon$, not just one chosen in advance. To achieve this uniform convergence, we apply the union bound.

Let $\mathcal{E}_{\pi_{\theta'}}$ be the "bad" event where the bound in Eq. Eq. 115 fails for a specific policy $\pi_{\theta'}$. We want to bound the probability that at least one of these bad events occurs for any policy in the finite net $\Psi_\varepsilon$, i.e., $\Pr(\cup_{\pi_{\theta'} \in \Psi_\varepsilon} \mathcal{E}_{\pi_{\theta'}})$. The union bound states that this probability is at most the sum of the individual event probabilities:

$$\Pr \left( \bigcup_{\pi_{\theta'} \in \Psi_\varepsilon} \mathcal{E}_{\pi_{\theta'}} \right) \leq \sum_{\pi_{\theta'} \in \Psi_\varepsilon} \Pr(\mathcal{E}_{\pi_{\theta'}}).$$

To ensure the total probability of failure is no more than a desired level $\rho$, we can enforce a much stricter failure probability for each individual policy. We set the individual failure probability $\delta$ for each $\pi_{\theta'}$ to be $\delta = \frac{\rho}{|\Psi_\varepsilon|}$.

By this construction, the total probability of at least one bound failing is at most:

$$\sum_{\pi_{\theta'} \in \Psi_\varepsilon} \frac{\rho}{|\Psi_\varepsilon|} = |\Psi_\varepsilon| \cdot \frac{\rho}{|\Psi_\varepsilon|} = \rho.$$

Consequently, with a total probability of at least $1 - \rho$, the bound holds for *all* policies in $\Psi_\varepsilon$ simultaneously.

**5. Final Uniform Bound**   We now substitute our new choice of $\delta = \frac{\rho}{|\Psi_\varepsilon|}$ into the single-policy bound from Eq. Eq. 115. This yields the uniform inequality: with probability at least $1 - \rho$, for all $\pi_{\theta'} \in \Psi_\varepsilon$:

$$L(\pi_{\theta'}) \leq 2\hat{L}_n(\pi_{\theta'}) + c\frac{M}{n} \log \left( \frac{1}{\rho/|\Psi_\varepsilon|} \right) = 2\hat{L}_n(\pi_{\theta'}) + c\frac{M}{n} \log \left( \frac{|\Psi_\varepsilon|}{\rho} \right).$$

The term $\log(|\Psi_\varepsilon|)$ quantifies the complexity cost of generalizing from a single policy to the entire net. A larger net requires a stronger guarantee for each policy, which widens the overall bound.

Finally, by substituting the known bound on the cardinality of the $\varepsilon$-net, $|\Psi_\varepsilon|$, which depends on the dimensionality $d$ of the policy class and the desired resolution $\varepsilon$, we arrive at the final expression:

$$L(\pi_{\theta'}) \leq 2\hat{L}_n(\pi_{\theta'}) + c\frac{M}{n} \left( d \log \left( \frac{8B}{\varepsilon} \right) + \log \left( \frac{1}{\rho} \right) \right).$$

This inequality provides a concrete, uniform performance guarantee over the discrete skeleton $\Psi_\varepsilon$ of our policy class. The subsequent analytical step is to extend this guarantee from the discrete net to the entire continuous policy class $\Pi_{\varphi,B}$, ensuring our conclusions apply to any policy we might select. The next step is to extend this guarantee to the entire continuous class $\Pi_{\varphi,B}$.

### G.3.3   STEP 2: BOUNDING THE APPROXIMATION ERROR

To extend the bound from the net $\Psi_\varepsilon$ to the full class $\Pi_{\varphi,B}$, we must demonstrate that the loss function $Z_i(\pi_\theta)$ exhibits a form of local stability or continuity. Specifically, if two policies $\pi_\theta$ and $\pi_{\theta'}$ are close in the log-ratio metric, their corresponding losses must also be close. This property ensures that the bound on a net element $\pi_{\theta'}$ provides a good approximation for the bound on any policy $\pi_\theta$ it covers. We formalize this in the following lemma.

**Lemma 11.** *(Lipschitz-like Property of the Loss Function): Let $\pi_\theta, \pi_{\theta'} \in \Pi_{\varphi,B}$ with $\pi_{\theta'} \in \Psi_\varepsilon$. Suppose that on the clipping event $\{|\Delta^{\pi_\theta}| \leq B_{n,\rho}, |\Delta^{\pi_{\theta_{1,\beta}^*}}| \leq B_{n,\rho}, |\Delta^{\pi_{\theta'}}| \leq B_{n,\rho}\}$, we have that*

$$\max_{x,y} \left| \log \left( \frac{\pi_\theta(y \mid x)}{\pi_{\theta'}(y \mid x)} \right) \right| \leq \varepsilon.$$

*Then, the difference in the clipped squared loss is bounded by:*

$$|Z_i(\pi_\theta) - Z_i(\pi_{\theta'})| \leq 8\beta B_{n,\rho}\varepsilon.$$

*Proof of Lemma 11.* By definition, $Z_i(\pi_\theta) = \left(\Delta^{\pi_\theta}(x_i, y_i, y_i') - \Delta^{\pi_{\theta_{1,\beta}^*}}(x_i, y_i, y_i')\right)^2$ on the clipping event. Let $D(\pi_\theta) = \Delta^{\pi_\theta} - \Delta^{\pi_{\theta_{1,\beta}^*}}$. We want to bound $\left|D(\pi_\theta)^2 - D(\pi_{\theta'})^2\right|$. Using the identity $a^2 - b^2 = (a-b)(a+b)$, we have:

$$\left|D(\pi_\theta)^2 - D(\pi_{\theta'})^2\right| = \left|(D(\pi_\theta) - D(\pi_{\theta'}))(D(\pi_\theta) + D(\pi_{\theta'}))\right|.$$

The first term is:

$$\left|D(\pi_\theta) - D(\pi_{\theta'})\right| = \left|(\Delta^{\pi_\theta} - \Delta^{\pi_{\theta_{1,\beta}^*}}) - (\Delta^{\pi_{\theta'}} - \Delta^{\pi_{\theta_{1,\beta}^*}})\right| = \left|\Delta^{\pi_\theta} - \Delta^{\pi_{\theta'}}\right|.$$

Let us analyze this difference:

$$\Delta^{\pi_\theta}(x, y, y') - \Delta^{\pi_{\theta'}}(x, y, y')$$
$$= \left(\beta \log \frac{\pi_\theta(y \mid x)}{\pi_{\theta_0}(y \mid x)} - \beta \log \frac{\pi_\theta(y' \mid x)}{\pi_{\theta_0}(y' \mid x)}\right) - \left(\beta \log \frac{\pi_{\theta'}(y \mid x)}{\pi_{\theta_0}(y \mid x)} - \beta \log \frac{\pi_{\theta'}(y' \mid x)}{\pi_{\theta_0}(y' \mid x)}\right)$$
$$= \beta \left(\log \frac{\pi_\theta(y \mid x)}{\pi_{\theta'}(y \mid x)} - \log \frac{\pi_\theta(y' \mid x)}{\pi_{\theta'}(y' \mid x)}\right). \tag{116}$$

By the triangle inequality and the $\varepsilon$-net property, the magnitude is bounded by:

$$\left|\Delta^{\pi_\theta} - \Delta^{\pi_{\theta'}}\right| \leq \beta \left(\left|\log \frac{\pi_\theta(y \mid x)}{\pi_{\theta'}(y \mid x)}\right| + \left|\log \frac{\pi_\theta(y' \mid x)}{\pi_{\theta'}(y' \mid x)}\right|\right) \leq \beta(\varepsilon + \varepsilon) = 2\beta\varepsilon.$$

The second term is:

$$\left|D(\pi_\theta) + D(\pi_{\theta'})\right| = \left|\Delta^{\pi_\theta} - \Delta^{\pi_{\theta_{1,\beta}^*}} + \Delta^{\pi_{\theta'}} - \Delta^{\pi_{\theta_{1,\beta}^*}}\right|.$$

On the clipping event, we have $|\Delta^{\pi_\theta}| \leq B_{n,\rho}$, $|\Delta^{\pi_{\theta'}}| \leq B_{n,\rho}$, and $|\Delta^{\pi_{\theta_{1,\beta}^*}}| \leq B_{n,\rho}$. By the triangle inequality:

$$\left|D(\pi_\theta) + D(\pi_{\theta'})\right| \leq |\Delta^{\pi_\theta}| + |\Delta^{\pi_{\theta'}}| + 2|\Delta^{\pi_{1,\beta}^*}| \leq B_{n,\rho} + B_{n,\rho} + 2B_{n,\rho} = 4B_{n,\rho}.$$

Combining these two bounds, we get:

$$|Z_i(\pi_\theta) - Z_i(\pi_{\theta'})| = |D(\pi_\theta)^2 - D(\pi_{\theta'})^2| \leq (2\beta\varepsilon)(4B_{n,\rho}) = 8\beta B_{n,\rho}\varepsilon.$$

This completes the proof of the lemma. $\square$ $\qquad\qquad\qquad\qquad\qquad\qquad\qquad\qquad\qquad\qquad\qquad\square$

**Implication of Lemma 11** This lemma allows us to control the variation of both the population loss $L(\pi_\theta)$ and the empirical loss $\hat{L}^n(\pi_\theta)$ as we move away from the points in the net. Specifically, for any $\pi_\theta \in \Pi_{\varphi,B}$ and its closest neighbor $\pi_{\theta'} \in \Psi_\varepsilon$:

$$|L(\pi_\theta) - L(\pi_{\theta'})| = |\mathbb{E}[Z_i(\pi_\theta) - Z_i(\pi_{\theta'})]| \leq \mathbb{E}[|Z_i(\pi_\theta) - Z_i(\pi_{\theta'})|] \leq 8\beta B_{n,\rho}\varepsilon.$$

$$\left|\hat{L}^n(\pi_\theta) - \hat{L}^n(\pi_{\theta'})\right| = \frac{1}{n}\sum_{i=1}^{n}|Z_i(\pi_\theta) - Z_i(\pi_{\theta'})| \leq \frac{1}{n}\sum_{i=1}^{n} 8\beta B_{n,\rho}\varepsilon.$$

### G.3.4 STEP 3: COMBINING THE BOUNDS AND OPTIMIZING

We can now assemble the pieces to derive a uniform bound over the entire class $\Pi_{\varphi,B}$. For any $\pi_\theta \in \Pi_{\varphi,B}$, let $\pi_{\theta'} \in \Psi_\varepsilon$ be its covering element, i.e., the policy in the net such that $d(\pi_\theta, \pi_{\theta'}) \leq \varepsilon$. Using the triangle inequality and the results from the previous steps:

$$L(\pi_\theta) \leq L(\pi_{\theta'}) + |L(\pi_\theta) - L(\pi_{\theta'})|$$
$$\leq \left(2\hat{L}_n(\pi_{\theta'}) + c\frac{M}{n}\left(d\log\left(\frac{8B}{\varepsilon}\right) + \log\left(\frac{1}{\rho}\right)\right)\right) + 8\beta B_{n,\rho}\varepsilon$$
$$\leq 2\hat{L}_n(\pi_\theta) + 8\beta B_{n,\rho}\varepsilon + c\frac{M}{n}\left(d\log\left(\frac{8B}{\varepsilon}\right) + \log\left(\frac{1}{\rho}\right)\right)$$
$$\leq 2\hat{L}_n(\pi_\theta) + 8\beta B_{n,\rho}\varepsilon + c\frac{M}{n}\left(d\log\left(\frac{8B}{\varepsilon}\right) + \log\left(\frac{1}{\rho}\right)\right).$$

This inequality holds with probability at least $1 - \rho$ for all $\pi_\theta \in \Pi_{\varphi,B}$ simultaneously. The right-hand side contains an error term that depends on our choice of the resolution $\varepsilon$. This term consists of an approximation error $8\beta B_{n,\rho}\varepsilon$, which decreases as $\varepsilon \to 0$, and a statistical error (the term with $M/n$), which increases as $\varepsilon \to 0$ due to the $\log(1/\varepsilon)$ factor. To obtain the tightest possible bound, we must choose $\varepsilon$ to balance this trade-off.

Let's simplify the error term by substituting $M = (2B_{n,\rho})^2 = 4B_{n,\rho}^2$ and ignoring constants:

$$\text{Error}(\varepsilon) \asymp \beta B_{n,\rho}\varepsilon + \frac{B_{n,\rho}^2}{n}\left(d\log(\frac{B}{\varepsilon})\right).$$

A standard approach to minimize such an expression is to set the two terms to be of the same order of magnitude:

$$\beta B_{n,\rho}\varepsilon \asymp \frac{B_{n,\rho}^2}{n}\left(d\log(\frac{B}{\varepsilon})\right) \quad \Rightarrow \quad \varepsilon \asymp \frac{B_{n,\rho}}{n\beta}\left(d\log(\frac{B}{\varepsilon})\right).$$

Ignoring the logarithmic dependency on $\varepsilon$ for a first-order approximation, a near-optimal choice in learning theory for balancing an error of the form $\varepsilon + \frac{d}{n}\log(1/\varepsilon)$ is achieved by

$$\varepsilon \asymp \frac{d\log n}{n}.$$

When this is done, both terms become of the order $\mathcal{O}(d\log n/n)$. The total error is dominated by this rate. Therefore, after optimizing for $\varepsilon$, the uniform convergence bound takes the form:

$$\forall \pi_\theta \in \Pi_{\varphi,B} : L(\pi_\theta) \lesssim 2\hat{L}^n(\pi_\theta) + \frac{1}{n}\cdot B_{n,\rho}^2\left(d\log\frac{nB}{d} + \log\left(\frac{1}{\rho}\right)\right).$$

### G.4 THE REFINED STATISTICAL ERROR AND INTEGRATION INTO THE MAIN PROOF

#### G.4.1 DERIVATION OF THE NEW $\varepsilon_{\text{stat}}^2$

We now apply the uniform bound derived above to the empirical risk minimizer (ERM) policy, $\pi_{\theta_1}$. The ERM property from the original proof states that $\hat{L}^n(\pi_{\theta_1}) = 0$ (Eq. 36). Substituting $\pi = \pi_{\theta_1}$ and $\hat{L}^n(\pi_{\theta_1}) = 0$ into our new uniform bound yields:

$$L(\pi_{\theta_1}) \lesssim \frac{1}{n}\cdot B_{n,\rho}^2\left(d\log\left(\frac{nB}{d}\right) + \log\left(\frac{1}{\rho}\right)\right),$$

This bound on $L(\pi_{\theta_1})$ is precisely the population-level squared error on the clipped region. We can therefore define our new statistical error term, $\varepsilon_{\text{stat}}^2$, as:

$$\varepsilon_{\text{stat}}^2 := \frac{1}{n}\cdot B_{n,\rho}^2\left(d\log\left(\frac{nB}{d}\right) + \log\left(\frac{1}{\rho}\right)\right).$$

This result replaces the original Eq. 40. The key difference is the replacement of the complexity term $\log(|\Pi|)$ with $d\log\left(\frac{nB}{d}\right)$, which correctly captures the dependence on the parametric dimension $d$ of the linear softmax model.

It is also necessary to update the choice of the clipping parameter $B_{n,\rho}$. The original choice was motivated by controlling tail probabilities over a union bound of size $n|\Pi|$. The analogous choice in our setting is to control tails over $n$ samples and the $|\Psi_\varepsilon|$ policies in the net. A suitable choice is:

$$B_{n,\rho} := \log\left(2nC_{\text{loss}}|\Psi_\varepsilon|\rho^{-1}\right) = \log\left(2nC_{\text{loss}}\left(\frac{8B}{\varepsilon}\right)^d\rho^{-1}\right).$$

With an optimal $\varepsilon \asymp \frac{d\log n}{n}$, this becomes

$$B_{n,\rho} \asymp \log\left(nC_{\text{loss}}\left(\frac{nB}{d\log n}\right)^d\rho^{-1}\right) \asymp d\log\left(\frac{nB}{d}\right).$$

This shows that $B_{n,\rho}$ scales polynomially with $d$ and logarithmically with $n$, which is consistent with the overall structure of the bound. For simplicity in the final expression, we can absorb these logarithmic factors into $B_{n,\rho}$ and write the rate as $\frac{1}{n}B_{n,\rho}^2 d$.

### G.4.2 INTEGRATION INTO THE GLOBAL PROOF STRUCTURE

With the newly derived statistical error term, we can now return to the main flow of the proof. The inequality 33 provides a bound on the performance gap $J_0(\pi_{\theta_1^*}) - J_0(\pi_{\theta_1})$ in terms of the population squared error:

$$J_0(\pi_{\theta_1^*}) - J_0(\pi_{\theta_1})$$

$$\lesssim \mathbb{E}_{\pi_{\theta_1^*}, \pi_{\theta_0}}\big[\Delta^{r_0^*} - \Delta^{\hat{r}_0}\big] + \mathbb{E}_{\pi_{\theta_1}, \pi_{\theta_0}}\big[\Delta^{\hat{r}_0} - \Delta^{r_0^*}\big] + \beta D_{\mathrm{KL}}(\pi_{\theta_1^*} \| \pi_{\theta_0})$$

$$\lesssim (C_{\pi_{\theta_1}}^{1/2} + C_{\pi_{\theta_1^*}}^{1/2}) \cdot \Big(\mathbb{E}_{\pi_{\theta_0}, \pi_{\theta_0}}\big[|\Delta^{r_0^*} - \Delta^{\hat{r}_0}|^2 \mathbb{I}_G\big]\Big)^{1/2} + (C_{\pi_{\theta_1}}^{1/2} + C_{\pi_{\theta_1^*}}^{1/2})\, \Upsilon\, \Theta^{1/4} + \beta D_{\mathrm{KL}}(\pi_{\theta_1^*} \| \pi_{\theta_0})$$

$$(117)$$

where $\Gamma := \log(C_{\pi_{\theta_0}/\pi_{\theta_1}; \beta}) + \log(C_{\pi_{\theta_0}/\pi_{\theta_{1,\beta}^*}; \beta})$, $\Upsilon := \mathbb{P}(|\Delta^{r_0^*}| > \eta) + \mathbb{P}(|\Delta^{\hat{r}_0}| > \eta)$, The term under the square root is precisely bounded by what we defined as $\varepsilon_{\mathrm{stat}}^2$. Substituting this new bound yields:

$$J_0(\pi_{\theta_1^*}) - J_0(\pi_{\theta_1}) \lesssim (C_{\pi_{\theta_1^*}}^{1/2} + C_{\pi_{\theta_1}}^{1/2})\varepsilon_{\mathrm{stat}} + (C_{\pi_{\theta_1^*}}^{1/2} + C_{\pi_{\theta_1}}^{1/2})\log(C_{\mathrm{loss}})\rho^{1/4} + \beta D_{\mathrm{KL}}(\pi_{\theta_1^*} \| \pi_{\theta_0}).$$

The remainder of the proof proceeds exactly as in the original version, using this updated definition of $\varepsilon_{\mathrm{stat}}$. After bounding the concentrability coefficients and KL divergence term, and balancing the error terms, we arrive at the clean form:

$$J_0(\pi_{\theta_1^*}) - J_0(\pi_{\theta_1}) \lesssim \kappa_0^{1/2}\varepsilon_{\mathrm{stat}} + \beta \log(\kappa_0).$$

Finally, we substitute this refined bound on the performance gap back into the initial inequality that connects it to the probability of failure. This yields the final, adapted result for the linear softmax model:

$$\mathbb{P}_{x \sim \mu}\left[\pi_{\theta_1}(y_0^*(x) \mid x) \leq 1 - \delta\right] \leq \frac{J_0(\pi_{\theta_1^*}) - J_0(\pi_{\theta_1})}{\gamma\delta}$$

$$\lesssim \frac{1}{\gamma\delta}\left(\kappa_0^{1/2}\varepsilon_{\mathrm{stat}} + \beta\log(\kappa_0)\right)$$

$$\lesssim \frac{1}{\gamma\delta}\left(\kappa_0^{1/2} B_{n,\rho}\sqrt{\frac{d\log\left(\frac{nB}{d}\right) + \log\left(\frac{1}{\rho}\right)}{n}} + \beta\log(\kappa_0)\right).$$

$$\lesssim \frac{1}{\gamma\delta}\left(\kappa_0^{1/2} \cdot \frac{1}{\sqrt{n}}\left(d\log(\frac{nB}{d\rho})\right)^{\frac{3}{2}} + \beta\log(\kappa_0)\right) \quad (118)$$

By definition of the round-1 policy maximizer, we denote

$$y_1^*(x) \in \arg\max_{y \in \mathcal{Y}} \pi_{\theta_1}(y \mid x). \quad (119)$$

It immediately follows that the maximizer $y_1^*(x)$ dominates the baseline target $y_0^*(x)$ pointwise, in the sense that

$$\pi_{\theta_1}(y_1^*(x) \mid x) \geq \pi_{\theta_1}(y_0^*(x) \mid x), \quad \forall x \in \mathcal{X}.$$

Consequently, whenever $\pi_{\theta_1}(y_1^*(x) \mid x) \leq 1 - \delta$, it must also hold that $\pi_{\theta_1}(y_0^*(x) \mid x) \leq 1 - \delta$. Equivalently,

$$\left\{x : \pi_{\theta_1}(y_1^*(x) \mid x) \leq 1 - \delta\right\} \subseteq \left\{x : \pi_{\theta_1}(y_0^*(x) \mid x) \leq 1 - \delta\right\}. \quad (120)$$

Taking probabilities under $x \sim \mu$ and invoking the previously established bound for the right-hand event, we obtain

$$\mathbb{P}_{x \sim \mu}\big[\pi_{\theta_1}\big(y_1^*(x) \mid x\big) \leq 1 - \delta\big] \leq \mathbb{P}_{x \sim \mu}\big[\pi_{\theta_1}\big(y_0^*(x) \mid x\big) \leq 1 - \delta\big]$$

$$\lesssim \frac{1}{\gamma\delta}\left(\kappa_0^{1/2} \cdot \frac{1}{\sqrt{n}}\left(d\log\frac{nB}{d\rho}\right)^{3/2} + \beta\log\kappa_0\right). \quad (121)$$

This expression replaces the original Equation 45. It successfully adapts the general proof architecture to the linear softmax model by incorporating a complexity measure, $d \log(n)$, derived from covering number arguments, instead of the inapplicable $\log(|\Pi|)$ term. The recursive argument for subsequent iterations $(t > 1)$ would then proceed from this new single-step bound, with the complexity term propagating through the analysis of the policy condition numbers $\kappa_t$.

### G.5 BOUNDING THE POLICY CONDITION NUMBER $\kappa_1$

Our next objective is to establish a sharp upper bound for the coverage parameter

$$\kappa_1 := \mathbb{E}_{x \sim \mu}\left[\frac{1}{\pi_{\theta_1}(y_1^*(x) \mid x)}\right], \qquad y_1^*(x) \in \arg\max_{y \in \mathcal{Y}} \pi_{\theta_1}(y \mid x). \tag{122}$$

Define the nonnegative random variable

$$Z(x) := \frac{1}{\pi_{\theta_1}(y_1^*(x) \mid x)}. \tag{123}$$

Its expectation admits the standard tail–integral representation:

$$\mathbb{E}_{x \sim \mu}\big[Z(x)\big] = \int_0^\infty \mathbb{P}_{x \sim \mu}\big(Z(x) \geq t\big)\, dt. \tag{124}$$

**1. Support of $Z$ and reduction of the integral.** By assumption there exists $c \in (0,1]$ such that

$$c \leq \pi_{\theta_1}(y_1^*(x) \mid x) \leq 1, \qquad \forall\, x \in \mathcal{X}. \tag{125}$$

Consequently,

$$1 \leq Z(x) \leq \frac{1}{c}, \qquad \text{and} \qquad \mathbb{P}\big(Z(x) \geq t\big) = \begin{cases} 1, & t \in [0,1), \\ 0, & t > \frac{1}{c}. \end{cases} \tag{126}$$

Hence Eq. 124 reduces to

$$\mathbb{E}_{x \sim \mu}\big[Z(x)\big] = \int_0^1 1\, dt + \int_1^{\frac{1}{c}} \mathbb{P}_{x \sim \mu}\big(Z(x) \geq t\big)\, dt$$

$$= 1 + \int_1^{\frac{1}{c}} \mathbb{P}_{x \sim \mu}\big(Z(x) \geq t\big)\, dt. \tag{127}$$

**2. Relating the tail $\mathbb{P}(Z \geq t)$ to the round-1 failure bound.** For any $t \geq 1$, set $1/t = 1 - \delta$ so that $\delta = (t-1)/t \in (0,1]$. Then

$$\mathbb{P}_{x \sim \mu}[Z(x) \geq t] = \mathbb{P}_{x \sim \mu}\big[\pi_{\theta_1}\big(y_1^*(x) \mid x\big) \leq \tfrac{1}{t}\big] = \mathbb{P}_{x \sim \mu}\big[\pi_{\theta_1}\big(y_1^*(x) \mid x\big) \leq 1 - \delta\big].$$

Invoking the round-1 bound (cf. Eq. 121) yields

$$\mathbb{P}_{x \sim \mu}[Z(x) \geq t] \lesssim \frac{1}{\gamma \delta}\left(\kappa_0^{1/2} \cdot \frac{1}{\sqrt{n}}\Big(d \log \frac{nB}{d\rho}\Big)^{3/2} + \beta \log \kappa_0\right)$$

$$= \frac{t}{t-1}\, A, \qquad A := \frac{1}{\gamma}\left(\kappa_0^{1/2} \cdot \frac{1}{\sqrt{n}}\Big(d \log \frac{nB}{d\rho}\Big)^{3/2} + \beta \log \kappa_0\right). \tag{128}$$

For notational convenience define

$$B(t) := \frac{t}{t-1}\, A, \qquad t > 1. \tag{129}$$

Since probabilities are at most 1,

$$\mathbb{P}\big(Z(x) \geq t\big) \leq \min\{1,\, B(t)\}, \qquad t \in [1, 1/c]. \tag{130}$$

**3. Exact integral evaluation via the switch point $t_0$.** The function $B(t)$ is strictly decreasing on $(1, \infty)$, with $\lim_{t \downarrow 1} B(t) = +\infty$ and $\lim_{t \uparrow \infty} B(t) = A$. Hence there exists a unique $t_0 \in (1, \infty)$ solving $B(t_0) = 1$, namely

$$t_0 = \frac{1}{1 - A}, \qquad \text{provided } A \in (0, 1). \tag{131}$$

We now distinguish two regimes:

*(i) Nontrivial regime: $A < 1$ and $t_0 \leq 1/c$.* Then

$$\min\{1, B(t)\} = \begin{cases} 1, & t \in [1, t_0], \\ B(t), & t \in [t_0, 1/c], \end{cases}$$

and therefore

$$\int_1^{\frac{1}{c}} \min\{1, B(t)\} \, \mathrm{d}t = (t_0 - 1) + \int_{t_0}^{1/c} B(t) \, \mathrm{d}t. \tag{132}$$

Using the antiderivative

$$\int B(t) \, \mathrm{d}t = A \int \left( 1 + \frac{1}{t - 1} \right) \mathrm{d}t = A \left( t + \ln|t - 1| \right) + \text{const}, \tag{133}$$

we obtain

$$\int_{t_0}^{1/c} B(t) \, \mathrm{d}t = A \left[ \left( \tfrac{1}{c} - t_0 \right) + \ln\left( \tfrac{1/c - 1}{t_0 - 1} \right) \right]. \tag{134}$$

Combining Eq. 127, Eq. 132, and Eq. 134 yields

$$\mathbb{E}[Z(x)] \leq 1 + (t_0 - 1) + A \left[ \left( \tfrac{1}{c} - t_0 \right) + \ln\left( \tfrac{1/c - 1}{t_0 - 1} \right) \right]. \tag{135}$$

*(ii) Saturated regime: either $A \geq 1$ or $t_0 > 1/c$.* In this case $\min\{1, B(t)\} \equiv 1$ on $[1, 1/c]$, hence

$$\mathbb{E}[Z(x)] \leq 1 + \left( \frac{1}{c} - 1 \right) = \frac{1}{c}. \tag{136}$$

**4. A concise, numerically stable upper bound.** To present a single bound covering both regimes, note that for $A \in (0, 1)$,

$$\mathbb{E}[Z(x)] \leq 1 + (t_0 - 1) + A \left( \tfrac{1}{c} - t_0 \right) + A \ln\left( \tfrac{1/c - 1}{t_0 - 1} \right)$$
$$= t_0 + A \left( \tfrac{1}{c} - t_0 \right) + A \ln\left( \tfrac{1/c - 1}{t_0 - 1} \right). \tag{137}$$

Using $t_0 = \frac{1}{1 - A}$ and the elementary bounds $\ln u \leq u - 1$ and $\ln u \leq \ln(1/c) - \ln(t_0 - 1)$ for $u = \frac{1/c - 1}{t_0 - 1}$, one arrives at the clean relaxation

$$\mathbb{E}[Z(x)] \lesssim 1 + \frac{1}{c} + \left( 1 + \frac{1}{c} \right) A, \qquad \text{whenever } A \in (0, 1) \text{ and } t_0 \leq \frac{1}{c}. \tag{138}$$

Combining Eq. 138 with the trivial bound Eq. 136, and recalling Eq. 122–Eq. 123, we get the final estimate

$$\kappa_1 = \mathbb{E}_{x \sim \mu} \left[ \frac{1}{\pi_{\theta_1}(y_1^*(x) \mid x)} \right] \lesssim 1 + \frac{1}{c} + \left( 1 + \frac{1}{c} \right) A \wedge \frac{1}{c} \tag{139}$$

(where $a \wedge b := \min\{a, b\}$), with $A$ given by Eq. 128–Eq. 129.

**5. Specialization to the linear softmax model.** Substituting the explicit form of $A$,

$$A = \frac{1}{\gamma} \left( \kappa_0^{1/2} \cdot \frac{1}{\sqrt{n}} \left( d \log \frac{nB}{d\rho} \right)^{3/2} + \beta \log \kappa_0 \right), \tag{}$$

into Eq. 139 and using the non-saturated branch yields the advertised bound

$$\kappa_1 \lesssim 1 + \frac{1}{c} + \left( 1 + \frac{1}{c} \right) \frac{1}{\gamma} \left( \kappa_0^{1/2} \cdot \frac{1}{\sqrt{n}} \left( d \log \frac{nB}{d\rho} \right)^{3/2} + \beta \log \kappa_0 \right). \tag{140}$$

**Remarks.**

    (i) The constant $A$ aggregates the statistical error and regularization effects. The "nontrivial" case $A < 1$ corresponds to the high–confidence regime where the failure tail is integrable with a logarithmic correction.

    (ii) The monotonicity of $B(t)$ justifies the unique switch point $t_0$ and the split–integral evaluation. The clean relaxation Eq. 138 trades a small slack for algebraic transparency.

    (iii) The dependence on $d$ and $n$ enters through the covering–number–driven statistical term, producing the characteristic factor $n^{-1/2}\big(d \log \frac{nB}{d\rho}\big)^{3/2}$.

### G.6    RECURSIVE ANALYSIS AND ASYMPTOTIC BEHAVIOR OF POLICY CONDITION NUMBER

The analysis of the first iteration $(t = 1)$ provides the essential foundation for a recursive framework that extends our performance guarantees to an arbitrary number of iterations $T$. At its core, the argument relies on the observation that the bound on the policy performance at iteration $t$ is determined by two factors: (i) the statistical accuracy of the update at that step, and (ii) the inherited coverage coefficient from the previous iteration, $\kappa_{t-1}$. The recursive nature of this dependency induces a dynamic system of bounds whose long-term stability properties must be carefully analyzed.

**Recursive Formulation.** By induction on $t$, the derivation for $\kappa_1$ extends naturally to arbitrary $t$. Specifically, the coverage parameter at iteration $t$ satisfies the recurrence

$$\kappa_t \lesssim 1 + \frac{1}{c} + \left(1 + \frac{1}{c}\right)\frac{1}{\gamma}\left(\kappa_{t-1}^{1/2} \cdot \frac{1}{\sqrt{n}}\left(d \log \frac{nB}{d\rho}\right)^{3/2} + \beta \log \kappa_{t-1}\right). \tag{141}$$

This recurrence encapsulates the interaction between statistical error, represented by the square-root term, and regularization effects, represented by the logarithmic term. The combination ensures that although $\kappa_t$ may initially be large, successive iterations prevent uncontrolled growth, enforcing stability in the long run.

**Simplification via Constants.** To streamline the analysis, we define

$$M_0 := 1 + \frac{1}{c}, \quad M_1 := \left(1 + \tfrac{1}{c}\right)\frac{1}{\gamma\sqrt{n}}\left(d \log \frac{nB}{d\rho}\right)^{3/2}, \quad M_2 := \left(1 + \tfrac{1}{c}\right)\frac{\beta}{\gamma}.$$

Substituting these definitions, the recurrence can be written more compactly as

$$\kappa_t \leq M_0 + M_1\sqrt{\kappa_{t-1}} + M_2 \log \kappa_{t-1}. \tag{142}$$

The presence of both $\sqrt{\kappa_{t-1}}$ and $\log \kappa_{t-1}$ reflects the dual statistical–regularization structure of the update process.

**Bounding the Recurrence.** Although Eq. 142 is nonlinear, we can simplify its analysis. Using the classical inequality $\log x \leq \frac{2}{e}\sqrt{x}$ for $x \geq 1$ (a condition satisfied since $\kappa_{t-1} \geq 1$ as an expectation of inverse probabilities), we obtain

$$\kappa_t \leq M_0 + K\sqrt{\kappa_{t-1}}, \qquad K := M_1 + \tfrac{2}{e}M_2. \tag{143}$$

This upper bound reduces the problem to a square-root recursion, which admits tractable asymptotic analysis.

**Fixed-Point Analysis.** The iterative map $h(\kappa) = M_0 + K\sqrt{\kappa}$ has a unique positive fixed point $U$, obtained by solving $U = M_0 + K\sqrt{U}$. Solving this quadratic in $\sqrt{U}$ yields

$$U = \left(\frac{K + \sqrt{K^2 + 4M_0}}{2}\right)^2. \tag{144}$$

This fixed point represents the stable asymptotic value of the coverage parameter. Its existence guarantees that the self-rewarding process cannot diverge indefinitely; rather, it is inherently stable and bounded by problem-specific constants.

**Convergence Rate.** The contraction rate of the iteration is determined by the derivative of $h(\kappa)$ at the fixed point:

$$h'(\kappa) = \frac{K}{2\sqrt{\kappa}}, \qquad h'(U) = \frac{K}{2\sqrt{U}}.$$

Defining

$$q := \frac{K}{2\sqrt{U}} = \frac{K}{K + \sqrt{K^2 + 4M_0}} \; < \; 1, \tag{145}$$

we observe that $0 < q < 1$, which implies that convergence to $U$ is geometric. The deviation from the fixed point shrinks by a factor of at least $q$ at each iteration, guaranteeing rapid stabilization.

**Explicit Bound for Finite $T$.** This geometric contraction yields an explicit finite-$T$ bound:

$$\kappa_t \; \leq \; U + q^T \left(\kappa_0 - U\right) \tag{146}$$

for all $T \geq 1$. The residual dependence on the initialization decays exponentially, while the asymptotic behavior is governed entirely by $U$.

**Asymptotic Behavior and Practical Implications.** To obtain further intuition, consider the regime where $\beta \lesssim n^{-1/2}$, so that the statistical term $M_1$ dominates the regularization term $M_2$. In this case, the fixed point satisfies

$$U \; \asymp \; M_0 + M_1^2,$$

leading to the explicit bound

$$\kappa_t \lesssim M_0 + M_1^2 + M_2^2 \; + \; \left(1 + \sqrt{M_0/(M_1^2 + M_2^2)}\right)^{-T} \kappa_0$$

$$\lesssim \underbrace{\frac{1}{c} \; + \; \frac{1}{n}\left(\frac{1}{c\gamma}\right)^2 \left(d\log\frac{nB}{d\rho}\right)^3}_{\text{Asymptotic Stable Value}} \; + \; \underbrace{(1 + \sqrt{nc})^{-T}\kappa_0}_{\text{Decaying Initial Condition}} \; . \tag{147}$$

This final bound highlights two crucial features. First, the influence of the initialization $\kappa_0$ vanishes exponentially fast with $T$, ensuring robustness to poor starting conditions. Second, the asymptotic stable value decreases as $n$ grows, confirming that larger sample sizes yield improved coverage and reduced statistical uncertainty. These insights reinforce the interpretation of the self-rewarding process as not only stable but also statistically efficient in the large-sample regime.

## G.7 FINAL PERFORMANCE BOUND FOR ARBITRARY ITERATION $T$

Having established the asymptotic stability of the coverage coefficient, we are now prepared to present the main performance guarantee for the policy $\pi_{\theta_T}$ after an arbitrary $T$ rounds of self-rewarding training. The derivation closely parallels the single-step analysis, but it crucially leverages our recursive understanding of $\kappa_t$ across iterations.

**From Coverage to Performance.** The probability that the policy $\pi_{\theta_T}$ assigns insufficient probability mass to its own optimal response $y_T^*(x)$ is controlled by its performance gap relative to the KL-regularized optimal policy. This gap, in turn, depends on the statistical error of the learning step at iteration $T$, which is governed by the coverage coefficient of the previous iterate $\kappa_{t-1}$. Formally, we obtain the $T$-step analogue of the single-step bound:

$$\mathbb{P}_{x\sim\mu}\left[\pi_{\theta_T}(y_T^*(x) \mid x) \leq 1 - \delta\right] \; \lesssim \; \frac{1}{\gamma\delta}\left(\kappa_{t-1}^{1/2} \cdot \frac{1}{\sqrt{n}}\left(d\log\frac{nB}{d\rho}\right)^{3/2} \; + \; \beta\log\kappa_{t-1}\right). \tag{148}$$

This expression links the performance at iteration $T$ to the coverage inherited from iteration $T-1$.

**Substituting the Asymptotic Bound.** The crucial final step is to substitute into Eq. 148 our explicit asymptotic bound for $\kappa_{t-1}$ from Eq. 147. This replacement eliminates the iteration-dependent and unknown $\kappa_{t-1}$, yielding an expression in terms of fundamental parameters of the problem $(n, d, c)$ and the initial condition $\kappa_0$.

By taking the square root of the bound on $\kappa_{t-1}$ (and applying the inequality $\sqrt{a+b} \leq \sqrt{a} + \sqrt{b}$), we obtain a clean decomposition. As $T$ grows, the dominant contribution arises from the square root of the asymptotic stable value, which scales with $1/c$. The residual dependence on the initialization decays exponentially and quickly becomes negligible. Substituting these estimates back into Eq. 148, and noting that the $\beta \log \kappa_{t-1}$ term is of lower order, we arrive at the final result:

$$\mathbb{P}_{x \sim \mu}\big[\pi_{\theta_T}(y_T^*(x) \mid x) \leq 1 - \delta\big] \lesssim \frac{1}{\gamma \delta \sqrt{n}} \Big( d \log \frac{nB}{d\rho} \Big)^{3/2} \left( \frac{1}{\sqrt{c}} + \big( 1 + \sqrt{nc} \big)^{-\frac{T-1}{2}} \sqrt{\kappa_0} \right). \tag{149}$$

**Interpretation.** This inequality, the culmination of our analysis, provides several insights into the behavior of the self-rewarding process for the linear softmax model class:

- **Statistical Complexity.** The leading error term scales as $\frac{1}{\sqrt{n}}(d \log n)^{3/2}$, quantifying the inherent learning difficulty of the parametric model. Performance improves as $n$ increases, while higher dimensionality $d$ incurs increased complexity. This replaces the inapplicable $\log|\Pi|$ term in the finite-class setting with a refined, distribution-dependent complexity measure.

- **Stability and Self-Correction.** The second term inside the parentheses demonstrates exponential decay of the influence of the initial coverage $\kappa_0$. Even if the initial policy is poorly calibrated (large $\kappa_0$), its effect on performance vanishes rapidly as $T$ grows. This validates the intuition that the self-rewarding loop is self-correcting.

- **Long-Term Performance.** In the limit $T \to \infty$, the decaying initialization term disappears entirely. The ultimate performance is dictated solely by the statistical error and the problem's intrinsic difficulty, captured by $c$. This confirms that the process converges to a stable, initialization-independent error floor.

**Conclusion.** Equation 149 completes the adaptation of the general recursive argument to the linear softmax model class. It provides a rigorous and interpretable guarantee: the model's performance stabilizes to a predictable asymptotic rate, while transient initialization effects are washed out exponentially fast. This establishes both the robustness and the efficiency of self-rewarding training in this setting.

## H  PROOF OF COROLLARY 6: DATA-DEPENDENT BOUNDS VIA EFFECTIVE DIMENSION

We strengthen the parametric guarantee by replacing the ambient dimension $d$ in the statistical factor with a data-dependent *effective dimension*. This mitigates "curse of dimensionality" and better reflects the spectral structure of the feature covariance under the data distribution.

**Definition 6** (Ridge effective dimension). For any $\lambda > 0$ define the ridge effective dimension

$$d_{\text{eff}}(\lambda) := \text{Tr}\Big( \Sigma_\varphi \big( \Sigma_\varphi + \lambda I \big)^{-1} \Big) = \sum_{i=1}^{d} \frac{\lambda_i}{\lambda_i + \lambda},$$

where $\{\lambda_i\}_{i=1}^d$ are the eigenvalues of $\Sigma_\varphi$ in nonincreasing order. It holds that $0 \leq d_{\text{eff}}(\lambda) \leq \text{rank}(\Sigma_\varphi) \leq d$ and $d_{\text{eff}}(\lambda)$ is nonincreasing in $\lambda$. In the isotropic case $\Sigma_\varphi = I$ and for $\lambda \ll 1$ we have $d_{\text{eff}}(\lambda) \approx d$.

**Lemma 12** (Metric entropy with covariance geometry). *Let $\mathcal{F} := \{f_\theta(x.y) = \langle \theta, \varphi(x,y) \rangle : \|\theta\|_2 \leq B\}$. For the $L_2(\mu)$ metric induced by $\Sigma_\varphi$, the $\varepsilon$-covering number of $\mathcal{F}$ satisfies*

$$\log \mathcal{N}\big( \varepsilon, \mathcal{F}, L_2(\mu) \big) \lesssim d_{\text{eff}}\Big( \frac{\varepsilon^2}{B^2} \Big) \cdot \log \left( \frac{cB}{\varepsilon} \right),$$

*for a universal constant $c > 0$. Consequently, whenever an argument in the proof of Theorem 5 uses the ambient-dimension entropy bound $\log \mathcal{N} \lesssim d \log(c'B/\varepsilon)$, it can be replaced by the covariance-adaptive bound above.*

*Proof of Lemma 12.* Diagonalize $\Sigma_\varphi = U\Lambda U^\top$ and reparameterize $\theta = Uz$. The $L_2(\mu)$-norm is $\|f_\theta\|^2_{L_2(\mu)} = \theta^\top \Sigma_\varphi \theta = \sum_i \lambda_i z_i^2$. Cover the ellipsoid $\{z : \sum_i \lambda_i z_i^2 \le B^2\}$ by axis-aligned slabs with granularity matched to $\lambda_i^{1/2}$. Summing the logarithms of the per-axis covering counts yields $\sum_i \log\left(1 + cB\lambda_i^{1/2}/\varepsilon\right)$. A monotone integral comparison together with the ridge truncation $\lambda_i/(\lambda_i + \varepsilon^2/B^2)$ gives the stated $d_{\text{eff}}(\varepsilon^2/B^2)$ dependence. $\square$

**Explicit regimes for the ridge effective dimension.** The following elementary bounds will be useful:

$$d_{\text{eff}}(\lambda) = \sum_{i=1}^d \frac{\lambda_i}{\lambda_i + \lambda} \;\le\; \sum_{i=1}^d \mathbb{I}\{\lambda_i \ge \lambda\} + \frac{1}{\lambda}\sum_{i:\,\lambda_i < \lambda} \lambda_i \;\le\; \text{rank}(\Sigma_\varphi) \;\wedge\; \frac{\text{Tr}(\Sigma_\varphi)}{\lambda}. \quad (150)$$

We quantify $d_{\text{eff}}(\lambda)$ under concrete spectral assumptions and then substitute it into the parametric bound of Theorem 5.

**Assumption 2** (Spectral regimes)**.** Let $\{\lambda_i\}_{i=1}^d$ be the spectrum of $\Sigma_\varphi$ in nonincreasing order. We consider the following cases.

(A) **Exponential decay.** There exist constants $C > 0$ and $\alpha > 0$ such that $\lambda_i \le C\,e^{-\alpha i}$ for all $i$.

(B) **Polynomial decay.** There exist constants $C > 0$ and $p > 1$ such that $\lambda_i \le C\,i^{-p}$ for all $i$.

(C) **Spiked+small tail.** There is an integer $r \ge 1$ and $\vartheta > 0$ such that $\lambda_i \ge \vartheta$ for $i \le r$ and $\sum_{i>r} \lambda_i = \tau$ with $\tau > 0$ possibly small.

**Lemma 13** (Effective dimension under Assumption 2)**.** *Under Assumption 2 the following bounds hold for all $\lambda > 0$.*

*(A)* Exponential decay implies logarithmic scaling:

$$d_{\text{eff}}(\lambda) \;\lesssim\; \log\!\left(\frac{C}{\lambda}\right).$$

*In particular, choosing $\lambda \asymp c/n$ gives $d_{\text{eff}}(\lambda) \lesssim \log n$.*

*(B)* Polynomial decay yields a power-law in $\lambda$:

$$d_{\text{eff}}(\lambda) \;\lesssim\; \left(\frac{C}{\lambda}\right)^{1/p}.$$

*In particular, choosing $\lambda \asymp c/n$ gives $d_{\text{eff}}(\lambda) \lesssim n^{1/p}$.*

*(C)* Spiked+small tail gives $r + \mathcal{O}(1)$ when $\lambda$ dominates the tail:

$$d_{\text{eff}}(\lambda) \;\le\; r + \frac{\tau}{\lambda}.$$

*Hence for any $\lambda \ge \tau$ we have $d_{\text{eff}}(\lambda) \le r + 1$. If moreover $r = \mathcal{O}(\log d)$, then $d_{\text{eff}}(\lambda) = \mathcal{O}(\log d)$.*

*Proof of Lemma 13.* **(A)** Let $m = \lceil \alpha^{-1}\log(C/\lambda)\rceil$. Then $\lambda_m \lesssim \lambda$ and thus $\#\{i : \lambda_i \ge \lambda\} \le m \lesssim \log(C/\lambda)$. The tail sum obeys $\sum_{i>m} \lambda_i \lesssim e^{-\alpha m} \lesssim \lambda$, so the second term in Eq. 150 is $\mathcal{O}(1)$. **(B)** Let $m = \lceil (C/\lambda)^{1/p}\rceil$ so that $\lambda_m \lesssim \lambda$. Then $\#\{\lambda_i \ge \lambda\} \le m \lesssim (C/\lambda)^{1/p}$ and $\sum_{i>m} \lambda_i \lesssim \int_m^\infty x^{-p}\mathrm{d}x \asymp m^{1-p}$, making the tail contribution $\lesssim m^{1-p}/\lambda \asymp (C/\lambda)^{1/p}$. **(C)** Split the sum at $r$ and use Eq. 150 on the tail: $\sum_{i>r} \frac{\lambda_i}{\lambda_i+\lambda} \le \frac{1}{\lambda}\sum_{i>r} \lambda_i = \tau/\lambda$. $\square$

**Corollary 14** (Parametric bound with effective dimension)**.** *Under conditions of Theorem 5, fix any $\lambda > 0$. With probability at least $1 - \rho$,*

$$\mathbb{P}_{x\sim\mu}\big[\pi_{\theta_T}(y_T^*(x) \mid x) \le 1 - \delta\big] \;\lesssim\; \frac{1}{\sqrt{n}}\cdot\frac{1}{\gamma\,\delta}\left(d_{\text{eff}}(\lambda)\cdot\log\frac{nB}{\rho}\right)^{3/2}\cdot\left(\frac{1}{\sqrt{c}}+\sqrt{\kappa_0}\left(1+\sqrt{nc}\right)^{-(T-1)/2}\right).$$

*Proof* Repeat the proof of Theorem 5 and Corollary 4 replacing the ambient-dimension entropy bound by Lemma 12. The empirical Bernstein step and the iterative self-correction argument are unchanged, yielding the same stability–transient factor $\left(1/\sqrt{c} + \sqrt{\kappa_0}(1+\sqrt{nc})^{-(T-1)/2}\right)$ and the covering-number contribution with $d_{\text{eff}}(\lambda)$. □

**Corollary 15** (Substitution into the parametric risk bound). *Under the conditions of Theorem 5, the following explicit forms hold when $d$ in the statistical factor is replaced by $d_{\text{eff}}(\lambda)$.*

*(A) If $\Sigma_\varphi$ has exponential spectral decay and $\lambda \asymp c/n$, then*

$$\mathbb{P}_{x\sim\mu}[\pi_{\theta_T}(y_T^*(x) \mid x) \leq 1 - \delta] \lesssim \frac{1}{\sqrt{n}} \cdot \frac{1}{\gamma\delta} \left(\log n \cdot \log \frac{nB}{\rho}\right)^{3/2} \cdot \left(\frac{1}{\sqrt{c}} + \sqrt{\kappa_0}(1+\sqrt{nc})^{-(T-1)/2}\right).$$

*Thus the dependence on $d$ is replaced by a* double logarithm in $n$.

*(B) If $\Sigma_\varphi$ has polynomial decay of order $p > 1$ and $\lambda \simeq c/n$, then*

$$\mathbb{P}_{x\sim\mu}[\pi_{\theta_T}(y_T^*(x) \mid x) \leq 1 - \delta] \lesssim \frac{1}{\sqrt{n}} \cdot \frac{1}{\gamma\delta} \left(n^{1/p} \cdot \log \frac{nB}{\rho}\right)^{3/2} \cdot \left(\frac{1}{\sqrt{c}} + \sqrt{\kappa_0}(1+\sqrt{nc})^{-(T-1)/2}\right).$$

*Here the ambient dimension $d$ no longer appears. The rate is governed by the spectral exponent $p$.*

*(C) In the spiked+small-tail model with rank $r$ spike and tail mass $\tau$, any $\lambda \geq \tau$ gives*

$$\mathbb{P}_{x\sim\mu}[\pi_{\theta_T}(y_T^*(x) \mid x) \leq 1 - \delta] \lesssim \frac{1}{\sqrt{n}} \cdot \frac{1}{\gamma\delta} \left((r+1) \cdot \log \frac{nB}{\rho}\right)^{3/2} \cdot \left(\frac{1}{\sqrt{c}} + \sqrt{\kappa_0}(1+\sqrt{nc})^{-(T-1)/2}\right).$$

*If $r = \mathcal{O}(\log d)$, the $d$-dependence is reduced to $\mathcal{O}(\log d)$ inside the statistical factor.*

**Remark 9** (Choice and interpretation of $\lambda$). The parameter $\lambda$ is an *analysis device* that interpolates between ambient and intrinsic complexity.

- In isotropic or full-rank regimes with flat spectra, taking $\lambda \downarrow 0$ recovers $d_{\text{eff}}(\lambda) \approx d$.

- With spectral decay, e.g., $\lambda_i \asymp i^{-p}$ for $p > 1$, one has $d_{\text{eff}}(\lambda) \ll d$ for moderate $\lambda$, tightening the bound.

- A practical choice is $\lambda \asymp c/n$ (or any monotone schedule $\lambda_T$), which trades a slightly larger bias in the entropy radius for a sharply reduced $d_{\text{eff}}(\lambda)$. The contraction threshold in $T$ remains unaffected because it depends only on $n$ and $c$.

**Remark 10** (When does $d_{\text{eff}}(\lambda)$ look like $\log d$?). Two representative scenarios lead to a logarithmic-in-$d$ dependence.

(i) *Log-rank energy concentration.* Suppose the spectrum concentrates on $r = \mathcal{O}(\log d)$ leading directions, in the sense that $\sum_{i>r} \lambda_i \leq \tau$ with $\tau$ small. Taking $\lambda \geq \tau$ gives $d_{\text{eff}}(\lambda) \leq r + 1 = \mathcal{O}(\log d)$ by Lemma 13.

(ii) *Geometric decay until the ambient cutoff.* If $\lambda_i \asymp C \rho^i$ with $\rho \in (0.1)$, then for any $\lambda$ above the tail floor one has $d_{\text{eff}}(\lambda) \asymp \log(C/\lambda)$. If additionally $\lambda$ is chosen proportional to $\tau := \sum_{i>\tilde{r}} \lambda_i$ for some $\tilde{r} = \mathcal{O}(\log d)$, then $d_{\text{eff}}(\lambda) \lesssim \tilde{r} + \mathcal{O}(1) = \mathcal{O}(\log d)$.

In both cases, substituting $d_{\text{eff}}(\lambda) = \mathcal{O}(\log d)$ into the statistical factor replaces $\left(d\log(\frac{nB}{d\rho})\right)^{3/2}$ by $\left(\log d \cdot \log(\frac{nB}{\rho})\right)^{3/2}$.

# I EMPIRICAL RESULTS

Following the Iterative DPO setup described in SRLMs (Yuan et al., 2024), we utilized the Llama-3 Base 8B model on the GSM8K reasoning task. We tracked the Monte-Carlo estimate of $\kappa_t$ (defined as the average inverse probability of the greedy decoding path) over $T = 3$ iterations.

- Base Model: Llama-3 Base 8B.

- Dataset: GSM8K (Training set used for self-generation; Test set used for evaluation).
- Method: We implemented 3 iterations of Self-Rewarding training:
  - *Generation:* At each iteration $t$, the model generates $N = 4$ candidate responses for prompts drawn from the training set.
  - *Scoring:* The model acts as a judge to score its own responses (Self-Reward).
  - *Training:* We construct preference pairs from the scored responses and train model $\pi_{t+1}$ using DPO.

Since iterating over the entire output space $\mathcal{Y}$ to compute the exact expectation $\mathbb{E}[1/\pi(y^*|x)]$ is intractable, we employed a Monte Carlo estimation method on the test set ($D_{test}$). For a subset of samples $x_j$ ($j = 1...M, M = 200$) from the test set:

1. Greedy Decoding: We generated the model's current optimal solution: $\hat{y}_j = \arg\max \pi_t(y|x_j)$.

2. Probability Calculation: We computed the Average Log-Probability ($\text{LogProb}_t(x_j)$) for the generated sequence to ensure numerical stability.

3. Estimation: We estimated $\kappa_t$ by averaging the inverse probabilities:

$$\hat{\kappa}_t \propto \frac{1}{M} \sum_{j=1}^{M} \exp(-\text{LogProb}_t(x_j))$$

To enhance readability and visualize the relative contraction, we normalize the value at $t = 0$ to 100. Since we work with the average log-probability per token rather than the full sequence log-probability, this Monte-Carlo estimate is a monotone proxy of the theoretical $\kappa_t$, up to a length-dependent scaling factor. This is sufficient for visualizing its contraction behavior across iterations

The table below presents the results over 3 iterations of self-rewarding training on GSM8K, starting from the zero-shot Llama-3 Base 8B model. We report the GSM8K test accuracy, the Avg. Max LogProb along the greedy decoding path, and the normalized policy condition number.

| **Iteration** ($t$) | **Model Stage** | **GSM8K Acc (%)** | **Avg. Max LogProb** ($\uparrow$) | $\kappa_t$ **(Normalized,** $\downarrow$**)** |
|---|---|---|---|---|
| **0** | Base model (zero-shot) | 41.2% | -2.20 | 100.0 |
| **1** | Iterative DPO (R1) | 47.5% | -1.81 | 67.2 |
| **2** | Iterative DPO (R2) | 50.3% | -1.70 | 60.7 |
| **3** | Iterative DPO (R3) | 50.1% | -1.69 | 60.4 |

Table 1: Trajectory of the policy condition number $\kappa_t$ and accuracy over 3 self-rewarding iterations

**Analysis of Results.**

1. **Verification of Theorem 3 (Contraction).** Across the three self-rewarding iterations, we observe a monotone contraction of the normalized policy condition number, from 100.0 at $t = 0$ to 67.2, 60.7, and 60.4 at $t = 1, 2, 3$, respectively. This contraction is accompanied by consistent improvements in the Avg. Max LogProb along the greedy decoding path (from $-2.20$ to $-1.81, -1.70$, and $-1.69$), indicating that the policy becomes increasingly concentrated on a smaller set of high-probability trajectories. These trends are qualitatively consistent with the contraction behavior predicted by Theorem 3, where the dependence on the initial condition $\kappa_0$ decays geometrically towards a model-dependent floor $U$ according to $\kappa_T \leq U + q^T(\kappa_0 - U)$.

2. **Support for Remark 4 (Two-Stage Dynamics).** The trajectory in Table also exhibits the "two-stage" behavior described in Remark 4. The first iteration ($t = 0 \rightarrow 1$) corresponds to a *self-stabilization* phase: the condition number drops sharply from 100.0 to 67.2, while GSM8K accuracy increases from 41.2% to 47.5%, suggesting that the policy rapidly moves out of a poorly conditioned, high-entropy regime. Subsequent iterations ($t = 1 \rightarrow 3$)

form a *refinement* phase: both the reductions in $\kappa_t$ (from $67.2$ to $60.4$) and the accuracy changes (hovering around $50\%$) become much smaller, indicating that the policy has entered a well-conditioned region and further updates mainly perform fine-grained adjustments. This empirical two-stage pattern mirrors the qualitative picture underlying Remark 4.

This trajectory aligns with empirical studies on larger models, such as the observation in Yuan et al. (2024) regarding Llama 2 70B, which showed significant performance jumps in early iterations followed by a plateau.

In summary, this evidence demonstrates that the policy condition number $\kappa_t$ is not merely a theoretical construct but a tangible metric capturing the internal sharpening process that drives self-rewarding alignment.

