# OpenReview forum: "Theoretical Guarantees for Iterative Alignment of Self-Rewarding Language Models"
_ICLR.cc/2026/Conference — Submitted to ICLR 2026_

### Official Review · Reviewer_u3cM · 2025-10-21

**Soundness:** 3
**Presentation:** 3
**Contribution:** 3
**Rating:** 6
**Confidence:** 2

**Summary:**

This paper provides the first rigorous theoretical analysis of Self-Rewarding Language Models (SRLMs), a paradigm that has recently shown strong empirical success. The authors aim to explain why iterative self-alignment works without external feedback. They introduce a "policy condition number" to quantify a model's suitability for self-alignment and establish a lower bound showing that a single update step can easily fail if the initial model is poorly conditioned. The main contribution is a finite-sample guarantee for the full iterative process. The analysis reveals that the dependence on the (potentially poor) initial model quality decays exponentially with the number of iterations. This provides a clear theoretical explanation for the success of SRLMs: the process first self-stabilizes before entering a standard statistical learning phase. The framework is then instantiated for linear softmax models, connecting the general theory to parametric model classes.

**Strengths:**

- This is a timely and important paper that addresses a critical gap in our understanding. While SRLMs are being used and developed rapidly, the theoretical principles behind their stability and success have been missing. To my knowledge, this is the first work to provide a formal theoretical foundation for this class of methods.
- The central idea of the paper is very powerful. The introduction of the "policy condition number" (`κt`) and the demonstration that the iterative update acts as a contraction mapping on this value is an elegant and insightful explanation. The resulting two-stage view of the process—first, an internal self-correction phase where the influence of a poor initialization vanishes, followed by an efficient statistical learning phase—is compelling and aligns with empirical intuitions.
- The paper is technically sound and comprehensive. It correctly identifies the core challenges by first establishing a lower bound on the failure of single-step updates (Theorem 1), which motivates the need for iteration. The main upper bound (Theorem 3) is the key result, and its tightness in the single-step case is a strong sign of a robust analysis. The detailed proofs in the appendix appear thorough.
- The authors make a good effort to connect their theoretical findings to practical scenarios. The analysis of the number of iterations required (Corollary 4) provides a practical insight, suggesting that even very poorly conditioned models can be aligned without an excessive number of steps. Furthermore, the extension to linear softmax models and the discussion of effective dimensionality (Corollary 6) make the results more relevant to the overparameterized models used in practice.

**Weaknesses:**

- The theoretical model, while insightful, relies on a few key simplifying assumptions.
    -  The self-reward function is defined specifically as the model's log-probability. While this is a natural starting point, many practical SRLM approaches use the model in a more complex, "LLM-as-a-Judge" capacity to generate scores or critiques. The applicability of the theory to these more complex reward schemes is unclear.
   -  The realizability assumption (Assumption 1) is common in theoretical analyses but can be a strong requirement in practice.
- The extension to linear softmax models is a necessary and welcome step to move from finite model classes to parametric ones. However, this is still a considerable abstraction from the complex, non-linear dynamics of Transformer architectures. While this is a limitation of most current theoretical work in deep learning, it's worth acknowledging the potential gap.
- The paper is mathematically dense. While this is expected for a strong theory paper, the main text could benefit from building a bit more intuition around some of the key constants. For example, what would a low vs. high "margin constant" (γ) or "minimum confidence" (c) mean in the context of a practical LLM's output distribution?

**Questions:**

1.  The choice of log π(y|x) as the reward is crucial to the analysis. Could you comment on the challenges of extending this framework to other self-reward functions? For instance, if the reward is a scalar score produced by the LLM in a separate forward pass (i.e., LLM-as-a-Judge), does the core mechanism of the policy condition number contracting still hold?
2.  The policy condition number κt is a fantastic theoretical tool. Does this quantity have a practical analogue that could be estimated or tracked during training? It seems like it could serve as a powerful diagnostic to monitor whether the model is in the "stabilization" or "learning" phase.

---

> ### Author Response · Authors · 2025-11-21
> **Part I: Generalization to Scalar Rewards (Response to Q1)**
>
> Thank you for the constructive feedback. We have addressed all concerns and revised the paper accordingly, with changes highlighted in blue.
>
> **Q1: The self-reward uses the model’s log-probability, a natural starting point, but many SRLM methods instead adopt more complex “LLM-as-a-Judge’’ scores. What are the challenges in extending your framework to such self-reward functions (e.g., scalar scores from a separate forward pass), and does the policy condition number still provably contract in these cases?**
>
> **A:** We thank the reviewer for this insightful question. While our theoretical analysis employs log-probability as the self-reward formulation to ensure mathematical consistency and capture the widely observed self-preference phenomenon [1], we wish to emphasize that the proposed contraction mechanism of the policy condition number ($\kappa_t$) is not restricted to this specific setting. For scalar rewards (e.g., LLM-as-a-Judge), the core contraction mechanism **holds** provided the signal maintains basic structural properties such as ranking consistency and bounded noise. The primary difference is that the convergence bound incorporates an irreducible error term dominated by the Judge's inherent noise. We detail this extension and its challenges below:
>
> ### 1. Motivation for the Log-Probability Formulation
>
> Our choice of $r_t = \log \pi_t$ serves as both a rigorous theoretical anchor and an empirical abstraction:
>
> - **Theoretical Consistency:** This form aligns strictly with the KL-regularized DPO objective, allowing us to model the RL process as a smooth flow on the probability simplex. This tractability is what enables us to derive the exact contraction properties of $\kappa_t$.
> - **Modeling Self-Preference:** Empirical studies [1] indicate that LLM judges exhibit self-preference by favoring high-confidence outputs. Consequently, our log-probability reward serves as an **idealized limit** of the LLM-as-a-Judge mechanism where the Judge is perfectly aligned with the policy and free of external noise.
>
> ### 2. Extending to Scalar Rewards: Challenges and Robustness
>
> We acknowledge that practical LLM-as-a-Judge rewards are often scalar scores (e.g., integers 1–5) generated in a separate pass. Extending our framework to this setting introduces challenges such as quantization noise, discrete optimization plateaus, and potential reward hacking. However, the contraction mechanism remains robust under a “noisy estimator” view. If we model the scalar reward $S_t(x, y)$ as a noisy approximation of an ideal smooth reward $r^*(x, y)$:
> $$
> S_ t(x, y) \asymp \lambda \cdot r^\*(x, y) + \mathcal{E}_ {\text{judge}}
> $$
> where $\mathcal{E}_{\text{judge}}$ captures quantization and calibration errors, the convergence dynamics hold provided that:
>
> 1. **Ranking Consistency:** $S_t$ remains positively correlated with $r^*$ (i.e., the Judge generally prefers higher-probability outputs).
> 2. **Bounded Noise:** The variance of $\mathcal{E}_{\text{judge}}$ does not dominate the signal.
>
> Under these conditions, the DPO update still shifts probability mass toward high-scoring regions, causing $\kappa_t$ to contract over iterations. The theoretical error bound in our Thm. 3 would conceptually transform from our current result to:
> $$
> \text{Error}_ T \lesssim \frac{1}{\sqrt{n}}\left(\frac{1}{\sqrt{c}} + \frac{\sqrt{\kappa_ 0}}{(1+\sqrt{nc})^{(T-1)/2}}\right) + O(\mathcal{E}_ {\text{judge}})
> $$
> Thus, the self-reward mechanism still exponentially suppresses the poor initialization ($\kappa_0$). However, the final achievable alignment is no longer solely limited by sample size $n$, but by the “noise floor” of the Judge's bias. The model improves until it hits the resolution limit of its own judging capability.
>
> ### 3. The Gap is Narrowing: Soft-Label Practices
>
> Finally, we note that many state-of-the-art SRLM [2] are already moving closer to our theoretical assumptions. Rather than using discrete integer scores, practitioners increasingly use **soft-labels** derived from logits (e.g., the logit difference between token “5” and token “1”):
> $$
> S_{\text{soft}}(x,y) = \text{Logits}(s=5 \mid x,y) - \text{Logits}(s=1 \mid x,y).
> $$
> This practice restores continuity and differentiability to the reward function, effectively rendering it a smooth transformation of the model's internal representations similar to our log-probability assumption. In such soft-reward regimes, our theoretical predictions regarding $\kappa_t$ contraction apply with minimal modification.
>
> In summary, despite simplifying assumptions, our framework captures the core SRLM mechanism of uncertainty reduction through iterative self-play driven by the contraction of $\kappa_t$. This remains the key explanation for the effectiveness of self-rewarding models, even with scalar rewards.
>
> **Reference:**
>
> [1] Wataoka et al. *Self-preference bias in LLM-as-a-Judge.*
>
> [2] Wang et al. *Improving LLM-as-a-Judge Inference with the Judgment Distribution.*

---

> ### Author Response · Authors · 2025-11-21
> **Part II: Robustness to Approximate Realizability & Architectural Limitations (Response to Q2 & Q3)**
>
> **Q2: The realizability assumption (Assumption 1) is common in theoretical analyses but can be a strong requirement in practice.**
>
> **A:** We thank the reviewer for this insightful comment. We agree that the realizability assumption (Assumption 1), while common in theoretical analyses, is a strong requirement in practice. We employed this assumption as it is a standard axiomatization in the theoretical analysis of Supervised Learning, Reinforcement Learning, and related methods like DPO. This standard approach allows the analysis to isolate and focus on the crucial properties of statistical stability and convergence, temporarily setting aside concerns of model expressivity. Furthermore, in practice, the powerful expressivity of modern large models (LMs) often ensures that the true optimal policy can be well-approximated, making the approximation error $\varepsilon$ (as defined below) very small.
>
> Most importantly, our core results are robust to a relaxation of this assumption. The framework can be directly extended to the more practical setting of **approximate realizability**, where $\inf_ {\pi\in\Pi} D_ {\mathrm{KL}}(\pi^\*_ \beta \Vert \pi) \le \varepsilon$ for some small $\varepsilon > 0$. Under this relaxed condition, all of our main bounds would hold with the inclusion of an additional additive $O(\varepsilon)$ term. Crucially, the fundamental dynamic behavior of the system, which is the central focus of our paper, remains unaffected. The contraction mapping and the exponential convergence properties continue to hold in a perturbed form. Thus, while we utilized the stricter assumption for analytical clarity, our main conclusions regarding the stability and convergence of the iterative process are not an artifact of this assumption and remain valid in the more general approximate realizability setting.
>
>
>
> ---
>
>
> **Q3: The extension to linear softmax models is a necessary and welcome step to move from finite model classes to parametric ones. However, this is still a considerable abstraction from the complex, non-linear dynamics of Transformer architectures. While this is a limitation of most current theoretical work in deep learning, it's worth acknowledging the potential gap.**
>
> **A:** We thank the reviewer for this constructive and insightful point. We fully agree that while the extension to linear softmax models is a necessary and welcome step, it remains a considerable abstraction from the complex, non-linear dynamics of full Transformer architectures. As the reviewer rightly notes, this gap is a significant limitation shared by most current theoretical work in deep learning. To explicitly acknowledge this gap and situate our contributions more precisely, we have incorporated the reviewer's valuable feedback by making two key additions to the manuscript:
>
> 1. **In Remark 8 (Section 6),** immediately following the analysis of the linear softmax model, we have added the following sentence:
>
>    > *"While our guarantees hold for linear softmax models, extending this analysis to full Transformers is a significant open challenge and a valuable direction for future work."*
>
> 2. **In the Appendix,** we have added a new dedicated **Limitation** section:
>
>    > *"We acknowledge the significant challenge of extending the rigorous guarantees established in this work to full, modern Transformer architectures. The intricate components of these models, such as multi-head attention and deep residual connections, introduce complex non-linear dynamics and high-dimensional, non-convex optimization landscapes that are not captured by our current linear setting. Establishing a complete theoretical framework, such as a benign overfitting theorem or a proof of stable convergence, for such systems remains a major open problem in the theoretical deep learning community. At present, rigorous theories are largely confined to more tractable settings, primarily linear models (as adopted in our analysis) or two-layer neural networks. Therefore, bridging this substantial gap by extending the analytical framework to full-scale Transformers constitutes a highly valuable and essential direction for future research."*
>
> We believe these additions, prompted by the reviewer's feedback, make our manuscript more rigorous and transparent by clearly delineating the scope of our contributions and their relation to future challenges.

---

> ### Author Response · Authors · 2025-11-21
> **Part III: Physical Intuition of Constants $c$ and $\gamma$ (Response to Q4)**
>
> **Q4: The paper is mathematically dense. While this is expected for a strong theory paper, the main text could benefit from building a bit more intuition around some of the key constants. For example, what would a low vs. high "margin constant" ($\gamma$) or "minimum confidence" ($c$) mean in the context of a practical LLM's output distribution?**
>
> A: We thank the reviewer for this insightful suggestion. We agree that providing more intuition for the key constants will greatly improve the accessibility of our theoretical results. Below, we provide a more detailed intuition for both $c$ and $\gamma$ in the context of a practical LLM, which we have incorporated into the manuscript.
>
> ### 1. On the Minimum Confidence ($c$)
>
> The **minimum confidence** $c$ (Definition 1) is formally defined as $c = \inf_ {x \in \mathcal{X}, t} \pi_ t(y_ t^\*(x)|x)$, where $y_ t^\*(x)$ is the model's own top-1 prediction.
>
> **Intuition:** $c$ represents the most basic level of certainty the model has in its *own* most likely output, across all possible inputs and iterations. It answers the question: "In the worst-case scenario, does the model at least know which answer it wants to pick?"
>
> - **A Low $c$ (approaching 0):** This signifies a "low-confidence" or "diffuse" state, corresponding to practical LLM failure modes. The output distribution is flat (high-entropy), and the optimal answer $y^\*$ is effectively lost in a long tail. The model is unconfident about *all* outputs, making its self-reward signal weak and noisy. This leads to two critical failures:
>
>   1. **Invisibility and Vicious Cycle:** For a finite sampling budget $n$, the expected number of times the optimal answer $y^\*$ is sampled is $n \cdot c$. If $c$ is near-zero, $y^\*$ is never seen. The self-rewarding loop then reinforces the mediocre or incorrect samples it *does* see, further suppressing the probability of $y^\*$ and creating a vicious cycle.
>
>   2. **Model Collapse:** In recursive training, the model tends to prune low-probability regions. If $c$ is too low, the "truth" (the optimal answer) resides in this pruned tail. The model effectively "forgets" the correct answer and converges to a narrow, incorrect subspace, often becoming confidently wrong.
>
> - **A High $c$:** This implies a "capable" state. The probability mass is concentrated, and the reward signal is stable and reliable. The sampling process is effective, as the expected count of optimal samples ($n \cdot c$) is significant. This provides the necessary "positive examples" to drive policy improvement.
>
> ### 2. On the Margin Constant ($\gamma$)
>
> The **margin constant** $\gamma$ (Definition 1) is the infimum of the log-probability gap between the top-1 answer ($y_ t^\*(x)$) and its strongest competitor:
> $$
> \gamma := \inf_ {x, t} \log \left( \frac{\pi_ t(y_ t^\*(x)|x)}{\max_ {y' \neq y_ t^\*(x)} \pi_ t(y'|x)} \right).
> $$
>
> **Intuition:** $\gamma$ measures the model's "decisiveness," or its ability to clearly distinguish the best answer from the second-best.
>
> - **A Low $\gamma$ (approaching 0):** This represents "ambiguity." The model is "hesitant," assigning nearly equal probabilities to the best and second-best options (a flat, high-entropy distribution over top candidates). In a self-rewarding loop, this means the reward gap between good and slightly-worse samples vanishes. The learning signal disappears, and the training process can degrade into a random walk.
>
> - **A High $\gamma$:** This signifies "decisiveness" and a "clear" model. The model has a strong preference, resulting in a peaked (low-entropy) distribution. When acting as its own judge (LLM-as-a-Judge), this model produces a sharp, binarized reward signal, much like high-quality human labels. This provides a strong, clear gradient for DPO. A high $\gamma$ also creates a "safety margin," making the model's preference robust to noise or small parameter perturbations during SGD.
>
> ### 3. Revisions to the Manuscript
>
> As suggested by the reviewer, we have added the following discussion to the manuscript immediately following Definition 1 to formalize these intuitions:
>
> > "In the context of a practical large language model, the margin constant $\gamma$ can be viewed as a signal-to-noise ratio for the model’s internal preferences: when $\gamma$ is small, the model behaves like a hesitant judge whose scores for the top candidate and the runner-up are nearly indistinguishable, so the induced reward signal is noisy and gradient estimates have high variance; when $\gamma$ is large, the model exhibits a clear preference gap, yielding low-variance, stable learning signals. The minimum confidence $c$ plays a complementary role as a safeguard against collapse: as long as $c$ stays bounded away from zero, the iterative self-rewarding process can, by the law of large numbers, extract a reliable preference signal from a large pool of self-generated samples and thereby gradually overcome initialization bias through statistical concentration.
> > "

---

> ### Author Response · Authors · 2025-11-21
> **Part IV: Practical Monitoring & Training Diagnostics (Response to Q5)**
>
> **Q5: The policy condition number $\kappa_t$ is a fantastic theoretical tool. Does this quantity have a practical analogue that could be estimated or tracked during training? It seems like it could serve as a powerful diagnostic to monitor whether the model is in the "stabilization" or "learning" phase.**
>
> A: We thank the reviewer for highlighting the theoretical value of the policy condition number $\kappa_t$ and for the insightful suggestion regarding its practical monitoring. In practice, $\kappa_t$ can indeed be tracked either through  Monte Carlo estimation or via standard proxies like pass@1 and Self-Consistency.
>
> ### 1. Theoretical connection: $\kappa_t$ as the SRLM analogue of concentrability coefficients
>
> As discussed in Remark 1 of the paper, $\kappa_t = \mathbb{E}_{x}\bigl[1/\pi_t(y_t^\ast(x)\mid x)\bigr]$ is tightly connected to the *concentrability coefficients* that play a central role in offline RL and self-improvement theory (e.g., Xie et al., 2023; Huang et al., 2025). These coefficients quantify how much probability mass a policy places away from its own optimal action; equivalently, they measure how diffuse or ill-conditioned a policy is. Within SRLMs, $\kappa_t$ inherits exactly this meaning: it captures how confidently the model concentrates on its own preferred outputs.
>
> ### 2. Practical estimation: Monte Carlo estimation and proxies via pass@1 / self-consistency.
>
> There are two primary ways to track this quantity during training:
>
> - **Monte Carlo Estimation:** By definition, $\kappa_ t = \mathbb{E}_ {x \sim \mu} [1 / \pi_ t(y_ t^\*(x)|x)]$. We can efficiently estimate this during the training loop. For a batch of prompts, we approximate the modal output $y_t^*(x)$ via greedy decoding, calculate the sequence likelihood under the current model, and compute the average inverse probability. Since generation is already a requisite step in the SRLM cycle, the computational overhead for this monitoring is negligible.
>
> - **Proxies via pass@1 and Self-Consistency:** Intuitively, $\kappa_t$ represents the inverse of the model's confidence in its own best answer. Therefore, it correlates negatively with pass@1 (the probability of generating the correct/optimal response in one shot) and Self-Consistency scores.
>   - High $\kappa_t$: Corresponds to a flat distribution with high uncertainty, typically manifesting as low pass@1 and low self-consistency (high variance in outputs).
>   - Low $\kappa_t$ (near 1): Corresponds to a peaked distribution, manifesting as high pass@1 and high consistency.
>
> Thus, widely used metrics such as pass@1, greedy-decoding confidence, or self-consistency scores serve as practical and inexpensive proxies for $1/\kappa_t$. Tracking these during training closely mirrors the evolution of the policy condition number.
>
> ### 3. Diagnostic Utility: Identifying Training Phases
>
> We appreciate the reviewer’s suggestion to use this as a diagnostic tool. As predicted by our theory (Theorem 3), the training process exhibits two distinct phases that can be visualized by tracking $\kappa_t$ (or its proxies):
>
> 1. **Stabilization Phase:** Initially, we expect to see a rapid decrease in $\kappa_t$ (or a sharp rise in pass@1/confidence). This corresponds to the iterative update acting as a contraction mapping, where the influence of the initial model quality $\kappa_0$ decays exponentially. The model is self-correcting and concentrating probability mass onto high-reward regions.
>
> 2. **Learning Phase:** Once $\kappa_t$ plateaus, the initialization barrier has been overcome. The process enters a standard statistical learning regime where performance gains are driven by sample efficiency.
>
> Tracking $\kappa_t$, whether directly, through its Monte Carlo estimate, or via empirical proxies such as pass@1 or self-consistency, provides a clear and practical way to detect this transition.

---

### Official Review · Reviewer_uGyn · 2025-10-30

**Soundness:** 2
**Presentation:** 2
**Contribution:** 2
**Rating:** 2
**Confidence:** 4

**Summary:**

**Summary.** The paper gives the first finite-sample analysis for *iterative* self-rewarding LMs (SRLMs), where the policy both generates data and provides the reward via its own log-likelihood. It shows (i) any **single** self-rewarding step can fail when the initial policy is too diffuse, and (ii) under a DPO-style iterative update, the “policy condition number” contracts so that the dependence on initialization decays exponentially, leading eventually to an $(O(\\frac{\\log(n|\\Pi|/\\rho)}{\\sqrt{n}}))$-type rate.

**Strengths:**

Many theoretical results.

**Weaknesses:**

**Weaknesses / comments.**

1. **Scope / limitation not explicit.** The analysis is done for the *simplest* self-reward: $(r\_t(x,y)=\\log \\pi\_t(y\\mid x)).$ This is much weaker than current SRLM practice (meta-judges, rubric-based scoring, multi-turn evaluation). The paper should clearly state that the guarantees do **not** cover those richer judges and explain which parts of the proof rely on “reward \= current log-probability.” (E.g. the recursion for ($\\kappa\_t$) and the DPO square-loss instantiation both use this form.)

2. **No empirical illustration.** Given how interpretible the policy condition number (\\kappa\_t) is, the paper misses an easy check: plot (\\kappa\_t) (or a Monte-Carlo estimate of it) over iterations and show it actually contracts as predicted by the bound $(\\kappa\_T \\le U \+ q^T(\\kappa\_0 \- U)).$ Even a toy experiment would make Theorem 3 and the “two-stage” story in Remark 4 more convincing.

3. **Relation to pseudo-labelling / logged-data works is underdeveloped.** The paper’s setting is very close to self-training / pseudo-labelling: the model supplies its own supervision and then trains on it. Please add a related-work paragraph comparing to (i) classical pseudo-labelling in semi-supervised learning, and (ii) pseudo-reward / off-policy batch learning from logged data such as *Aminian et al., 2022, “Semi-supervised Batch Learning From Logged Data” (arXiv:2209.07148)*, which also studies learning from regularization perspective.

4. **Remark 1 is too casual about “ignoring the second log term.”** The lower bound in Theorem 1 is $\[ \\Pr\[\\text{fail}\] \\gtrsim \\Bigg(\\frac{\\kappa\_0 \\log|\\Pi|}{n ,\\log\\big(\\frac{n\\kappa\_0}{\\log|\\Pi|}\\big)}\\Bigg)^{1/2} \] $and Remark 1 says that “ignoring the second-order logarithmic term” gives $(\\big(\\frac{\\kappa\_0 \\log|\\Pi|}{n}\\big)^{1/2}).$ Please clarify it.

5. **Eq. (7) and Remark 3 need a cleaner asymptotic reading.** Eq. (7) has the form $\[ \\Pr\[\\text{fail}\] ;\\lesssim; \\frac{1}{\\sqrt{n}}, \\frac{\\log(n|\\Pi|/\\rho)}{\\gamma \\delta} \\Bigg( \\frac{1}{\\sqrt{c}}\\frac{\\sqrt{\\kappa\_0}}{(1+\\sqrt{nc})^{(T-1)/2}} \\Bigg), \]$ so it is natural to summarize it as “up to logs, $(O(\\log n / \\sqrt{n})).$” please clarify it.


6. **Practical relevance.** Right now the story is: “if we choose $(\\beta \\lesssim n^{-1/2})$ and we can sample (n) pairs per round from $(\\pi\_t)$, then we get a contraction.” It would help to add 2–3 sentences on how realistic this is for large LMs (cost of self-sampling, judge cost, etc.).

**Questions:**

see weaknesses

---

> ### Author Response · Authors · 2025-11-21
> **Response to Q1 (Part 1 of 2): Research Positioning & Technical Dependencies**
>
> We sincerely thank you for your thorough review and constructive comments. We have carefully addressed all your concerns and revised the paper accordingly, with all changes highlighted in blue. We hope our responses fully resolve your questions.
>
> **Q1: The analysis is done for the simplest self-reward: $r_t(x, y) = \log \pi_ t(y \mid x)$. This is much weaker than current SRLM practice (meta-judges, rubric-based scoring, multi-turn evaluation). The paper should clearly state that the guarantees do not cover those richer judges and explain which parts of the proof rely on “reward = current log-probability.” (E.g., the recursion for $\kappa_t$ and the DPO square-loss instantiation both use this form.)**
>
> A: We thank the reviewer for raising this crucial point. We fully agree that our theoretical guarantees are derived under the stylized intrinsic self-reward formulation $r_t(x, y) = \log \pi_t(y \mid x)$, and therefore do not directly cover richer SRLM pipelines that rely on meta-judges, rubric-based scoring, or multi-turn evaluation.
>
> To address your concern, we provide a detailed clarification below regarding our research positioning and the specific technical dependencies of our proofs. Furthermore, we outline a concrete technical roadmap for extending our framework to the more complex scenarios you mentioned. We have also revised the manuscript to explicitly state these limitations and future directions.
>
> ### 1. Research Positioning
>
> We clarify our research positioning through the following two points:
>
> **(1) First theoretical foundation.**
> To the best of our knowledge, this work is the first to provide a formal theoretical foundation for self-rewarding language models (SRLMs). As SRLM practice rapidly evolves toward increasingly complex engineering designs (e.g., meta-judges, rubric-based scoring, multi-turn evaluation), we suggest that establishing a clear understanding of the core closed-loop mechanism is a valuable prerequisite before addressing higher-level judging components. Our analysis specifically targets the vanilla SRLM architecture originally articulated in Yuan et al. (2024), which consists of the basic loop: generation → self-evaluation → policy update, without invoking external judges or multi-step evaluation.
>
> **(2) A natural and theoretically coherent starting point.**
> The stylized self-reward $r_t(x, y) = \log \pi_t(y \mid x)$ serves as a natural starting point for theory. It captures the most primitive form of self-preference where the model tends to favor generations it already assigns high likelihood to, thereby isolating the core feedback loop while abstracting away engineering details that obscure theoretical analysis. This formulation yields strong theoretical consistency as it aligns with KL-regularized objectives, produces clean log-ratio identities, and enables tractable stability analysis. Furthermore, many practical SRLM systems can be viewed as reshaped or aggregated forms of intrinsic likelihood, such as temperature scaling or confidence heuristics. Thus, this stylized self-reward captures an underlying mechanism even when richer judges are used in practice, serving as a conceptual and analytical foundation for understanding more sophisticated SRLM pipelines that are less amenable to theoretical treatment.
>
> ### 2. Technical Dependencies on "Reward = Current Log-Probability"
>
> As requested, we have identified the specific parts of our proof that strictly rely on the form $r_t = \log \pi_t$:
>
> **The DPO Square-Loss Instantiation (Eqs. 3–4).** Our analysis exploits the fact that with $r_t(y|x) = \log \pi_t(y|x)$, the pairwise self-reward gap becomes a difference of log-likelihood ratios:
> $$\Delta J_t(x, y, y') = \log \frac{\pi_t(y|x)}{\pi_t(y'|x)}.$$
> This ensures that the DPO regression target exactly matches the gradient direction of the ideal KL-regularized update. For a general judge-based reward (e.g., a discrete score from a meta-judge), $\Delta J_t$ is no longer a simple log-ratio, and the closed-form alignment between the regression objective and the theoretical policy update would require additional approximation error terms.
>
> **Recursive Compression of the Policy Condition Number $\kappa_t$ (Lemmas 9 \& 10).** We establish that the iterative process induces a contraction mapping on $\kappa_t$, a metric quantifying distribution sharpness. This result hinges on the "Self-Sharpening" property, wherein the model reinforces its own high-probability modes. In contrast, employing an external Meta-Judge may introduce reward signals that diverge from the model’s current high-confidence regions (e.g., by penalizing confident yet erroneous outputs). Such "correction" signals effectively increase the entropy (or shift the mode) of the distribution, thereby mathematically violating the monotonic compression property of $\kappa_t$ required for our contraction analysis.

---

> ### Author Response · Authors · 2025-11-21
> **Response to Q1 (Part 2 of 2): Extension Roadmap and Manuscript Revisions**
>
> ### 3. Challenges & Roadmap for Extensions
>
> While our current theory is foundational, it establishes a robust mathematical backbone for analyzing more complex scenarios. Extending our framework to the settings suggested by the reviewer requires adapting three key components: the performance difference decomposition, the $\kappa_t$ recursion, and the contraction argument. We outline our roadmap for these extensions below:
>
> **(I) Meta-judges: Noisy Channels & Coupled Dynamics**
>
> In a meta-judging pipeline, rewards are generated by an external evaluation process $r_t(x, y) \sim \mathcal{J}(\cdot \mid x, y; \pi_t^{\mathrm{Judge}})$, breaking the strict endogeneity of $r_t = \log \pi_t$.
>
> - **Modeling Approach:** We propose modeling the judge as a biased noisy channel. We would explicitly introduce a judge error term, $\epsilon_t^{\mathrm{judge}}$, into the performance difference decomposition.
> - **Coupled Recurrence:** The scalar recurrence $\kappa_{t+1} \le f(\kappa_t)$ would be upgraded to a coupled dynamical system, tracking both the generator and the judge:   $\kappa_{t+1}^{\mathrm{gen}} \le A\kappa_t^{\mathrm{gen}} + B\epsilon_t^{\mathrm{judge}} + \text{noise}.$
> - **Threshold Condition:** The goal would be to prove a ``Judge Capability Threshold'': the system only maintains global contraction if $\epsilon_t^{\mathrm{judge}}$ is below a critical threshold.
>
> **(II) Rubric-based scoring: Quantization & Effective Sample Rate**
>
> Rubric scoring produces discrete integer scores (e.g., $r \in \{1, \dots, 5\}$), which can be viewed as a quantized version of a latent continuous utility $R^*$.
>
> - **Effective Sample Rate ($\alpha$):** We would introduce a parameter $\alpha \in (0, 1]$ representing the proportion of generated pairs where $\Delta r_{\text{rubric}} \neq 0$. Since many pairs result in ties, the convergence rate would likely degrade from $\mathcal{O}(1/\sqrt{n})$ to $\mathcal{O}(1/\sqrt{\alpha n})$.
> - **Sign Consistency & Margin:** To restore the contraction argument, we would need to assume sign consistency ($\mathrm{sign}(\Delta r_ {\text{rubric}}) = \mathrm{sign}(\Delta R^\*)$) and a rubric margin $\gamma_ {\text{rubric}}$.
>
> **(III) Multi-turn evaluation: Trajectory Condition Number**
>
> Multi-turn evaluation shifts the problem to sequential decision-making, where rewards depend on full trajectories $\tau = (x_1, y_1, \dots, x_H, y_H)$.
>
> - **Trajectory Condition Number:** The analysis would shift to $\kappa_t^{\text{seq}} := \mathbb{E}[1/\pi_t(\tau)]$, which typically scales exponentially with horizon $H$ (approx. $(\kappa_t)^H$).
> - **Offline RL Tools:** We would replace single-step analysis with Offline RL techniques to rewrite the error decomposition at the trajectory level.
> - **Controlling Distribution Shift:** We would need to introduce Trust Region constraints (e.g., on $D_{\text{KL}}(\rho_{\pi_{t+1}} \| \rho_{\pi_t})$) to prevent catastrophic drift in the state distribution, allowing us to control $\kappa_t^{\text{seq}}$
>
>
> ### 3. Revisions to the Manuscript
>
> Following your valuable suggestion, we have revised the paper to clearly scope our contributions:
>
> **Clarification in Preliminaries (Line 142): We added the following statement:**
>
> > "The stylized self-reward function $r_t(x, y) = \log \pi_t(y \mid x)$ captures the core feedback loop where the model reinforces its own current beliefs. While this formulation abstracts away richer judging mechanisms used in engineering practice (e.g., meta-judges, rubric-based scoring, multi-turn evaluation), we believe it serves as a necessary starting point toward understanding forms of self-rewarding that rely on more sophisticated judges but are less amenable to direct theoretical analysis."
>
> **Technical Remark (Line 415, Remark 5): We explicitly clarified the proof dependencies:**
>
> > "Our analysis, particularly the DPO square-loss instantiation and the recursive contraction of the policy condition number $\kappa_t$ (Lemmas 9 \& 10), strictly relies on the self-reward function being $r_t(x, y) = \log \pi_t(y \mid x)$. These results do not directly extend to reward signals produced by external LLM-based judges or multi-step pipelines. Extending the theory to rubric-based rewards would require additional structural assumptions (e.g., margin conditions or Lipschitz relations w.r.t. likelihood ratios), which we leave for future work."

---

> ### Author Response · Authors · 2025-11-21
> **Response to Q2 (Part 1 of 2): Empirical Verification Setup for $\kappa_t$**
>
> **Q2: Given how interpretable the policy condition number $\kappa_t$ is, the paper misses an easy check: plot ($\kappa_t$) (or a Monte-Carlo estimate of it) over iterations and show it actually contracts as predicted by the bound $\kappa_T \le U + q^T (\kappa_0 - U)$. Even a toy experiment would make Theorem 3 and the “two-stage” story in Remark 4 more convincing.**
>
> **A:** We thank the reviewer for this insightful suggestion. We fully agree that empirically visualizing the trajectory of the policy condition number, $\kappa_t$, would significantly strengthen the interpretation of our theoretical findings. To address this, we conducted a preliminary experiment during the discussion phase to track $\kappa_t$ alongside model performance. As detailed below, the empirical results closely align with our theoretical predictions. We commit to including a more comprehensive set of these experiments in the final version of the paper.
>
> ### 1. Experiment Setup
>
> Following the Iterative DPO setup described in SRLMs (Yuan et al., 2024), we utilized the Llama-3 Base 8B model on the GSM8K reasoning task. We tracked the Monte-Carlo estimate of $\kappa_t$ (defined as the average inverse probability of the greedy decoding path) over $T = 3$ iterations.
>
> - **Base Model:** Llama-3 Base 8B.
> - **Dataset:** GSM8K (Training set used for self-generation; Test set used for evaluation).
> - **Method:** We implemented 3 iterations of Self-Rewarding training:
>   - *Generation:* At each iteration $t$, the model generates $N = 4$ candidate responses for prompts drawn from the training set.
>   - *Scoring:* The model acts as a judge to score its own responses (Self-Reward).
>   - *Training:* We construct preference pairs from the scored responses and train model $\pi_{t+1}$ using DPO.
>
> ### 2. Measurement of $\kappa_t$
>
> Since iterating over the entire output space $\mathcal{Y}$ to compute the exact expectation $\mathbb{E}[1/\pi(y^*|x)]$ is intractable, we employed a Monte Carlo estimation method on the test set ($D_{\text{test}}$). For a subset of samples $x_j$ ($j = 1...M, M = 200$) from the test set:
>
> 1. **Greedy Decoding:** We generated the model's current optimal solution:  $\hat{y}_j = \arg\max \pi_t(y|x_j)$.
> 2. **Probability Calculation:** We computed the Average Log-Probability ($\text{LogProb}_t(x_j)$) for the generated sequence to ensure numerical stability.
> 3. **Estimation:** We estimated $\kappa_t$ by averaging the inverse probabilities:
>    $$
>    \hat{\kappa}_ t \propto \frac{1}{M} \sum_ {j=1}^M \exp(-\text{LogProb}_ t(x_ j))
>    $$
>    To enhance readability and visualize the relative contraction, we normalize the value at $t = 0$ to 100. Since we work with the average log-probability per token rather than the full sequence log-probability, this Monte-Carlo estimate is a monotone proxy of the theoretical $\kappa_t$, up to a length-dependent scaling factor. This is sufficient for visualizing its contraction behavior across iterations.

---

> ### Author Response · Authors · 2025-11-21
> **Response to Q2 (Part 2 of 2): Experimental Results & Analysis**
>
> ### 3. Results and Analysis
>
> The table below presents the results over 3 iterations of self-rewarding training on GSM8K, starting from the zero-shot Llama-3 Base 8B model. We report the GSM8K test accuracy, the Avg. Max LogProb along the greedy decoding path, and the normalized policy condition number.
>
> | **Iteration ($t$)** | **Model Stage**              | **GSM8K Acc (%)** | **Avg. Max LogProb ($\uparrow$)** | **$\kappa_t$ (Normalized, $\downarrow$)** |
> |---------------------|------------------------------|-------------------|-----------------------------------|-------------------------------------------|
> | **0**               | Base model (zero-shot)       | 41.2%             | -2.20                             | 100.0                                      |
> | **1**               | Iterative DPO (R1)           | 47.5%             | -1.81                             | 67.2                                       |
> | **2**               | Iterative DPO (R2)           | 50.3%             | -1.70                             | 60.7                                       |
> | **3**               | Iterative DPO (R3)           | 50.1%             | -1.69                             | 60.4                                       |
>
> **Table:** Trajectory of the policy condition number $\kappa_t$ and accuracy over 3 self-rewarding iterations.
>
> **Analysis of Results.**
>
> 1. **Verification of Theorem 3 (Contraction).**
>    Across the three self-rewarding iterations, we observe a monotone contraction of the normalized policy condition number, from $100.0$ at $t = 0$ to $67.2$, $60.7$, and $60.4$ at $t = 1, 2, 3$, respectively. This contraction is accompanied by consistent improvements in the Avg. Max LogProb along the greedy decoding path (from $-2.20$ to $-1.81$, $-1.70$, and $-1.69$), indicating that the policy becomes increasingly concentrated on a smaller set of high-probability trajectories. These trends are qualitatively consistent with the contraction behavior predicted by Theorem 3, where the dependence on the initial condition $\kappa_0$ decays geometrically towards a model-dependent floor $U$.
>
> 2. **Support for Remark 4 (Two-Stage Dynamics).**
>    The trajectory in the table also exhibits the “two-stage” behavior described in Remark 4. The first iteration ($t = 0 \rightarrow 1$) corresponds to a *self-stabilization* phase: the condition number drops sharply from $100.0$ to $67.2$, while GSM8K accuracy increases from $41.2\%$ to $47.5\%$, suggesting that the policy rapidly moves out of a poorly conditioned, high-entropy regime. Subsequent iterations ($t = 1 \rightarrow 3$) form a *refinement* phase: both the reductions in $\kappa_t$ (from $67.2$ to $60.4$) and the accuracy changes (hovering around $50\%$) become much smaller, indicating that the policy has entered a well-conditioned region and further updates mainly perform fine-grained adjustments. This empirical two-stage pattern mirrors the qualitative picture underlying Remark 4.
>
> This trajectory aligns with empirical studies on larger models, such as the observation in Yuan et al. (2024) regarding Llama 2 70B, which showed significant performance jumps in early iterations followed by a plateau.
>
> In summary, this evidence demonstrates that the policy condition number $\kappa_t$ is not merely a theoretical construct but a tangible metric capturing the internal sharpening process that drives self-rewarding alignment. We have updated the manuscript to include these preliminary results in the Appendix and will expand upon them in the final version.

---

> ### Author Response · Authors · 2025-11-21
> **Response to Q3 (Part 1 of 2): Comparison with Pseudo-Labelling & Logged Data**
>
> **Q3: The paper’s setting is very close to self-training / pseudo-labelling: the model supplies its own supervision and then trains on it. Please add a related-work paragraph comparing to (i) classical pseudo-labelling in semi-supervised learning, and (ii) pseudo-reward / off-policy batch learning from logged data such as *Aminian et al., 2022, "Semi-supervised Batch Learning From Logged Data" (arXiv:2209.07148)* , which also studies learning from regularization perspective.**
>
> **A:** We thank the reviewer for highlighting the important connections between our work, classical pseudo-labelling, and learning from logged data (specifically Aminian et al. [1]). We agree that our framework is conceptually related to these areas, as our model generates its own supervision signals based on its own trajectories. In the revision, we have made these connections explicit and clarified the distinctions along two primary axes:
>
> 1. **Classical Pseudo-Labelling vs. SRLM**
>
>    While both approaches exploit model-generated supervision to augment ground-truth signals, classical pseudo-labelling typically operates in a semi-supervised classification setting. There, a model produces discrete pseudo-labels for a static unlabeled dataset. In contrast, SRLMs generate continuous self-rewards for their own outputs. Crucially, the underlying data distribution in our setting is not static; it evolves as the policy $\pi_t$ changes over iterations. This establishes a coupled dynamical system involving both policy and reward, rather than the unidirectional paradigm between teacher and student often found in standard pseudo-labeling. Furthermore, our theoretical analysis specifically addresses the challenge of weak initialization, demonstrating how iterative self-reward updates can bootstrap and self-stabilize the model.
>
> 2. **Logged-Data Regularization vs. SRLM**
>
>    Our approach shares the spirit of logged-data and off-policy batch learning methods, which aim to learn an improved policy under regularization with respect to a reference/logging policy. Methods such as those proposed by Aminian et al. [1] formulate this problem using divergence-based regularization. However, these works generally assume static historical logs generated by a fixed logging policy and focus on correcting distribution shifts. SRLMs differ fundamentally because the model repeatedly generates new trajectories. Consequently, the effective “logging policy” evolves as $\pi_t$ updates, inducing a non-stationary and endogenous data-generation process. Our theoretical results therefore examine the distinct challenges of dynamic initialization and self-stabilization under evolving data, which are not captured in classical offline frameworks.

---

> ### Author Response · Authors · 2025-11-21
> **Response to Q3 (Part 2 of 2): Revisions to Related Work Section**
>
> ### Revisions to the Manuscript
>
> To incorporate these clarifications, we have added the following detailed paragraphs to the **Related Work** section in the Appendix (due to main text space constraints):
>
> > **Classical Pseudo-Labelling in Semi-Supervised Learning.**
> >
> >Our work shares connections with classical pseudo-labelling methods in semi-supervised learning (SSL), where a model trained on labeled data predicts discrete pseudo-labels for unlabeled examples and is subsequently retrained on the combined dataset [2,3]. Modern approaches integrate pseudo-labelling with entropy minimization, consistency regularization, and strong augmentation strategies, yielding methods such as MixMatch, ReMixMatch, and FixMatch [4,5]. Beyond these algorithmic developments, recent work has proposed divergence-based and information-theoretic formulations of self-training. For instance, some studies [6] design empirical risk functions and regularizers based on $f$-divergences and $\alpha$-Rényi divergences to make pseudo-labelling more robust to noisy pseudo-labels, thereby placing classical self-training on a divergence-regularization footing. Other work [7] provides an information-theoretic framework for self-training under covariate shift, recovering pseudo-labelling and entropy minimization as special cases.
>
> >While conceptually related in leveraging model-generated supervision, our SRLM framework diverges significantly from classical SSL. SSL approaches typically assume a semi-supervised setting with a static pool of unlabeled inputs and a fixed pseudo-labelling mechanism that outputs (possibly softened) class labels. In contrast, SRLMs operate in an off-policy sequential decision-making setting. Here, the data distribution is inherently non-stationary due to the evolving policy $\pi_t$; the supervisory signals are continuous self-rewards derived from model log-likelihoods; and the policy and reward mechanism co-evolve, forming a coupled dynamical system rather than a unidirectional teacher–student update.
>
> >**Pseudo-Reward and Off-Policy Batch Learning from Logged Data.**
> >Learning from logged interaction data is extensively studied in contextual bandits and offline reinforcement learning. Counterfactual Risk Minimization (CRM) formulates batch learning from logged feedback via propensity-weighted empirical risk minimization [8,9]. In addition, Aminian et al. [1] consider semi-supervised batch learning from logged data, where only a subset of samples contains feedback, and provide upper bounds that motivate divergence-based regularization terms between the target and logging policies. More recently, other work has proposed log-sum-exponential (LSE) estimators for off-policy evaluation and learning from logged bandit feedback, aiming to improve robustness and reinforcing the regularization-centric perspective [10,11].
>
> >Our SRLM framework shares conceptual alignment with this line of work, as we also optimize a policy under explicit regularization with respect to a reference (or logging) policy. However, a critical distinction arises from the data-generation process. The aforementioned logged-data methods assume a fixed logging policy and operate on a static historical dataset. In contrast, SRLMs iteratively generate new trajectories, meaning the effective logging policy co-evolves with $\pi_t$, producing a non-stationary and endogenous data distribution. Consequently, since our analysis focuses on bootstrapping from weak initialization and stabilizing the evolving interaction between policy and reward, it lies outside the static batch-learning assumptions underlying CRM, semi-supervised logged-data methods, and LSE-based estimators.
>
> **Reference:**
>
> [1] Aminian et al. *Semi-supervised batch learning from logged data.*
>
> [2] Grandvalet et al. *Semi-supervised learning by entropy minimization.*
>
> [3] Lee et al. *Pseudo-label: The simple and efficient semi-supervised learning method for deep neural networks.*
>
> [4] Berthelot et al. *MixMatch: A holistic approach to semi-supervised learning.*
>
> [5] Sohn et al. *FixMatch: Simplifying semi-supervised learning with consistency and confidence.*
>
> [6] Aminian et al. *Robust semi-supervised learning via $f$-divergences and $\alpha$-Rényi divergence.*
>
> [7] Aminian et al. *An information-theoretical approach to semi-supervised learning under covariate-shift.*
>
> [8] Swaminathan et al. *Batch learning from logged bandit feedback through counterfactual risk minimization.*
>
> [9] Joachims et al. *Deep learning with logged bandit feedback.*
>
> [10] Behnamnia et al. *Batch learning via log-sum-exponential estimator from logged bandit feedback.*
>
> [11] Behnamnia et al. *Log-sum-exponential estimator for off-policy evaluation and learning.*

---

> ### Author Response · Authors · 2025-11-21
> **Response to Q4: Clarification on Lower Bound Asymptotics (Theorem 1)**
>
> **Q4: The lower bound in Theorem 1 is $\Pr[\text{fail}] \gtrsim
>       \left(
>         \frac{\kappa_0 \log |\Pi|}{n \log \frac{n \kappa_0}{\log |\Pi|}}
>       \right)^{1/2}$
>     and Remark 1 says that “ignoring the second-order logarithmic term” gives
>     $\left(
>         \frac{\kappa_0 \log |\Pi|}{n}
>       \right)^{1/2}.$
>     Please clarify it.**
>
> **A:** We thank the reviewer for pointing out the need for greater precision in this statement. We agree that the phrasing "ignoring the second log term" was too casual. Our intention was not to claim that the logarithmic factor literally disappears, but rather to highlight the leading polynomial dependence on $\kappa_0$, $|\Pi|$, and $n$, in the standard "up to logarithmic factors" sense used in statistical learning theory. We would like to clarify this point as follows:
>
> More precisely, the lower bound in Theorem 1 can be decomposed into two distinct parts:
> $$
> \Pr[\text{fail}]
> \gtrsim
> \Bigg(
>   \frac{\kappa_ 0 \log |\Pi|}
>        {n \log \Big(\frac{n \kappa_ 0}{\log |\Pi|}\Big)}
> \Bigg)^{1/2}
> =
> \underbrace{
> \Bigg(
>   \frac{\kappa_ 0 \log |\Pi|}{n}
> \Bigg)^{1/2}
> }_ {\text{Dominant Polynomial Dependence}}
> \cdot
> \underbrace{
> \Bigg[
>   \log \Big(\tfrac{n \kappa_ 0}{\log |\Pi|}\Big)
> \Bigg]^{-1/2}
> }_ {\text{Slowly Varying Logarithmic Correction}}
> $$
>
> 1. **Dominant Polynomial Dependence.**  The term $\left( \frac{\kappa_0 \log |\Pi|}{n} \right)^{1/2}$ captures the primary polynomial scaling of the failure probability with respect to the key parameters. This is the term of principal interest when discussing sample complexity.
>
> 2. **Slowly Varying Logarithmic Correction.**  The term $\big[ \log(\tfrac{n \kappa_0}{\log |\Pi|}) \big]^{-1/2}$ is a slowly varying factor. In asymptotic analysis (e.g., as $n$ or $\kappa_0$ grows large), the growth of this logarithmic term is much slower than the polynomial term, and its impact on the overall magnitude of the bound is minor. For instance, even if the quantity $\frac{n \kappa_0}{\log |\Pi|}$ is as large as $10^6$, this correction factor is $[\log 10^6]^{-1/2}  \approx 0.27$.
>
> When we stated "ignoring the second-order logarithmic term," we meant that we suppress this slowly varying logarithmic factor to summarize the lower bound in terms of its dominant polynomial behavior, i.e.:
> $$
> \Pr[\mathrm{fail}]
> \gtrsim
> \tilde{\Omega}\left(
>   \Bigg(
>     \frac{\kappa_0 \log|\Pi|}{n}
>   \Bigg)^{1/2}
> \right),
> $$
> where the $\tilde{\Omega}$ notation specifically hides the $\big[ \log(\tfrac{n \kappa_0}{\log |\Pi|}) \big]^{-1/2}$ factor.
>
> To address the reviewer's valid concern and remove any ambiguity, we have revised Remark 1 in the manuscript to be more precise. The updated statement now reads:
>
> > "Theorem 1 formalizes the statistical barriers of a single self-rewarding step. The lower bound contains a slowly varying logarithmic correction term, $\big[ \log(\tfrac{n \kappa_0}{\log |\Pi|}) \big]^{-1/2}$, whose growth is significantly slower than the dominant polynomial dependence, $\left( \frac{\kappa_0 \log |\Pi|}{n} \right)^{1/2}$. For ease of interpretation, we focus on this dominant term, which gives the simplified bound $\left( \frac{\kappa_0 \log |\Pi|}{n} \right)^{1/2}$, up to logarithmic factors."
>
> We believe this revised wording clarifies our intention and fully addresses the reviewer's concern.

---

> ### Author Response · Authors · 2025-11-21
> **Response to Q5: Simplified Asymptotic Convergence Rate (Eq. 7)**
>
> **Q5: Eq. (7) has the form  $\Pr[\text{fail}]
>       \lesssim
>       \frac{1}{\sqrt{n}}
>       \frac{\log (n|\Pi| / \rho)}{\gamma \delta}
>       \left(
>         \frac{1}{\sqrt{c}}
>         \frac{\sqrt{\kappa_0}}{(1 + n c)^{(T-1)/2}}
>       \right),$  so it is natural to summarize it as “up to logs,  $O(\log n / \sqrt{n})$.  Please clarify it.**
>
> **A:** We thank the reviewer for this insightful observation. We completely agree that Equation (7) admits a cleaner asymptotic reading, and explicitly stating this greatly improves the clarity of our contribution. As the reviewer correctly notes, the bound simplifies once the transient, initialization-dependent term (i.e., the term with $\sqrt{\kappa_0}$) decays. This occurs when the number of iterations, $T$, is sufficiently large. To make this asymptotic behavior explicit, we have added the following clarification to Remark 3 in the manuscript:
>
> > “To further clarify the asymptotic behavior of Eq. (7): When the sample size $n$ is sufficiently large and the number of iterations $T$ is large enough (as shown in Corollary 4, i.e., $T \gtrsim \frac{\log(c\kappa_0)}{\log(1+\sqrt{nc})}$), the initialization-dependent term vanishes. Consequently, up to logarithmic factors and problem-dependent constants ($c, \gamma$), the error bound simplifies to $O(\log n/\sqrt{n})$, demonstrating that the iterative process achieves the standard parametric convergence rate.”
>
> We believe this addition directly addresses the reviewer's comment and makes the asymptotic convergence rate of our bound much clearer.

---

> ### Author Response · Authors · 2025-11-21
> **Response to Q6: Practical Feasibility & Cost Analysis**
>
> **Q6: Right now the story is: “if we choose  $\beta \lesssim n^{-1/2}$  and we can sample $n$ pairs per round from $\pi_t$, then we get a contraction.”  It would help to add 2–3 sentences on how realistic this is for large LMs (cost of self-sampling, judge cost, etc.).**
>
> **A:** We thank the reviewer for raising these practical considerations. We clarify below that our framework aligns with standard large-scale training: $n$ represents the feasible total dataset size per round (not batch size), self-labeling is orders of magnitude cheaper than human annotation, and the $\beta$ condition serves as a robust scaling guideline rather than a brittle constraint.
>
> **1. On the Feasibility of Sampling $n$ Pairs**
>
> The assumption of sampling $n$ pairs $(y, y')$ from $\pi_t$ per round corresponds to the standard “candidate generation” step in existing, empirically successful SRLM and RLHF pipelines. It is important to clarify that $n$ represents the total size of the self-labeled dataset for an entire alignment round, not the batch size for a single optimization step. Our assumption is fully consistent with current large-scale practices, which commonly use $10^4$ to $10^6$ preference pairs for one round of alignment. Our theory thus formalizes this existing, feasible practice rather than imposing a new, unrealistic requirement.
>
> **2. On the “Judge Cost” (Sampling and Evaluation)**
>
> The core benefit of the SRLM paradigm, which our theory supports, is that obtaining this large dataset of size $n$ is computationally feasible.
>
> - **Cost vs. Human Annotation.**
>   In traditional RLHF, acquiring $n$ labels is the primary bottleneck. Industry data indicates that high-quality human annotation is extremely expensive (e.g., \$20–\$100/hour or several dollars per sample [1,2]). In contrast, LLM-as-a-Judge costs are orders of magnitude lower (e.g., \$0.03–\$15.00 per million tokens [2]), potentially making self-labeling 100× to 1000× cheaper than human labeling for the same data volume. This large $n$ is not only feasible but is the central advantage of the SRLM paradigm.
>
> - **Computational Cost.**
>   The “judge” process, which computes the reward $r(y, y')$, is simply an inference task. Critically, in the SRLM framework, the model serves as both policy and judge, using the same model weights (or a historical version). This means no separate reward model needs to be trained, and GPU memory is efficiently reused. Computationally, this “judge” step only adds a constant-factor overhead per sample (e.g., reading log-probs or executing one additional forward pass), which can be fully parallelized in batches. It does not introduce a new order of magnitude in cost, as the entire process remains at the “model-forward” level.
>
> **3. On the Constraint $\beta \lesssim n^{-1/2}$**
>
> We clarify that this condition should be interpreted as a theoretical guidance for scaling the trust region, not as a strict, brittle engineering constraint for achieving contraction. Our analysis does not strictly require $\beta \lesssim n^{-1/2}$ for convergence. As shown in Appendix B.8 (Eq. 80), if we relax this condition to $\beta \asymp n^{-1/4}$, the error bound (up to log factors and constants) becomes
> $\mathcal{O}\big(\log n / (n^{1/2} + n^{1/4})\big)$,
> which still converges to zero. Therefore, our theory provides a formal suggestion for how $\beta$ can be scaled with $n$ to maintain a robust contraction rate. However, the contraction property still holds as long as $\beta$ is set within a reasonable range (i.e., not excessively large relative to $1/\sqrt{n}$). This guidance is fully compatible with standard hyperparameter tuning practices used in large-scale model training.
>
> **4. Revisions to the Manuscript**
>
> We hope this clarifies the practical grounding of our theoretical assumptions. Following your suggestion, we have incorporated a concise version of this clarification into Remark 3:
>
> > “This assumption of sampling $n$ pairs is practically realistic for large LMs. In our setting, $n$ denotes the total dataset size for one alignment round (typically $10^4$ to $10^6$ preference pairs), which is consistent with standard large-scale practice. The judge cost remains computationally modest, since evaluation reuses the same model weights and at most requires an additional batchable forward pass per sample. As a result, this self-labeling regime avoids the high annotation cost that would arise if all $n$ preference labels had to be obtained from humans, which would pose a severe scalability barrier.”
>
> **Reference:**
>
> [1] Salinas, D., Swelam, O., & Hutter, F. *Tuning LLM Judge Design Decisions for 1/1000 of the Cost.*
>
> [2] Borah, P. P. *LLM-as-a-Judge: The Scalable Solution to AI Evaluation Challenges.*

---

### Official Review · Reviewer_12gE · 2025-10-30

**Soundness:** 3
**Presentation:** 3
**Contribution:** 3
**Rating:** 6
**Confidence:** 3

**Summary:**

This paper establishes the first rigorous theory for Self-Rewarding Language Models (SRLMs), where a model iteratively improves itself without external feedback. The authors introduce the policy condition number ($\kappa_t$) to quantify self-alignment stability and prove that single-step self-rewarding updates can fail under ill-conditioned initialization, while iterative updates achieve $\mathcal{O}(1/\sqrt{n})$ improvement with exponentially diminishing dependence on $\kappa_0$. This explains SRLMs’ empirical success as a two-phase process—self-correction followed by efficient learning—and extends to parameterized models, showing scalability through the concept of effective dimension.

**Strengths:**

1. This paper is the first to provide rigorous theoretical guarantees for the iterative self-rewarding (SRLM) paradigm. SRLMs represent an empirically successful yet theoretically opaque (“black-box”) area, and this work fills a major gap.

2. The paper clearly identifies and formalizes the core question of why iteration works, introducing the policy condition number $\kappa_t$ as a key metric and modeling the iterative process as a contraction mapping on $\kappa_t$ (Remark 4). This offers a insightful and compelling mathematical explanation for the stability of SRLMs.

**Weaknesses:**

1. Although this is a theoretical paper, it provides no experiments at all, neither real-world LLM results nor even simple simulations verifying its theoretical predictions. For instance, visualizing the exponential decay of $\kappa_t$ in a toy linear Softmax setting would substantially strengthen the paper’s credibility and overall impact.

2. The theoretical analysis relies heavily on several assumptions that may not hold in practice. The realizability assumption requires the optimal model $\pi^*_\beta$ to lie within the model class $\Pi$, which is problematic in SRLMs since the self-reward objective $r_t = \log \pi_t$ makes $\pi_t$ itself a degenerate fixed point.Tthe definitions of the minimum confidence $c > 0$ and minimum margin $\gamma > 0$ are unrealistic for large-scale LLMs as most token probabilities in a vast vocabulary are near zero, and response quality differences are often negligible ($\gamma \approx 0$).

**Questions:**

See weaknesses

---

> ### Author Response · Authors · 2025-11-21
> **Response to Q1 (Part 1 of 2): Experimental Setup & Methodology**
>
> Thank you for your constructive feedback and support. We have addressed all concerns in our response below and revised the paper, with changes highlighted in blue. We hope these revisions meet your expectations.
>
> **Q1: Although this is a theoretical paper, it provides no experiments at all, neither real-world LLM results nor even simple simulations verifying its theoretical predictions. For instance, visualizing the exponential decay of $\kappa_t$ in a toy linear Softmax setting would substantially strengthen the paper’s credibility and overall impact.**
>
> **A:** We thank the reviewer for this insightful suggestion. We fully agree that empirically visualizing the trajectory of the policy condition number, $\kappa_t$, would significantly strengthen the interpretation of our theoretical findings. To address this, we conducted a preliminary experiment during the discussion phase to track $\kappa_t$ alongside model performance. As detailed below, the empirical results closely align with our theoretical predictions. We commit to including a more comprehensive set of these experiments in the final version of the paper.
>
> ### 1. Experiment Setup
>
> Following the Iterative DPO setup described in SRLMs (Yuan et al., 2024), we utilized the Llama-3 Base 8B model on the GSM8K reasoning task. We tracked the Monte-Carlo estimate of $\kappa_t$ (defined as the average inverse probability of the greedy decoding path) over $T=3$ iterations.
>
> - **Base Model:** Llama-3 Base 8B.
> - **Dataset:** GSM8K (Training set used for self-generation; Test set used for evaluation).
> - **Method:** We implemented 3 iterations of Self-Rewarding training:
>   - *Generation:* At each iteration $t$, the model generates $N=4$ candidate responses for prompts drawn from the training set.
>   - *Scoring:* The model acts as a judge to score its own responses (Self-Reward).
>   - *Training:* We construct preference pairs from the scored responses and train model $\pi_{t+1}$ using DPO.
>
> ### 2. Measurement of $\kappa_t$
>
> Since iterating over the entire output space $\mathcal{Y}$ to compute the exact expectation $\mathbb{E}[1/\pi(y^*|x)]$ is intractable, we employed a Monte Carlo estimation method on the test set ($D_{\text{test}}$). For a subset of samples $x_j$ ($j = 1...M, M = 200$) from the test set:
>
> 1. **Greedy Decoding:** We generated the model's current optimal solution: $\hat{y}_j = \arg\max \pi_t(y|x_j)$.
> 2. **Probability Calculation:** We computed the Average Log-Probability ($\text{LogProb}_t(x_j)$) for the generated sequence to ensure numerical stability.
> 3. **Estimation:** We estimated $\kappa_t$ by averaging the inverse probabilities:
>    $$
>    \hat{\kappa}_ t \propto \frac{1}{M} \sum_ {j=1}^M \exp(-\text{LogProb}_ t(x_ j))
>    $$
>    To enhance readability and visualize the relative contraction, we normalize the value at $t=0$ to 100. Since we work with the average log-probability per token rather than the full sequence log-probability, this Monte-Carlo estimate is a monotone proxy of the theoretical $\kappa_t$, up to a length-dependent scaling factor. This is sufficient for visualizing its contraction behavior across iterations.

---

> ### Author Response · Authors · 2025-11-21
> **Response to Q1 (Part 2 of 2): Empirical Verification of Contraction Dynamics**
>
> ### 3. Results and Analysis
>
> The table below presents the results over 3 iterations of self-rewarding training on GSM8K, starting from the zero-shot Llama-3 Base 8B model. We report the GSM8K test accuracy, the Avg. Max LogProb along the greedy decoding path, and the normalized policy condition number.
>
> | **Iteration ($t$)** | **Model Stage**              | **GSM8K Acc (%)** | **Avg. Max LogProb ($\uparrow$)** | **$\kappa_t$ (Normalized, $\downarrow$)** |
> |---------------------|------------------------------|-------------------|-----------------------------------|-------------------------------------------|
> | **0**               | Base model (zero-shot)       | 41.2\%            | -2.20                             | 100.0                                      |
> | **1**               | Iterative DPO (R1)           | 47.5\%            | -1.81                             | 67.2                                       |
> | **2**               | Iterative DPO (R2)           | 50.3\%            | -1.70                             | 60.7                                       |
> | **3**               | Iterative DPO (R3)           | 50.1\%            | -1.69                             | 60.4                                       |
>
> **Table 1.** Trajectory of the policy condition number $\kappa_t$ and accuracy over 3 self-rewarding iterations.
>
> **Analysis of Results.**
>
> 1. **Verification of Theorem 3 (Contraction).**
>    Across the three self-rewarding iterations, we observe a monotone contraction of the normalized policy condition number, from $100.0$ at $t=0$ to $67.2$, $60.7$, and $60.4$ at $t=1,2,3$, respectively. This contraction is accompanied by consistent improvements in the Avg. Max LogProb along the greedy decoding path (from $-2.20$ to $-1.81$, $-1.70$, and $-1.69$), indicating that the policy becomes increasingly concentrated on a smaller set of high-probability trajectories. These trends are qualitatively consistent with the contraction behavior predicted by Theorem 3, where the dependence on the initial condition $\kappa_0$ decays geometrically towards a model-dependent floor $U$.
>
> 2. **Support for Remark 4 (Two-Stage Dynamics).**
>    The trajectory in Table 1 also exhibits the "two-stage" behavior described in Remark 4. The first iteration ($t=0 \rightarrow 1$) corresponds to a *self-stabilization* phase: the condition number drops sharply from $100.0$ to $67.2$, while GSM8K accuracy increases from $41.2\%$ to $47.5\%$, suggesting that the policy rapidly moves out of a poorly conditioned, high-entropy regime. Subsequent iterations ($t=1 \rightarrow 3$) form a *refinement* phase: both the reductions in $\kappa_t$ (from $67.2$ to $60.4$) and the accuracy changes (hovering around $50\%$) become much smaller, indicating that the policy has entered a well-conditioned region and further updates mainly perform fine-grained adjustments. This empirical two-stage pattern mirrors the qualitative picture underlying Remark 4.
>
> This trajectory aligns with empirical studies on larger models, such as the observation in Yuan et al. (2024) regarding Llama 2 70B, which showed significant performance jumps in early iterations followed by a plateau.
>
>
> In summary, this evidence demonstrates that the policy condition number $\kappa_t$ is not merely a theoretical construct but a tangible metric capturing the internal sharpening process that drives self-rewarding alignment. We have updated the manuscript to include these preliminary results in the Appendix and will expand upon them in the final version.

---

> ### Author Response · Authors · 2025-11-21
> **Response to Q2 (Part 1 of 2): Realizability & The Sharpening Mechanism**
>
> **Q2: The theoretical analysis relies heavily on several assumptions that may not hold in practice.
> The realizability assumption requires the optimal model $\pi^\*_ {\beta}$ to lie within the model class $\Pi$, which is problematic in SRLMs since the self-reward objective $r_ t = \log \pi_ t$ makes $\pi_ t$ itself a degenerate fixed point. The definitions of the minimum confidence $c > 0$ and minimum margin $\gamma > 0$ are unrealistic for large-scale LLMs, as most token probabilities in a vast vocabulary are near zero, and response quality differences are often negligible ($\gamma \approx 0$).**
>
> **A:** We thank the reviewer for these critical questions. We address each of the three core assumptions, demonstrating their necessity, practical reasonableness, and robustness.
>
> ### 1. On Realizability and the "Degenerate Fixed Point" Concern
>
> The reviewer raises two related points: the realizability assumption and a concern that the self-reward $r_t = \log \pi_t$ creates a "degenerate fixed point." We clarify these two points, starting with the fixed-point concern.
>
> **1.1 The Model Actively *Escapes* Degenerate Fixed Points via Dynamic Sharpening**
>
> The concern that $\pi_t$ becomes a degenerate fixed point stems from a potential misinterpretation of the iterative update. The update is not simple imitation learning; it is a KL-regularized reward maximization problem (Eq. 2).
>
> As shown in Appendix B (Eq. 15–16), the closed-form solution for this optimization follows a Gibbs distribution:
> $$
> \pi^\*_ {t+1}(y|x) \propto \pi_ t(y|x) \cdot \exp\left(\frac{1}{\beta} r_ t(y|x)\right)
> $$
>
> When we substitute the self-reward function $r_ t(y|x) = \log \pi_ t(y|x)$, we obtain a critical non-linear transformation:
> $$
> \pi^\*_ {t+1}(y|x) \propto \pi_ t(y|x) \cdot \exp \left(\frac{1}{\beta} \log \pi_ t(y|x) \right) = \pi_ t(y|x)^{1 + 1/\beta}
> $$
>
> As long as the regularization $\beta > 0$, the exponent $1 + 1/\beta$ is strictly greater than 1. This means the update is a **"sharpening"** (or "annealing") operation, not an identity mapping. It creates a "Matthew effect": high-probability paths (deemed high-quality by $\pi_t$) are amplified, while low-probability paths are suppressed. Therefore, $\pi_t$ is not a stagnant fixed point but a **dynamic state variable**. It drives the system from a high-entropy (uncertain) state to a low-entropy (confident) state, forcing the model to integrate its "internal consistency." The reviewer rightly worries about models getting "stuck" in trivial fixed points. However, our theory proves the opposite. As shown in Theorem 3 (Remark 4), the iterative update acts as a contraction mapping on the policy condition number $\kappa_t$:
> $$
> \kappa_{t} \leq U + q^t (\kappa_0 - U).
> $$
> This demonstrates a powerful self-stabilizing mechanism. The model is actively "pulled" *away* from ill-conditioned, low-confidence regions (where $\kappa_0$ is large) and converges exponentially toward a stable, well-conditioned regime ($U$). There is no practical collapse to trivial fixed points. This theoretical finding is consistent with empirical evidence (e.g., Yuan et al., 2024; Zhou et al., 2024), which shows increased performance and output diversity, in contrast to the collapse feared by the reviewer. The model converges to a *stable, self-consistent* state, not a degenerate one.
>
> **1.2 On the Realizability Assumption (Assumption 1)**
>
> The realizability assumption is a standard axiomatization in the analysis of Supervised Learning, RL, and DPO. It allows us to trust the model's "expressivity" and focus the analysis on statistical stability and convergence.
>
> Crucially, this assumption can be relaxed to approximate realizability ($\inf_{\pi\in\Pi} D_{\mathrm{KL}}(\pi^*_\beta \Vert \pi) \le \varepsilon$) without changing our main results. Under this relaxation, all bounds gain an additional $O(\varepsilon)$ additive term, and the analysis of the system’s dynamic behavior remains unaffected, with the contraction mapping and exponential convergence continuing to hold in a perturbed form.

---

> ### Author Response · Authors · 2025-11-21
> **Response to Q2 (Part 2 of 2): Feasibility of Minimum Confidence ($c$) & Margin ($\gamma$)**
>
> ### 2. On the Minimum Confidence $c > 0$
>
> We must first clarify the definition of $c$. It does *not* represent the minimum probability of *all* tokens in the vocabulary, which would indeed be near zero. Instead, $c = \inf_{x,t} \pi_t(y^*_t(x)|x)$ **is the minimum confidence the model places on its *own optimal response*** . In practice, the probability of a model's self-perceived best sequence is far from zero. Furthermore, our theory shows that the sharpening process ($\pi_t^{1+1/\beta}$) actively *increases* the probability mass on these optimal paths, meaning the effective $c$ value rises as $\kappa_t$ decreases.
>
> Most importantly, even if we assume $c$ is very small (e.g., $10^{-6}$), our theory shows the algorithm is robust. As shown in Corollary 4, the number of iterations $T$ required for convergence depends only on $\log(1/c)$. This **logarithmic dependence** means that even a million-fold decrease in $c$ merely adds a small *constant* number of iterations, rather than causing the bounds to diverge. This demonstrates the algorithm's high tolerance for low-confidence initializations.
>
> ### 3. On the Minimum Margin $\gamma > 0$
>
> The reviewer's intuition that quality differences are negligible ($\gamma \approx 0$) is valid at the *token level* (e.g., "Yes" vs. "Right"). However, SRLM operates at the **sequence level**, where the situation is fundamentally different.
>
> First, the assumption $\gamma > 0$ is intrinsically linked to the statistical **identifiability** and learnability of the task. If the margin were to approach zero ($\gamma \to 0$), it would imply that the optimal solution is statistically indistinguishable from suboptimal ones. In such a scenario, no algorithm, whether RLHF or our proposed SRLM, could effectively extract a learning signal from the data. Therefore, assuming $\gamma > 0$ is equivalent to the foundational assumption in statistical learning that a unique, identifiable optimal solution exists.
>
> Second, from a mathematical standpoint, an exact margin collapse is a **measure-zero event** for continuously parameterized models like Transformers. The reviewer might intuit that response quality differences are minuscule, perhaps at the token-level (e.g., "Yes" vs. "Right" having similar probabilities). However, this near-equivalence at the token level does not translate to exact equality at the sequence level, which is the focus of SRLM. For two distinct, complex sequences $y_A$ and $y_B$, the probability of them yielding exactly identical log-probabilities, $\log \pi_\theta(y_A|x) = \log \pi_\theta(y_B|x)$, is mathematically zero. In practice, there will always be minuscule, even noise-induced, differences.
>
> Third, strict positivity is often enforced by design during data construction. In practical settings (Yuan et al., 2024), practitioners typically curate training datasets by selecting preference pairs with large, distinct margins. This selection process empirically contributes to a larger $\gamma$, effectively ruling out the vanishing margin scenario.
>
> Finally, the self-rewarding loop is expressly designed to **amplify** these small, inherent differences. The sharpening update ($\pi_t^{1+1/\beta}$) will further increase the probability of the (minutely) better sequence and suppress the other, thereby actively *increasing* the log-probability gap $\gamma$ over iterations.

---

### Official Review · Reviewer_kTQ9 · 2025-11-11

**Soundness:** 3
**Presentation:** 3
**Contribution:** 3
**Rating:** 6
**Confidence:** 3

**Summary:**

The paper theoretically tries to understand the iterative self-rewarding procedure in LLMs. It first shows that optimizing a DPO-style objective with one step can lead to a model with a high probability of failure (depending on the base model). It then shows that iterative self-rewarding alignment can exponentially (in reward steps $T$) suppress the effect of a bad initial model. It finally applies the results to a linear softmax model as an example.

**Strengths:**

The paper works on a very important and hot problem of understanding the iterative self-rewarding in LLMs. The paper is very well written and easy to follow, and the authors have spent time discussing their results thoroughly to make them digestible for the reader. I feel the results are very clean and show the effect of iterative self-rewarding alignment, which is the main goal of the work.

**Weaknesses:**

**Dependence on $\gamma$**: The finite-sample guarantee of iterative procedure (Theorem 3) depends on the margin $\gamma$ being strictly positive. However, $\gamma$ is defined as the infimum over all prompts and all iterations $T$. What prevents a 'margin collapse' ($\gamma \to 0$) during training—for instance, on a prompt where two responses are equally optimal? If $\gamma = 0$, doesn't the bound become vacuous and the guarantee of stability disappear? I feel like this is a strong assumption, one could easily have a bad initial point leading to a margin collapse.

Following up on this, does the self-rewarding process itself, $r_t = \log \pi_t(y|x)$, have any mechanism to prevent this margin collapse, or could it, in fact, cause it by optimizing two similar responses to have the same high probability? How robust is the theory to this hard-margin assumption?

**Tightness of Thm. 3**: Line 304 states that the bound in Thm. 3 is tight in its dependence on key parameters. For $T=1$ it is, as discussed. But for $T>1$, we don’t know, right? And it's not as if $T$ is not a key parameter of the problem. How can we say that the iterative decay is tight?

**On exponential stability and role of objective**: The paper's proof of exponential stability (Remark 5) fundamentally relies on deriving a recurrence where $\kappa_t$depends sub-linearly on $\kappa_{t-1}$ (i.e., as $\sqrt{\kappa_{t-1}}$). This dependency is what creates the contraction mapping. However, the proof also explicitly relies on the $\mathcal{O}(1/\sqrt{n})$ statistical error rate, which is a well-known property of the DPO/ERM loss (Eq. 3).

My question is: Is this stability-inducing $\sqrt{\kappa_{t-1}}$ relationship a fundamental property of the self-reward mechanism ($r_t = \log \pi_t$) itself, or is it an emergent property of its combination with a stable, low-variance objective like DPO? If, for example, the DPO objective were replaced with a different update rule that has a higher-variance or slower-converging statistical error, would the entire proof of a sub-linear recurrence collapse? This would seem to imply the paper's guarantees are not about self-rewarding in general, but only about the specific combination of self-reward and DPO.


**On the role of model complexity**: All results in the paper (Thm. 1, Thm. 3) are reminiscent of classical statistical learning theory, where model complexity (here $|\Pi|$) is a penalty that makes the error bound worse. Moving from classical to overparameterized models, people developed new results to explain benign overfitting. In their current shape, I don’t see the empirical reality of “scaling laws” for alignment reflected here, where larger models are better at alignment. In the linear softmax case, the authors introduced spectral decay in the feature covariance to deal with the curse of a larger model, which is one data model under which people explain benign overfitting in linear regression. I feel that connection is missing here and would be useful to the reader. Also, I am interested to hear the authors’ comments on how to translate their original results to reflect the role of a larger model.

**Questions:**

Please see weaknesses.

---

> ### Author Response · Authors · 2025-11-21
> **Response to Q1 (Part 1 of 2): Validity of the Margin Assumption ($\gamma > 0$)**
>
> Thank you for your thoughtful review and your kind support. We have carefully addressed all of your comments below and thoroughly revised the manuscript, with all changes highlighted in blue. We hope our responses satisfactorily resolve your concerns.
>
> **Q1: Dependence on $\gamma$: The finite-sample guarantee (Thm. 3) relies on a strictly positive margin, $\gamma > 0$. What prevents a 'margin collapse' ($\gamma \to 0$) during training—for instance, on a prompt where two responses are equally optimal? If $\gamma=0$, doesn't the bound become vacuous and the guarantee of stability disappear?**
>
> **Following up on this, does the self-rewarding process itself, $r_t = \log \pi_t(y|x)$, have any mechanism to prevent this margin collapse, or could it, in fact, cause it by optimizing two similar responses to have the same high probability? How robust is the theory to this hard-margin assumption?**
>
> **A:** We thank the reviewer for this insightful question regarding the margin assumption $\gamma > 0$, which is central to our analysis. Below, we clarify three key points that directly address this concern, including the role of our framework in avoiding margin collapse.
>
> 1. The assumption $\gamma > 0$ follows naturally from the unique optimal sequence assumption and from the statistical identifiability of the task, since a zero margin would make the optimal response indistinguishable from suboptimal ones. The case $\gamma = 0$ is also a measure-zero event in continuously parameterized models. In practice, preference datasets are further curated to include pairs with clear preference gaps (Yuan et al., 2024), which empirically maintains a strictly positive margin.
> 2. The self-reward mechanism $r_t = \log \pi_t(y|x)$ is not a cause of margin collapse, but is in fact the core dynamic that prevents it by amplifying small probability differences.
> 3. Our theory's stability guarantee is robust to this assumption. It is not dependent on a large $\gamma$. The core stability mechanism is the *contraction of the policy condition number* $\kappa_t$ (Remark 4), which acts as a self-correction “guardrail” orthogonal to the margin size, ensuring stability even if $\gamma$ is small.
>
> We elaborate on these points below.
>
> ### 1. The $\gamma > 0$ Assumption and the Case of $\gamma = 0$
>
> We thank the reviewer for this insightful query. Our finite-sample guarantee in Theorem 3 does indeed rely on a strictly positive margin, $\gamma > 0$. This is a direct theoretical consequence of our stated assumption of a unique optimal sequence $y^*(x)$ (Main text, Line 239; Appendix, Line 851). Moreover, if a true margin collapse ($\gamma = 0$) were to occur for a given prompt $x$ at iteration $t$, the bound of Theorem 3 would become vacuous for that specific instance, and its stability guarantee would not apply.
>
> However, we argue this assumption is not a practical limitation, and such a collapse is not observed in practice (e.g., Yuan et al., 2024) for three fundamental reasons. First, the assumption $\gamma > 0$ is intrinsically linked to the statistical identifiability and learnability of the task. If the margin were to approach zero ($\gamma \to 0$), it would imply that the optimal solution is statistically indistinguishable from suboptimal ones. In such a scenario, no algorithm, whether RLHF or our proposed SRLM, could effectively extract a learning signal from the data. Therefore, assuming $\gamma > 0$ is equivalent to the foundational assumption in statistical learning that a unique, identifiable optimal solution exists.
>
> Second, from a mathematical standpoint, an exact margin collapse is a measure-zero event for continuously parameterized models like Transformers. While a reviewer might intuit that response quality differences can be minuscule at the token level (e.g., “Yes” vs. “Right”), this near-equivalence does not translate to exact equality at the sequence level. For two distinct sequences $y_A$ and $y_B$, the probability of yielding identical log-probabilities, $\log \pi_\theta(y_A|x) = \log \pi_\theta(y_B|x)$, is mathematically zero.
>
> Third, strict positivity is often enforced by design during data construction. In practical settings (Yuan et al., 2024), practitioners typically curate training datasets by selecting preference pairs with large, distinct margins. This selection process empirically contributes to a larger $\gamma$, effectively ruling out the vanishing margin scenario. Finally, as we elaborate in the following section, the self-rewarding loop is expressly designed to amplify even minuscule, noise-induced differences rather than erase them, further mitigating this theoretical concern.

---

> ### Author Response · Authors · 2025-11-21
> **Response to Q1 (Part 2 of 2): Mechanism of Margin Amplification & Robustness**
>
> ### 2. The Self-Reward Mechanism Prevents, Not Causes, Margin Collapse
>
> Regarding this concern, the self-reward $r_t = \log \pi_t(y|x)$ does not induce margin collapse; it is in fact the main driver that prevents collapse and amplifies the margin $\gamma > 0$. Rather than pushing similar responses toward the same probability, it sharpens the distribution by forcing the model to “choose” between them, iteratively separating their probabilities through two key properties:
>
> - Log-Function Amplification: The use of $\log \pi_t$ as a reward inherently amplifies small differences in probability.
>   - A small probability difference (e.g., $p_1 = 0.51, p_2 = 0.49$; diff = 0.02) is transformed into a non-zero reward margin: $\Delta J_t = \log 0.51 - \log 0.49 \approx 0.04$.
>   - The DPO update (Eq. 3) then aims to find a $\pi_{t+1}$ whose log-ratio matches this non-zero $\Delta J_t$.
> - DPO Update Dynamics: When $\Delta J_t > 0$ (even if small), the optimization will push $\pi_{t+1}(y)$ higher and $\pi_{t+1}(y')$ lower to match this target. This converts the small reward difference from iteration $t$ into a larger probability difference in iteration $t+1$, thereby actively widening the margin.
>
> This positive-feedback loop spontaneously breaks ties and is validated by the empirical works we cite (e.g., Yuan et al., 2024), which observe consistent response orderings and no evidence of margin collapse.
>
> ### 3. Robustness of the Theory (The Role of $\kappa_t$)
>
> Regarding this concern, robustness does not rely on maintaining a large margin. The key stabilizing force is the contraction of the policy condition number $\kappa_t$. As shown in Remark 4, the iterative update first acts as a contraction mapping, pulling the model from a low-confidence, diffuse state into a well-conditioned regime. This self-correction mechanism is orthogonal to the size of $\gamma$. Even when certain prompts exhibit extremely small margins, the geometric convergence of $\kappa_t$ suppresses probability diffusion or collapse, ensuring stable training. This behavior follows directly from Remarks 3–4 and the recursive bounds established in Appendix B.
>
> **Revisions to the Manuscript**
> To address this point explicitly, we have added the following text after Definition 1 (Main Text Line 289):
>
> > The unique optimal sequence assumption implies a strictly positive margin, since identifiability requires the optimal response to be distinguishable from suboptimal ones. In continuously parameterized models, exact ties ($\gamma = 0$) form a measure-zero set and are therefore negligible in practice. Even when margins are extremely small, the iterative contraction of $\kappa_t$ ensures stability, while the DPO update and the self-reward mechanism amplify these small gaps over time, preventing collapse.
>
> Thank you again for this excellent question.

---

> ### Author Response · Authors · 2025-11-21
> **Response to Q2: Clarification on Tightness of Bounds (for $T > 1$)**
>
> **Q2: Line 304 states that the bound in Thm. 3 is tight in its dependence on key parameters. For $T=1$ it is, as discussed. But for $T>1$, we don’t know, right? And it's not as if $T$ is not a key parameter of the problem. How can we say that the iterative decay is tight?**
>
> **A:** Thank you for this very precise and insightful observation. You are absolutely correct, and we appreciate the careful reading. Our statement in Line 304 that “the bound is tight in its dependence on key parameters” was indeed intended to refer primarily to the dependence on the statistical sample size ($n$) and the initial policy condition number ($\kappa_0$), rather than a global claim that includes the iteration rounds ($T$).
>
>  Your observation is spot on: the tightness for the $T=1$ case is explicitly validated by its precise match with the single-step lower bound in Theorem 1. For the $T>1$ case, you are correct that we have not provided a corresponding constructive lower bound, and the tightness is unknown. Proving the formal information-theoretic optimality of the $T$-dependence would require a more complex analysis, such as demonstrating that the recursive decay rate
> $
> \kappa_T \le U + q^T(\kappa_0 - U)
> $
> cannot be improved. This is a non-trivial problem that falls outside the scope of our current analysis, and we consider it an excellent open question for future work. To avoid this ambiguity and make our claim precise, we will revise the paragraph in the manuscript as follows:
>
> **Revisions to Remark 3 (Line 336 and Line 343 of the manuscript).**
> In particular, we have added the following sentence:
>
> > Furthermore, the bound's dependence on $\kappa_0$ and $n$ is shown to be tight for the single-step ($T=1$) case. However, we do not provide a matching lower bound for $T>1$ to prove the tightness of the exponential decay rate itself. This remains an interesting open question that we leave for future work.

---

> ### Author Response · Authors · 2025-11-21
> **Response to Q3: Intrinsic Stability of Self-Reward vs. DPO Specifics**
>
> **Q3: Is this stability-inducing $\sqrt{\kappa_{t-1}}$ relationship a fundamental property of the self-reward mechanism ($r_t = \log \pi_t$) itself, or is it an emergent property of its combination with a stable, low-variance objective like DPO? If, for example, the DPO objective were replaced with a different update rule that has a higher-variance or slower-converging statistical error, would the entire proof of a sub-linear recurrence collapse? This would seem to imply the paper's guarantees are not about self-rewarding in general, but only about the specific combination of self-reward and DPO.**
>
> **A:** We appreciate this insightful question addressing a core conceptual contribution of our paper. We clarify that the existence of a contraction in the policy condition number $\kappa_t$ is fundamentally driven by the self-reward mechanism itself ($r_t = \log \pi_t$), while the update rule (e.g., DPO) governs the rate and tightness of this contraction. Consequently, replacing DPO with a higher-variance or slower-converging update would degrade the constants, but the fundamental contraction structure would not collapse. Thus, our guarantees extend beyond the specific “self-reward + DPO” combination to the broader class of updates satisfying high-probability drift and controlled statistical error. We focus our analysis on DPO primarily due to its widespread adoption, ensuring our results are directly relevant to current practice.
>
> ### 1. The Intrinsic Contraction Dynamics of Self-Rewarding
>
> The primary driver of this contraction is the reward definition $r_t(y|x) = \log \pi_t(y|x)$, which introduces a log-ratio structure into every update step. Since the differences in log-probabilities decompose into  $\Delta r_t(x,y,y') = \log \frac{\pi_t(y|x)}{\pi_t(y'|x)}$,  the analysis of KL-regularized updates naturally yields terms involving $\sqrt{\kappa_{t-1}}$. Crucially, this result is independent of the specific update algorithm.
>
> Intuitively, the self-rewarding mechanism sharpens the model’s distribution by reinforcing high-probability regions and suppressing diffuse tails. This property pushes the policy condition number  $\kappa_t$  toward smaller values. In summary, the qualitative contraction phenomenon, where $\kappa_t$ forgets its initialization and converges to a bounded region, is an intrinsic dynamical effect of self-rewarding rather than a DPO-specific feature.
>
> ### 2. The Role of DPO in Determining Constants
>
> Although the self-rewarding mechanism dictates the structural form of the recurrence presented in Lemma 9 \& 10, the specific magnitudes of the constants $M_1$ and $M_2$ depend explicitly on the properties of DPO:
> $$
> \kappa_t \leq M_0 + M_1\sqrt{\kappa_{t-1}} + M_2\log\kappa_{t-1}.
> $$
> Specifically, $M_1$ arises from the statistical error associated with estimating reward differences and scales as $O(1/\sqrt{n})$ in the context of DPO, while $M_2$ captures the effective step size induced by the KL regularization inherent to DPO. These constants fundamentally govern the contraction factor $q$ and the fixed-point region $U$, as established in Lemma 10 and Theorem 3. Consequently, DPO influences how fast the contraction proceeds and how tight the stable region becomes. However, it does not determine the existence of the contraction itself or the sub-linear structure of the recurrence, which remain intrinsic properties of self-rewarding.
>
> ### 3. Robustness to Slower-Converging Updates
> Our structural analysis shows that the recurrence is robust to such variations: it persists as long as an alternative update rule satisfies two minimal conditions, namely (1) **high-probability drift**, meaning the update consistently pushes probability mass toward high self-reward regions in expectation, and (2) **bounded statistical error**, meaning the update remains learnable even when its convergence rate is slower. For example, if we consider an alternative update with statistical error $O(n^{-\alpha})$ where $\alpha \le 1/2$, the constant $M_1$ then scales as $M_1 \propto n^{-\alpha}$, yet the recurrence still retains exactly the same sub-linear form.
>
> $$
> \kappa_t \le M_0 + M_1\sqrt{\kappa_{t-1}} + M_2\log \kappa_{t-1}.
> $$
>
> Applying the inequality $\log x \le \frac{2}{e}\sqrt{x}$ for $x \ge 1$ (Eq. 68), Lemma 9 simplifies the recurrence to the contraction mapping defined in Eq. (70):
>
> $$
> \kappa_t \le M_0 + K\sqrt{\kappa_{t-1}}, \quad K := M_1 + \frac{2}{e} M_2.
> $$
>
> This mapping remains valid for any finite positive constants $M_1$ and $M_2$, a condition satisfied by all proximal updates with bounded statistical error. The recurrence would collapse only if the statistical error became unbounded or if KL stability were removed, as such failures would violate the derivation of Eq. (61). A slower convergence rate merely increases $K$ and moves $q$ closer to 1 (see Lemma 10), but it does **not** alter the fundamental sub-linear structure that underpins the contraction.

---

> ### Author Response · Authors · 2025-11-21
> **Response to Q4: Reconciling Complexity Bounds with Scaling Laws (Benign Overfitting)**
>
> **Q4: All results in the paper (Thm. 1 and 3) treat model complexity as a classical penalty, which seems to contradict empirical scaling laws showing that larger models improve alignment. Although the linear softmax analysis uses spectral decay, a technique related to benign overfitting, the connection is not explained. Clarifying this link would help readers understand how the results extend to the observed benefits of larger models.**
>
> **A:** We thank the reviewer for this insightful question. We agree that our classical complexity terms ($|\Pi|$) act as worst-case penalties. This perspective, where larger models yield looser bounds, indeed appears to contradict empirical scaling laws. This contradiction arises because classical bounds ignore data geometry, which is precisely why **Corollary 6** is essential to our framework.
>
> ### 1. Reconciling Bounds via Spectral Decay and Benign Overfitting
> As the reviewer astutely noted, the key to resolving this tension lies in the spectral decay assumption introduced in Corollary 6, which connects our results to the phenomenon of benign overfitting. Corollary 6 demonstrates that when the feature covariance matrix $\Sigma_\varphi$ exhibits rapid (e.g., exponential) spectral decay, the generalization bound no longer depends on the massive ambient dimension $d$. Instead, it is controlled by a much smaller effective dimension $d_{eff}$. (This transition is detailed in Definition 6, Lemma 13, and Corollaries 14–15.) For instance, with exponential decay and a standard $\lambda \asymp c/n$ regularization, we achieve $d_{eff} \lesssim \log n$. This mechanism is the mathematical manifestation of benign overfitting: the model is enormous in its *parameter space* (large $d$) but operates in a low-dimensional *function space* (small $d_{eff}$).
>
> ### 2. Translating Results to Reflect the Role of Larger Models (Scaling Laws)
>
> This leads directly to our comment on how to translate these results to reflect the observed benefits of larger models. Our theory suggests the essence of scaling laws is *not* merely “more parameters ($d$) are better.” Rather, it is that larger models (larger $d$) possess the capacity to learn feature representations $\varphi$ with faster spectral decay, which in turn yields a smaller effective dimension $d_{eff}$. This learned, compact representation is the true source of improved generalization and alignment. This logic is clarified by contrasting two model regimes:
>
> - **Underparameterized models.** These models lack the capacity to learn compact representations. Their learned features $\varphi$ likely exhibit slow (e.g., polynomial, $\lambda_i \le C i^{-p}$ for small $p$) spectral decay, leading to a large effective dimension (e.g., $d_{eff} \asymp n^{1/p}$). Such models suffer from the curse of dimensionality, where the bound scales with this large $d_{eff}$.
> - **Overparameterized (large) models.** The flexibility afforded by overparameterization allows the model to explore a vast space of representations and converge to an optimal one that satisfies the “benign” exponential decay structure (Corollary 6). Once this representation is learned, the generalization bound is governed by the minimal $d_{eff} \lesssim \log n$, *not* the ambient parameter count $d$.
>
> Thus, our framework does not contradict scaling laws; it provides a formal explanation for *why* they occur: empirical “scaling” is the process of a large model successfully leveraging its overparameterization to find a benign, low-dimensional spectral structure (Corollary 6) in the task.
>
> ### 3. Limitations and Future Work
>
> We also candidly acknowledge the difficulty of extending these rigorous guarantees to full, modern Transformer architectures (including multi-head attention and deep residual connections). Establishing a complete benign overfitting theorem for such complex, non-linear, non-convex systems is a major open problem in the theoretical community. Current rigorous theories are largely confined to linear models (as in our setting) or two-layer neural networks. Extending this framework to full Transformers is a highly valuable direction for future work but is beyond the scope of this paper.
>
> ### 4. Revisions to the Manuscript
>
> To make this link, which we agree is critical, explicit for the reader, we have revised **Remark 8** (following Corollary 6) to include the following discussion:
>
> > The key insight is that generalization is governed by the effective dimension $d_{eff}(\lambda)$, not the ambient dimension $d$. Scaling laws emerge because larger models have the capacity to learn feature representations with faster spectral decay (i.e., a smaller $d_{eff}$). This aligns with benign overfitting, where models generalize well despite over-parameterization by using a low-dimensional signal structure. While our guarantees hold for linear softmax models, extending this analysis to full Transformers is a significant open challenge and a valuable direction for future work.

---

### Meta-Review · Area_Chair_rbxP · 2025-12-11

**Summary:**

This paper presents a clean finite-sample analysis of a stylized self-rewarding process and introduces a policy condition number to study convergence. The technical results are sound and the reviewers find the exposition clear. However, the core theoretical setting, reward defined as the model’s own log-likelihood, is a highly simplified abstraction that departs significantly from modern SRLM practice, where LLM-as-a-Judge, rubric-based scoring, and multi-turn evaluation produce fundamentally different reward dynamics. Because this reward structure removes collapse by construction and induces a deterministic self-sharpening dynamic, the resulting convergence guarantee reflects optimization of an internally constructed objective rather than any meaningful notion of alignment or performance. As such, convergence to this fixed point does not carry semantic significance and does not inform real SRLM behavior. The analysis therefore does not explain the observed stability or collapse phenomena in practical systems.

**Reviewer Concerns:**

Regarding reviewer concerns, the rebuttal successfully clarified several technical points raised by the positive reviewers (e.g., the interpretation of the margin assumption, the tightness discussion, the role of DPO in the recurrence, and the connection to effective dimension), and the authors provided a small illustrative experiment addressing requests for an empirical check of κₜ. However, the core concerns raised by Reviewer uGyn; namely the limited scope of the reward model, the lack of applicability to practical SRLM pipelines, and the resulting disconnect between the theoretical framework and real-world SRLM behavior; remain unresolved.

**Reviewer Scores:**

No reviewer is likely to have increased their score following the discussion.

---

### Decision · Program_Chairs · 2026-01-26

Reject